# URS: A U̲nified Neural R̲outing S̲olver for Cross-Problem Zero-Shot Generalization

## Abstract

Multi-task neural routing solvers have emerged as a promising paradigm for their ability to solve multiple vehicle routing problems (VRPs) using a single model. However, existing neural solvers typically rely on predefined problem constraints or require per-problem fine-tuning, which substantially limits their zero-shot generalization ability to unseen VRP variants. To address this critical bottleneck, we propose URS, a unified neural routing solver capable of zero-shot generalization across a wide range of unseen VRPs using a single model without any fine-tuning. The key component of URS is the unified data representation (UDR), which replaces problem enumeration with data unification, thereby broadening the problem coverage and reducing reliance on domain expertise. In addition, we propose a Mixed Bias Module (MBM) to efficiently learn the geometric and relational biases inherent in various problems. On top of the proposed UDR, we further develop a parameter generator that adaptively adjusts the decoder and bias weights of MBM to enhance zero-shot generalization. Moreover, we propose an LLM-driven constraint satisfaction mechanism, which translates raw problem descriptions into executable stepwise masking functions to ensure solution feasibility. Extensive experiments demonstrate that URS can consistently produce high-quality solutions for more than 100 distinct VRP variants composed of 10 different constraints without any fine-tuning, which includes more than 90 unseen variants. To the best of our knowledge, URS is the first neural solver capable of handling over 100 VRP variants with a single model.

## 1 Introduction

The Vehicle Routing Problem (VRP) is an essential class of combinatorial optimization problems (COPs) with extensive applications in logistics and supply chain management (Tiwari & Sharma, 2023; Sar & Ghadimi, 2023). Solving VRPs efficiently is challenging due to their NP-hard nature. While exact solvers can produce optimal solutions, they often become computationally prohibitive for real-world scenarios. In recent decades, classical heuristic solvers (Helsgaun, 2017; Vidal, 2022) have achieved impressive performance within acceptable timeframes. Nevertheless, these methods require considerable domain expertise to design specialized rules for each routing problem. Given the growing diversity of VRP variants in real-world applications, manually crafting tailored rules for every case has become impractical.

In recent years, neural combinatorial optimization (NCO) methods have attracted substantial attention for their potential to reduce reliance on handcrafted rules while maintaining competitive solution quality (Bengio et al., 2021; Li et al., 2022a). This has led to the development of many high-performing neural routing solvers that can automatically learn implicit problem-specific patterns from data under different training paradigms such as supervised learning (SL) (Drakulic et al., 2023; Luo et al., 2023; 2025b), reinforcement learning (RL) (Bello et al., 2016; Kool et al., 2019; Zhou et al., 2024a), and emerging self-improved learning (SIL) (Luo et al., 2023; 2025a; Pirnay & Grimm, 2024). Although these solvers have demonstrated impressive performance on specific problems (Kwon et al., 2020; Kim et al., 2022; Zhou et al., 2025), they typically require architectural customization and per-problem retraining to accommodate the distinct constraints and features of different VRP variants. These limitations increase the overall training cost and hinder practical deployment on new problems.

Table 1: Comparison between our URS and existing neural solvers with multi-task learning.

| Neural Solver | Training Paradigm | Routing Problems (#) | Zero-shot Generalization | Remarks |
|---|---|---|---|---|
| MTPOMO (Liu et al., 2024a) | RL | 16 | ✓ | Constraint Combination |
| MVMoE (Zhou et al., 2024b) | RL | 16 | ✓ | Constraint Combination |
| RouteFinder (Berto et al., 2024) | RL | 48 | ✓ | Constraint Combination |
| CaDA (Li et al., 2025a) | RL | 16 | ✓ | Constraint Combination |
| MTL-KD (Zheng et al., 2025a) | RL+KD | 16 | ✓ | Constraint Combination |
| TSP-FT (Lin et al., 2024) | RL | 3 | × | Adapter-based Fine-tuning |
| MTL-MAB (Wang et al., 2025a) | RL | 3 | × | Adapter-based Fine-tuning |
| GOAL (Drakulic et al., 2025) | SL | 9 | × | Adapter-based Fine-tuning |
| **URS (Ours)** | **RL** | **107** | **✓** | **Unified Data Representation** |

To address the challenge of cross-problem generalization, growing attention has been directed toward multi-task learning capable of handling diverse routing problems. As summarized in Table 1, existing efforts largely fall into two categories: 1) constraint combination-based multi-task learning and 2) adapter-based fine-tuning frameworks. In the first approach, VRP variants are treated as different combinations of constraint attributes, and a unified model is trained across these combinations to enable knowledge sharing for problems with seen constraints (Liu et al., 2024a; Zhou et al., 2024b; Li et al., 2025a; Berto et al., 2024; Zheng et al., 2025a). The second approach builds a shared model backbone for all problems and incorporates specific input/output adapters for each problem, thereby reducing the retraining costs (Lin et al., 2024; Drakulic et al., 2025; Wang et al., 2025a).

Despite the above advancements, existing methods still fall short in zero-shot generalization. The problem coverage of constraint combination-based approaches is inherently bounded by their manually specified constraint sets, while adapter-based approaches require fine-tuning and thus cannot perform zero-shot generalization. Moreover, both strategies fundamentally rely on the explicit enumeration of problems through a predefined set of problem tags. This practice is problematic because the constraint space of routing problems is open-ended and compositional, making any fixed tagging scheme fundamentally limited. More critically, creating and maintaining such a problem taxonomy requires considerable domain expertise, which is precisely what NCO always aims to avoid. A detailed review of related work on single-task and multi-task learning is provided in Appendix A.

In this paper, we propose a powerful **U**nified Neural **R**outing **S**olver (URS) to significantly improve the cross-problem zero-shot generalization ability for NCO methods. Unlike existing approaches, we introduce a unified data representation (UDR) that replaces problem enumeration with data unification, thereby broadening problem coverage and reducing reliance on domain expertise. In addition, we propose a Mixed Bias Module (MBM) to better learn the geometric and relational biases inherent in various problems. Based on UDR, we further develop a parameter generator that adaptively adjusts decoder and bias weights of MBM to improve zero-shot generalization. Furthermore, to handle complex and unseen routing problems, we introduce a constraint satisfaction mechanism that leverages a large language model (LLM) to translate raw problem constraint descriptions into executable node masking functions for each construction step, thereby ensuring our URS generates feasible solutions for various problems.

We evaluate URS on **more than 100** VRP variants composed of 10 different constraints, including **more than 90** unseen variants. The results demonstrate that URS not only obtains competitive performance against specialized neural solvers on seen variants but also exhibits strong zero-shot generalization capabilities on a wide range of unseen problems. To the best of our knowledge, URS is the first neural solver capable of efficiently solving over 100 distinct VRP variants using a single model without any retraining or fine-tuning.

## 2 PRELIMINARIES

In this section, we first introduce the definition of VRP, then provide an overview of recent constructive neural solvers for solution generation (Kool et al., 2019; Kwon et al., 2020).

### 2.1 VEHICLE ROUTING PROBLEMS

Consider a VRP instance with an optional depot indexed by $0$ and $n$ customers indexed by $\{1, 2, \ldots, n\}$, such as in the Capacitated VRP (CVRP). The instance can be represented as a graph

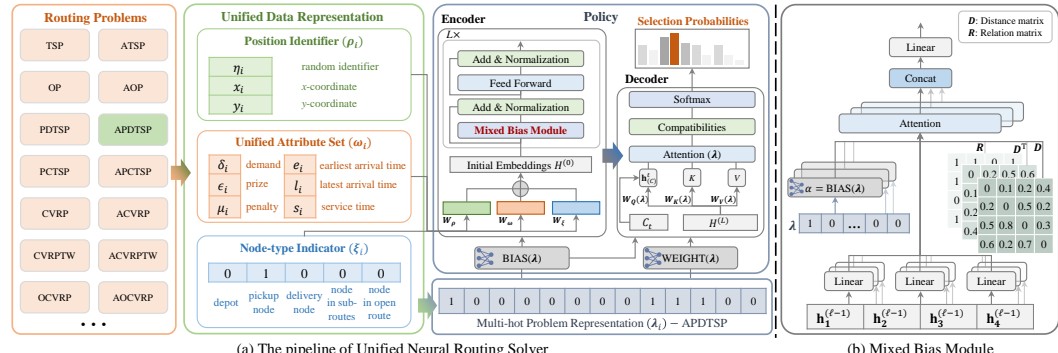

Figure 1: The overview of our URS with a 4-node APDTSP instance as an example. (a) The solving pipeline of URS; (b) The proposed mixed bias module that incorporates multiple geometric priors, enabling it to efficiently learn the geometric and relational biases inherent in various problems.

$\mathcal{G} = (\mathcal{V}, \mathcal{E})$, where the node set $\mathcal{V} = \{v_i\}_{i=0}^n$ has a total size of $|\mathcal{V}| = 1 + n$ unless the variant has no depot (e.g., the Traveling Salesman Problem), and the edge set is $\mathcal{E} = \{e(v_i, v_j) \mid v_i, v_j \in \mathcal{V}, v_i \neq v_j\}$. The travel cost between nodes is given by a distance matrix $\boldsymbol{D} = \{d_{ij} \mid \forall\, i, j \in 0, \ldots, n\}$. In VRP, each node $v_i \in \mathcal{V}$ includes node coordinates $\{x_i, y_i\}$ when available and problem-specific attributes (e.g., demands in CVRP). A feasible solution of a VRP calls for the determination of a set of routes, each performed by a single vehicle that starts and ends at its own depot or starting node, all the operational constraints are satisfied, denoted as $\boldsymbol{\pi} = (\pi_1, \pi_2, \ldots, \pi_m)$ (Toth & Vigo, 2002) (i.e., a finite sequence), which is a permutation of $m$ nodes. Let $\Omega$ denote the feasible sequence set that satisfies the constraints, given an objective function $f(\boldsymbol{\pi}|\mathcal{G})$, we aim to search for a sequence $\boldsymbol{\pi}^*$ with the maximum objective, i.e., $\boldsymbol{\pi}^* = \arg\max_{\boldsymbol{\pi} \in \Omega} f(\pi|\mathcal{G})$. In most routing problems, the objective can be represented as the negative value of the total distance of a feasible sequence $\boldsymbol{\pi}$.

In this paper, we consider over 100 VRP variants which may simultaneously have one or more constraints from the following categories: (1) Capacity (C); (2) Open Route (O); (3) Backhaul (B); (4) Backhaul and Priority (BP); (5) Duration Limit (L); (6) Time Windows (TW); (7) Multi-Depot (MD); (8) Prize Collecting (PC); (9) Asymmetric (A); (10) Pickup and Delivery (PD). Detailed definitions of these constraints are provided in Appendix B.

## 2.2 CONSTRUCTIVE NEURAL ROUTING SOLVER

Most constructive NCO methods employ an encoder-decoder architecture for solution construction (Luo et al., 2023; Kwon et al., 2020). Let learnable model parameters be denoted as $\boldsymbol{\theta} = \{\boldsymbol{\theta}_{enc}, \boldsymbol{\theta}_{dec}\}$. Without loss of generality, we present the prevailing construction pipeline using AM (Kool et al., 2019). Given an instance $\mathcal{G}$, raw node features are first mapped by a linear projection to initial embeddings $H^{(0)} = \{\mathbf{h}_i^{(0)}\}_{i=0}^n$. They are then processed by an encoder $\boldsymbol{\theta}_{enc}$ with $L$ attention layers to produce refined node embeddings $H^{(L)} = \{\mathbf{h}_i^{(L)}\}_{i=0}^n$. At each decoding step $t$, the decoder $\boldsymbol{\theta}_{dec}$ sequentially selects a node to append to the current partial solution $\pi_{1:t-1} = (\pi_1, \pi_2, ..., \pi_{t-1})$, where $\pi_1, \pi_{t-1} \in \mathcal{V}$ are the **first** and **last** visited node, respectively. This sequential process continues until a complete solution is formed. To guarantee the feasibility of generated solutions, a masking function $\mathcal{M}_t$ sets the selection probabilities of the following nodes to $-\infty$ during the construction process: nodes that (1) are visited or (2) violate problem-specific constraints (e.g., capacity).

## 3 THE PROPOSED MODEL: UNIFIED NEURAL ROUTING SOLVER

In this section, we propose a powerful **U**nified Neural **R**outing **S**olver (URS), which can significantly improve cross-problem zero-shot generalization for NCO methods. The framework is illustrated in Figure 1, and its key components are elaborated below.

## 3.1 Unified Data Representation

Existing multi-task neural solvers (Liu et al., 2024a; Zhou et al., 2024b; Drakulic et al., 2025; Wang et al., 2025a) fundamentally rely on the explicit problem enumeration through a predefined set of problem tags, which is challenging because the constraint space of routing problems is open-ended and compositional, making any fixed tagging scheme fundamentally limited. More critically, creating and maintaining such a problem taxonomy requires considerable domain expertise, which is precisely what NCO always aims to avoid.

From a data perspective, despite the substantial diversity in constraints among routing problems, their instances share a common instance representation, e.g., all input instances under the current constraint combinations (Liu et al., 2024a; Zhou et al., 2024b) can be formulated as instances of either CVRP or CVRPTW. Leveraging this commonality, we propose a Unified Data Representation (UDR) to replace problem enumeration with data unification, thereby broadening problem coverage and reducing reliance on domain expertise. For an instance $\mathcal{G}$, we define the unified representation $\boldsymbol{U} = \{\mathbf{u}_i\}_{i=0}^n$, each $\mathbf{u}_i = \{\boldsymbol{\rho}_i, \boldsymbol{\omega}_i, \boldsymbol{\xi}_i\}$, where $\boldsymbol{\rho}_i, \boldsymbol{\omega}_i, \boldsymbol{\xi}_i$ are a position identifier, unified attribute set, and node-type indicator, respectively.

**Position Identifier ($\boldsymbol{\rho}_i$)**     We define a unified positional identifier for each node as $\boldsymbol{\rho}_i = \{\eta_i, x_i, y_i\} \in [0,1]^3$. The $\eta_i \sim \mathrm{Uniform}(0,1) \in \mathbb{R}^1$ is an additional sampled scalar, which is used to address asymmetric problems, following Drakulic et al. (2023). For symmetric cases, we simply set $\eta_i = 0$.

**Unified Attribute Set ($\boldsymbol{\omega}_i$)**     We define the unified attribute set as $\boldsymbol{\omega}_i = \{\delta_i, \epsilon_i, \mu_i, e_i, l_i, s_i\}$, where they correspond to demand, prize, penalty, earliest arrival time, latest arrival time, and service time, respectively. The unification empowers the model to learn the relative importance of individual attributes in a shared embedding space. Furthermore, it supports effortless extension or ablation of attributes through simple zero-filling, which eliminates the need for changes to the model architecture.

**Node-type Indicator ($\boldsymbol{\xi}_i$)**     We provide an additional 5-dimensional binary vector $\boldsymbol{\xi}_i \in \{0,1\}^5$ for each node, encoding: depot, pickup node, delivery node, node in sub-routes, node in open route. The nodes in sub-routes and open routes mean the solution $\pi$ can have sub-routes or be an open route. We introduce explicit structural roles via $\boldsymbol{\xi}_i$ that help the model generalize over diverse routing problems. Further ablation analysis concerning $\boldsymbol{\xi}_i$ can be found in Appendix K.1.

**Relation Matrix ($\boldsymbol{R}$)**     To facilitate the handling of Pickup and Delivery (PD) problems, we introduce an optional relation matrix $\boldsymbol{R}$, which is applied exclusively to variants involving PD constraints. We define $\boldsymbol{R} = \{r_{ij} | \forall i, j \in 0, \ldots, n\}$, where $r_{ij} = 0$ if a predefined relation exists (e.g., pickup–delivery pairing in PDPs) and 1 otherwise. This ensures that smaller values are preferred, aligning with the distance metric where a smaller value corresponds to a larger selection bias.

**Multi-hot Problem Representation ($\boldsymbol{\lambda}_i$)**     Through the UDR, each problem instance activates only a specific subset of the unified feature space. Thus, for any instance $\mathcal{G}_i$ drawn from different problems, a corresponding multi-hot problem representation $\boldsymbol{\lambda}i$ is obtained by indicating active (non-zero) feature slots with a value of 1. This representation captures only the presence of features, in contrast to their raw values, and it subsequently provides conditional guidance to the position bias weight (see Equation (6)) and the adaptive decoder parameters $\boldsymbol{\theta}_{dec}(\boldsymbol{\lambda})$ (see Equation (8) and Equation (9)), thereby helping the model in distinguishing problem variants and refining its node selection process.

Instead of relying on a predefined problem taxonomy with a discrete set of problem tags, our proposed UDR decouples instance representation from constraint definitions and operates in a unified space characterized by both continuous and discrete features. It allows a single neural solver to address a much larger, open-ended space of VRP variants, as new problems can be seamlessly incorporated into this unified representation. Problem-specific constraints are delegated to the model-agnostic masking function $\mathcal{M}_t$. For example, for all capacity-constrained problems, only the remaining load is explicitly provided as input, while diverse additional constraints (e.g., time windows, duration limit, and backhaul) are enforced implicitly by masking infeasible candidate nodes during solution construction, rather than being fully enumerated in the decoder's input. This process is further automated by an LLM-driven constraint satisfaction mechanism that generates problem-specific mask generators, which convert constraint handling from manual engineering to automated programming

(see Section 4 for more details). Together, our proposed UDR enables a single model to handle various explored and even unexplored VRP variants while significantly reducing reliance on domain expertise. Detailed descriptions of $U$ and $\boldsymbol{\lambda}$ are provided in Appendix C and Appendix D, respectively.

## 3.2 MODEL ARCHITECTURE

As shown in Figure 1, we adopt AM (Kool et al., 2019) as our basic model. While UDR provides cross-problem consistency, different problems have diverse geometric priors. To efficiently learn the geometric and relational biases inherent in various problems and obtain better generalization performance, we propose two key enhancements: (1) Mixed Bias Modules (MBM) that integrate diverse geometric priors, and (2) adaptive parameter generators conditioned on the UDR. Their implementations are detailed below.

**Embedding Layer** Given an instance $\mathcal{G}$, for each $\mathbf{u}_i = \{\boldsymbol{\rho}_i, \boldsymbol{\omega}_i, \boldsymbol{\xi}_i\}$, we project each component into $d$-dimensional embeddings through three separate linear transformations:

$$\mathbf{h}_i^{(0)} = \boldsymbol{\rho}_i W_{\boldsymbol{\rho}} + \boldsymbol{\omega}_i W_{\boldsymbol{\omega}} + \boldsymbol{\xi}_i W_{\boldsymbol{\xi}}, \tag{1}$$

where $W_{\boldsymbol{\rho}} \in \mathbb{R}^{3 \times d}$, $W_{\boldsymbol{\omega}} \in \mathbb{R}^{6 \times d}$, $W_{\boldsymbol{\xi}} \in \mathbb{R}^{5 \times d}$ are learnable matrices. Then we obtain a set of initial embeddings $H^{(0)} = \{\mathbf{h}_i^{(0)}\}_{i=0}^n$ for all nodes in instance $\mathcal{G}$.

**Encoder** The encoder $\boldsymbol{\theta}_{enc}$ with $L$ stacked attention layers transforms $H^{(0)}$ into advanced node embeddings $H^{(L)} = \{\mathbf{h}_i^{(L)}\}_{i=0}^n$. The detailed process is provided in Appendix E.1. Unlike prior work, we replace the plain attention (Vaswani et al., 2017) with the proposed mixed bias module (MBM) that incorporates three structural bias matrices via three separate attention calculations: (1) an outgoing distance matrix $\boldsymbol{D}$; (2) an incoming distance matrix $\boldsymbol{D}^{\mathbf{T}}$; and (3) optional relation matrix $\boldsymbol{R}$, For the $\ell$-th layer, the MBM output $\hat{\mathbf{h}}_i^{(\ell)}$ can be expressed as:

$$\bar{\mathbf{h}}_i^{(0)} = \mathrm{Attention}(\mathbf{h}_i^{(\ell-1)}, H^{(\ell-1)}, f\left(\alpha, N, \boldsymbol{D}_i\right)), \tag{2}$$

$$\bar{\mathbf{h}}_i^{(1)} = \mathrm{Attention}(\mathbf{h}_i^{(\ell-1)}, H^{(\ell-1)}, f\left(\alpha, N, \boldsymbol{D}_i^{\mathrm{T}}\right)), \tag{3}$$

$$\bar{\mathbf{h}}_i^{(2)} = \begin{cases} \mathrm{Attention}(\mathbf{h}_i^{(\ell-1)}, H^{(\ell-1)}, f\left(\alpha, \boldsymbol{R}_i\right)) & \text{if } \boldsymbol{R} \neq \emptyset \\ \mathbf{0} & \text{otherwise,} \end{cases} \tag{4}$$

$$\hat{\mathbf{h}}_i^{(\ell)} = \left[\hat{\mathbf{h}}_i^{(0)}, \hat{\mathbf{h}}_i^{(1)}, \hat{\mathbf{h}}_i^{(2)}\right] W^O, \tag{5}$$

where $[\cdot, \cdot]$ denotes the horizontal concatenation operator, $W^O \in \mathbb{R}^{(3 \times d) \times d}$ is a learnable matrices. For $\mathrm{Attention}(\cdot)$, we adopt the Adaptation Attention Free Module (AAFM) (Zhou et al., 2024a), which boosts the perception of diverse geometric patterns through an adaptation function $f(\alpha, N, d_{ij})$ with bias weight $\alpha$ (see Appendix F for implementation details). We generate $\alpha$ via a lightweight bias network $\mathrm{BIAS}(\boldsymbol{\lambda})$ whose input is the instance's multi-hot representation $\boldsymbol{\lambda}$:

$$\mathrm{BIAS}(\boldsymbol{\lambda}) = \max\left(1, \left(\boldsymbol{\lambda} W_1 + \mathbf{b}_1\right) W_2 + \mathbf{b}_2\right), \tag{6}$$

where $W_1 \in \mathbb{R}^{|\boldsymbol{\lambda}| \times d}$, $W_2 \in \mathbb{R}^{d \times 1}$, $\mathbf{b}_1 \in \mathbb{R}^d$, $\mathbf{b}_2 \in \mathbb{R}^1$ are learnable parameters. We set the minimum value of the generated bias to 1, resulting in faster model convergence. Due to the relation $r_{ij}$ is scale-independent, we remove scale $N$ in adaptation function for $\boldsymbol{R}$, and if $\boldsymbol{R}$ is absent, we substitute a zero vector for $\bar{\mathbf{h}}_i^{(2)}$.

As shown in Figure 1(b), the Mixed Bias Module (MBM) is introduced to address the limitations of standard attention mechanisms in capturing geometric and relational biases inherent in diverse routing problems. For example, effectively representing asymmetric distances in AVRPs or pairing relationships in PDPs remains challenging, which hinders the achievement of high-quality solutions across a wide range of problems. While prior work, such as MatNet (Kwon et al., 2021), has proposed addressing this limitation by incorporating distance biases from different directions in separate parallel layers, this design leads to high model complexity and is only applicable to ATSP. In contrast, our MBM innovatively integrates multiple geometric priors (e.g., $D$, $D^T$, relation matrix $R$) within a single shared attention layer. This integrated design enhances flexibility while reducing architectural redundancy.

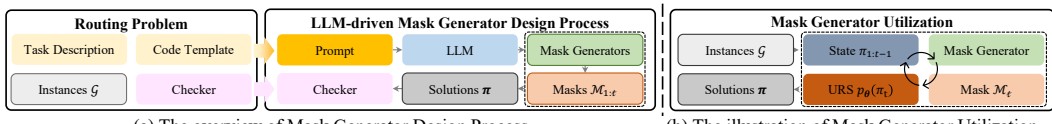

(a) The overview of Mask Generator Design Process   (b) The illustration of Mask Generator Utilization

Figure 3: The overview of LLM-driven Constraint Satisfaction Mechanism. (a) The generating pipeline of LLM-driven Constraint Satisfaction Mechanism for a general routing problem; (b) The illustration of mask generator utilization, handling the complex constraint.

**Decoder** In URS, all parameters of $\boldsymbol{\theta}_{dec}(\boldsymbol{\lambda})$ are generated based on the multi-hot problem representation $\boldsymbol{\lambda}$. Following Kwon et al. (2020), given the first and last visited node embeddings $\mathbf{h}_{\pi_1}^{(L)}$ and $\mathbf{h}_{\pi_{t-1}}^{(L)}$, and an optional constraint state $C_t \in \mathbb{R}^1$ (e.g., remaining load), the context embedding $\mathbf{h}_{(C)}^t$ can be expressed as:

$$\mathbf{h}_{(C)}^t = \left[\mathbf{h}_{\pi_1}^{(L)}, \ \mathbf{h}_{\pi_{t-1}}^{(L)}, \ C_t\right] W_Q(\boldsymbol{\lambda}), \tag{7}$$

where $[\cdot, \cdot]$ is the horizontal concatenation operator, $W_Q(\boldsymbol{\lambda}) \in \mathbb{R}^{(2d+1) \times d}$ is a linear projection matrix conditioned on $\boldsymbol{\lambda}$. Missing $C_t$ is zero-padded (see Appendix E.3 for detailed settings). The new context embedding $\hat{\mathbf{h}}_{(C)}^t$ is obtained via $\mathrm{Attention}(\cdot)$ (Zhou et al., 2024a) on $\mathbf{h}_{(C)}^t$ and $H^{(L)}$:

$$\hat{\mathbf{h}}_{(C)}^t = \mathrm{Attention}(\mathbf{h}_{(C)}^t, H^{(L)} W_K(\boldsymbol{\lambda}), H^{(L)} W_V(\boldsymbol{\lambda}), \mathcal{M}_t, f(\alpha, N, d_{t,i})), \tag{8}$$

where $W_K(\boldsymbol{\lambda}) \in \mathbb{R}^{d \times d}$, $W_V(\boldsymbol{\lambda}) \in \mathbb{R}^{d \times d}$ are linear projection matrices conditioned on $\boldsymbol{\lambda}$. Finally, we compute the selection probability for each feasible node $p_{\boldsymbol{\theta}}(\pi_t = i \mid \mathcal{G}, \pi_{1:t-1})$ based on the improved compatibility with adaptation bias (Zhou et al., 2024a):

$$p_{\boldsymbol{\theta}}(\pi_t = i \mid \mathcal{G}, \pi_{1:t-1}) = \mathrm{softmax}(\zeta \cdot \tanh(\frac{\hat{\mathbf{h}}_{(C)}^t (\mathbf{h}_i^{(L)})^{\mathrm{T}}}{\sqrt{d}} + f(\alpha, N, d_{t,i})) + \mathcal{M}_t) \tag{9}$$

where $\zeta$ is the clipping parameter. Inspired by Lin et al. (2022), we use a simple MLP hypernetwork $\mathrm{WEIGHT}(\boldsymbol{\lambda})$ to generate $\{W_Q(\boldsymbol{\lambda}), \ W_K(\boldsymbol{\lambda}), \ W_V(\boldsymbol{\lambda})\}$ (see Appendix E.2 for implementation details). For $\alpha$ in Equation (8) and Equation (9), we generate them via $\mathrm{BIAS}(\boldsymbol{\lambda})$ in Equation (6).

As shown in Figure 2, using a model with a single static set of parameters to solve over 100 diverse VRP variants is challenging and prone to underfitting. To mitigate this, we introduce a problem-adaptive mechanism consisting of two

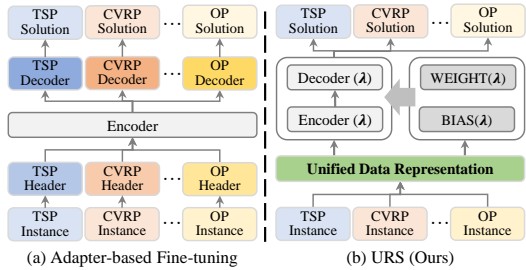

(a) Adapter-based Fine-tuning   (b) URS (Ours)

Figure 2: Comparison between our URS and existing adapter-based fine-tuning frameworks.

components: (1) $\mathrm{BIAS}(\lambda)$ dynamically adjusts how the model integrates different geometric priors in the attention mechanism based on the problem representation $\lambda$; and (2) $\mathrm{WEIGHT}(\lambda)$ generates adaptive decoder parameters for each problem. Unlike existing adapter-based fine-tuning methods (Lin et al., 2024; Drakulic et al., 2025; Wang et al., 2025a), URS adaptively generates the decoder parameters for each problem, thereby enabling the generation of high-quality solutions without requiring retraining or fine-tuning.

## 4 LLM-DRIVEN CONSTRAINT SATISFACTION MECHANISM

While the proposed UDR enables URS to handle a wide range of routing problems, ensuring solution feasibility across diverse and potentially unseen constraints remains a major challenge. Traditional NCO approaches rely on manually designed masks $\mathcal{M}_t$ for each specific VRP variant Kool et al. (2019); Kwon et al. (2020); Luo et al. (2023), which becomes impractical when solving over 100 distinct variants as targeted by URS. To address this bottleneck, we propose an LLM-driven constraint satisfaction mechanism that automatically produces problem-specific **mask generators**. This mechanism transforms constraint handling from a manual engineering task into an automated program design task, thereby effectively supporting URS's excellent zero-shot generalization capabilities.

As illustrated in Figure 3 (a), our constraint satisfaction mechanism operates through a systematic generate-and-validate pipeline. Notably, the generation of mask generators within this pipeline is performed offline: the process begins by prompting an LLM (we adopt GPT-3.5-turbo for this work, though this model selection is not mandatory—alternative models are equally competent, with only differences in computational efficiency) with both a clear and concise problem-specific natural language constraint description that explicitly articulates the task's core constraints and objectives and a standardized Python code template with well-defined input/output interfaces. The LLM then designs a set of candidate mask generators that implement the constraint logic. Each candidate is validated on a set of problem instances using a problem-specific checker to verify its reliability in producing feasible solutions. Invalid candidates are discarded, and the generation process iterates until a reliable mask generator is obtained.

Specifically, during validation, the candidate mask generator guides the stepwise solution construction. As illustrated in Figure 3 (b), at each step $t$, the mask generator takes the current state of the partial solution $\pi_{1:t-1}$ as input, which contains various information such as visited nodes, remaining load, etc. It then produces a binary mask $\mathcal{M}_t$ that identifies all permissible nodes. This mask is directly integrated into the URS's selection probability computation in Equation 9 to ensure feasibility.

To efficiently steer this generate-and-test process, we formulate it as an optimization problem that maximizes the solution validity rate. The well-developed LLM-based automated algorithm design frameworks, such as EoH (Liu et al., 2024b), can be effectively applied to tackle this optimization problem. The optimization process terminates immediately once a reliable mask generator is found, which significantly reduces computational overhead compared to exhaustive search approaches. Furthermore, generated mask generators can be cached and reused for different instances sharing the same problem variant, thereby amortizing the generation cost. For more details about our LLM-driven constraint satisfaction mechanism, please see Appendix G.

## 5 EXPERIMENTS

In this section, we comprehensively evaluate URS against classical and learning-based solvers across more than one hundred distinct VRP variants. For URS, we focus on: (1) performance on seen problems and (2) zero-shot generalization under unseen ones. To ensure comparability with prior neural routing studies, we adopt the 100-node setting as our primary evaluation scenario for URS. All experiments are conducted on a single NVIDIA GeForce RTX 4090 GPU (24GB memory).

### 5.1 EXPERIMENTAL SETUP

**Problem Setting**   We train URS on 11 mixed VRP variants to ensure that each feature in UDR has been seen at least once, i.e., $\mathcal{P} = \{P_i\}_{i=1}^{11}$ (see Appendix J.1 for more discussion about problem selection). Specifically, training problems and related data generations are as follows: (1) ATSP (Kwon et al., 2021);(2) TSP (Kool et al., 2019); (3) CVRP (Kool et al., 2019); (4) ACVRP (Kwon et al., 2021; Kool et al., 2019); (5) Orienteering Problem (OP) (Kool et al., 2019); (6) PCTSP (Kool et al., 2019); (7) PDTSP (Li et al., 2021); (8) CVRPTW (Zhou et al., 2024b); (9) OCVRP (Zhou et al., 2024b); (10) CVRPB (Zhou et al., 2024b), and (11) OCVRPTW (Zhou et al., 2024b). To evaluate the zero-shot generalization of URS, we construct 96 unseen variants to further highlight the broad applicability of URS. And each test dataset consists of 1000 instances. The oracle solver for each problem can be found in Appendix D.

**Model & Training Setting**   We use an embedding dimension $d$ of 128 and a feed-forward layer dimension of 512. We set the number of attention layers in the encoder to 12. The clipping parameter $\zeta = 50$ in Equation (9) for better training convergence (Jin et al., 2023). For training, URS is trained by the REINFORCE (Williams, 1992) algorithm, and the data is generated on the fly (see Appendix E.4 for more details). We train URS for 500 epochs with 2,000 batches per epoch, and the batch size is 128. At each training step, we randomly select one problem $P_i$ from the problem set $\mathcal{P}$. We employ the AdamW optimizer (Loshchilov & Hutter, 2017) with an initial learning rate of $10^{-4}$, which is decayed by a factor of 0.1 at the 451st epoch. Weight decay is set to $10^{-6}$. More details about the model training are provided in Appendix H.

Table 2: Performance on 1K test instances of 11 seen VRPs.

| Method | ATSP Gap | ATSP Time | TSP Gap | TSP Time | OP Gap | OP Time | PCTSP Gap | PCTSP Time | PDTSP Gap | PDTSP Time | – |
|---|---|---|---|---|---|---|---|---|---|---|---|
| Oracle Solver | 0.00% | 2m | 0.00% | 6m | 0.00% | 1.5m | 0.00% | 1.2h | 0.00% | 9.8m | – |
| Sym-NCO | – | | 0.14% | 5s | 0.68% | 15s | 0.14% | 16s | – | | – |
| BQ | 1.27% | 1m | 0.35% | 7s | 0.63% | 7s | – | | – | | – |
| LEHD | – | | 0.59% | 2s | – | | – | | – | | – |
| POMO | – | | 0.13% | 5s | – | | – | | – | | – |
| ICAM | 3.05% | 1m | 0.15% | 4s | – | | – | | – | | – |
| MatNet | 0.95% | 6.5m | – | | – | | – | | – | | – |
| Heter-AM | – | | – | | – | | – | | 6.61% | 5m | – |
| GOAL-MTL | 1.77% | 1m | – | | 1.20% | 38s | – | | – | | – |
| URS-STL | **0.62%** | 1.1m | **0.08%** | 6s | **0.11%** | 4s | **-0.09%** | 5s | **3.22%** | 4s | – |
| URS | 2.26% | 1.1m | 0.57% | 6s | 0.45% | 4s | 1.06% | 5s | 4.98% | 4s | – |

| Method | ACVRP Gap | ACVRP Time | CVRP Gap | CVRP Time | CVRPTW Gap | CVRPTW Time | CVRPB Gap | CVRPB Time | OCVRP Gap | OCVRP Time | OCVRPTW Gap | OCVRPTW Time |
|---|---|---|---|---|---|---|---|---|---|---|---|---|
| Oracle Solver | 0.00% | 6.6m | 0.00% | 9.1m | 0.00% | 19.6m | 0.00% | 20.8m | 0.00% | 5.3m | 0.00% | 20.8m |
| Sym-NCO | – | | 1.46% | 6s | – | | – | | – | | – | |
| BQ | – | | 3.24% | 8s | – | | – | | – | | – | |
| LEHD | – | | 3.68% | 2s | – | | – | | – | | – | |
| POMO | – | | **1.25%** | 6s | – | | – | | – | | – | |
| ICAM | – | | 2.04% | 4s | – | | – | | – | | – | |
| MVMoE | – | | 1.65% | 12s | 4.90% | 15s | 1.28% | 11s | 3.14% | 13s | 3.85% | 15s |
| MTPOMO | – | | 1.85% | 6s | 5.31% | 8s | 1.67% | 5s | 3.46% | 6s | 4.41% | 7s |
| ReLD-MTL | – | | 1.42% | 8s | 4.56% | 10s | **0.90%** | 6s | 2.32% | 7s | **3.10%** | 9s |
| GOAL-MTL | – | | 4.22% | 48s | 4.66% | 42s | – | | – | | – | |
| URS-STL | **2.01%** | 1.4m | 1.63% | 6s | **4.50%** | 8s | 1.59% | 6s | **2.25%** | 6s | 3.22% | 8s |
| URS | 3.06% | 1.4m | 1.81% | 6s | 6.13% | 8s | 1.46% | 6s | 3.24% | 6s | 5.07% | 8s |

**Baseline** We compare URS with the following methods: (1)**Classical Solvers:**[1] PyVRP (Wouda et al., 2024), LKH3 (Helsgaun, 2017), and OR-Tools (Perron & Furnon, 2023); (2)**Neural Solvers:** (i) *specialized solvers that support multiple VRP variants*: Sym-NCO (Kim et al., 2022), BQ (Drakulic et al., 2023), LEHD (Luo et al., 2023), ICAM (Zhou et al., 2024a) and POMO (Kwon et al., 2020); (ii) *general neural solvers capable of handling multiple variants*[2]: MTPOMO (Liu et al., 2024a), MVMoE (Zhou et al., 2024b), ReLD-MTL (Huang et al., 2025), GOAL-MTL (Drakulic et al., 2025); (iii) *Solvers for less-explored variants*: MatNet (Kwon et al., 2021), Heter-AM (Li et al., 2021).

**Metrics and Inference** We report the average objective value (Obj.), optimality gap (Gap), and total inference time (Time) for each method. The optimality gap quantifies the discrepancy between the solutions generated by the corresponding methods and the near-optimal solutions, which are obtained using classical solvers. For most NCO baseline methods, we execute the source code provided by the authors using default settings. We compare the performance of URS to the relevant baselines for each problem as well as its single-task version (i.e., URS-STL). For more details about URS-STL, please see Appendix H.2. For symmetric instances, we report the best result with $\times 8$ instance augmentation, following Kwon et al. (2020). For asymmetric instances, we report the best result with $\times 128$ instance augmentation, following Kwon et al. (2021). Furthermore, we also verify the performance of URS under different evaluation scenarios (e.g., Sym-NCO (Kim et al., 2022)), please see Appendix J.2 for more results.

## 5.2 PERFORMANCE EVALUATION

**Performance on Seen VRPs** The experimental results on trained VRPs with scale 100 are reported in Table 2. We observe that among all comparable neural solvers, URS-STL achieves the best performance on 8 out of the 11 routing problems, complemented by remarkably fast inference,

---

[1]We run LKH3 (Helsgaun, 2017) with 10000 trials and 1 run (Kool et al., 2019). For PyVRP and OR-Tools, we run them on a single CPU core with time limits of 20s (Li et al., 2025a).

[2]Given the differences in problem variants and visible task training among the baseline models, we only compare methods with the same training setup. Thus, RF (Berto et al., 2024) and CaDA (Li et al., 2025a) are not included in our experiments to ensure a fair comparison.

Table 3: Zero-shot generalization on 1K test instances of 30 unseen VRPs. Due to page limit, complete experimental results (including all 96 unseen variants) are deferred to Appendix I.

| Method | CVRPBP | | OCVRPBP | | CVRPBPL | | OCVRPBPTW | | CVRPBPLTW | |
|---|---|---|---|---|---|---|---|---|---|---|
| | Gap | Time | Gap | Time | Gap | Time | Gap | Time | Gap | Time |
| Oracle Solver | 0.00% | 6.6m | 0.00% | 6.6m | 0.00% | 6.6m | 0.00% | 6.6m | 0.00% | 6.6m |
| MVMoE | 13.95% | 14s | 17.36% | 14s | 13.38% | 14s | 2.29% | 15s | 10.46% | 16s |
| MTPOMO | 13.71% | 7s | 16.94% | 8s | 13.44% | 8s | 3.26% | 9s | 11.35% | 10s |
| ReLD-MTL | 13.57% | 9s | 14.83% | 9s | 12.96% | 10s | 2.27% | 11s | 10.80% | 12s |
| URS | **12.95%** | 7s | **14.30%** | 7s | **12.65%** | 8s | **-3.47%** | 8s | **9.74%** | 10s |

| Method | MDCVRP | | MDCVRPL | | MDOCVRPB | | MDOCVRPBPTW | | MDOCVRPBPLTW | |
|---|---|---|---|---|---|---|---|---|---|---|
| | Gap | Time | Gap | Time | Gap | Time | Gap | Time | Gap | Time |
| Oracle Solver | 0.00% | 6.6m | 0.00% | 6.6m | 0.00% | 6.6m | 0.00% | 6.6m | 0.00% | 6.6m |
| MVMoE | 31.90% | 31s | 33.42% | 44s | 28.21% | 26s | 46.18% | 40s | 45.23% | 40s |
| MTPOMO | 23.75% | 27s | 24.32% | 25s | 25.06% | 20s | 34.19% | 29s | 33.37% | 31s |
| ReLD-MTL | 17.37% | 36s | 18.48% | 28s | 18.50% | 21s | 34.08% | 33s | 33.83% | 36s |
| URS | **9.65%** | 22s | **15.32%** | 26s | **15.17%** | 18s | **11.17%** | 25s | **10.96%** | 28s |

| Method | ACVRPB | | ACVRPL | | ACVRPTW | | AOPDCVRP | | ACVRPBL | |
|---|---|---|---|---|---|---|---|---|---|---|
| | Gap | Time | Gap | Time | Gap | Time | Gap | Time | Gap | Time |
| Oracle Solver | 0.00% | 6.6m | 0.00% | 6.6m | 0.00% | 6.6m | 0.00% | 6.6m | 0.00% | 6.6m |
| URS | **7.10%** | 2.1m | **-2.24%** | 1.9m | **11.28%** | 2.5m | **11.13%** | 1.2m | **6.18%** | 2.4m |

| Method | AOCVRPBPTW | | ACVRPBPLTW | | ACVRPBPTW | | AOCVRPBPLTW | | ACVRPBTW | |
|---|---|---|---|---|---|---|---|---|---|---|
| | Gap | Time | Gap | Time | Gap | Time | Gap | Time | Gap | Time |
| Oracle Solver | 0.00% | 6.6m | 0.00% | 6.6m | 0.00% | 6.6m | 0.00% | 6.6m | 0.00% | 6.6m |
| URS | **3.28%** | 2.5m | **-0.58%** | 2.8m | **0.12%** | 2.5m | **2.54%** | 2.7m | **9.95%** | 2.2m |

| Method | AMDCVRPL | | AMDCVRPBPLTW | | AMDOCVRPBPLTW | | PDCVRP | | OPDCVRP | |
|---|---|---|---|---|---|---|---|---|---|---|
| | Gap | Time | Gap | Time | Gap | Time | Gap | Time | Gap | Time |
| Oracle Solver | 0.00% | 6.6m | 0.00% | 6.6m | 0.00% | 6.6m | 0.00% | 6.6m | 0.00% | 6.6m |
| URS | **7.15%** | 6.1m | **4.45%** | 8.3m | **8.63%** | 7.5m | **-1.47%** | 4.3s | **4.93%** | 4.3s |

| Method | AMDOCVRPBPTW | | AMDCVRPBPTW | | AMDCVRP | | APDTSP | | APDCVRP | |
|---|---|---|---|---|---|---|---|---|---|---|
| | Gap | Time | Gap | Time | Gap | Time | Gap | Time | Gap | Time |
| Oracle Solver | 0.00% | 6.6m | 0.00% | 6.6m | 0.00% | 6.6m | 0.00% | 6.6m | 0.00% | 6.6m |
| URS | **9.61%** | 6.9m | **4.63%** | 7.6m | **3.56%** | 5.3m | **6.21%** | 1m | **7.03%** | 1.2m |

highlighting the strength of the URS architecture when fully specialized. Meanwhile, mixing all problems into the training of URS incurs only a minor performance drop, yet still delivers competitive performance across all problems, demonstrating that potential connections among different problems are effectively captured by URS.

**Generalization to Unseen VRPs** We evaluate our URS model on zero-shot generalization across 96 unseen VRP variants, including 44 well-studied VRP variants from the NCO community and 52 newly introduced variants that have not been explored before. The detailed results are presented in Table 3 and Appendix I. On widely-studied constraints (i.e., C, O, L, TW, MD, B, BP), URS continues to deliver high-quality solutions. Notably, URS merely selects the next node from a given feasible candidates without any additional domain knowledge, making its performance especially meaningful. Benefiting from our UDR, URS can directly address previously unexplored variants while maintaining solution quality, which compounds asymmetry and PD relations. Without any fine-tuning, URS still constructs high-quality solutions for more than 90 VRP variants that are not seen in training, highlighting its strong zero-shot generalization ability.

**Results on Benchmark Dataset** We further evaluate the generalization ability of URS on benchmark instances from CVRPLIB Set-X (Uchoa et al., 2017) with scale $\in [500, 1000]$ and large-scale CVRPLib XXL (Arnold et al., 2019) benchmark dataset with scale $\in [3000, 7000]$, and we compare URS against several representative single-task and multi-task models. As demonstrated in Appendix J.3, URS maintains its position as the best overall performance across instances of varying scales in Set-X dataset. For CVRPLib XXL benchmark. The results clearly demonstrate that URS can significantly outperform existing multi-task neural routing solvers, especially on large-scale instances. Notably, URS also greatly surpasses many representative specialized models (e.g., LEHD, BQ, and ELG). These results further underscore the practical applicability of URS in real-world scenarios.

**Scalability to Larger-scale Realistic Problems**   Due to the limited benchmark datasets currently available for VRP variants, we have conducted further experiments on large-scale synthetic problem instances to validate the large-scale cross-problem generalization ability of URS. We generate a new large-scale (1K-5K nodes) synthetic dataset for a diverse set of problems (CVRPTW, CVRPB, and OCVRPTW). According to the results in Appendix J.4, our proposed URS consistently achieves the best solution quality, complemented by remarkably fast inference times, across various problem instances among all comparable neural solvers. The advantages of URS over existing multi-task neural solvers are significant. Remarkably, URS already outperforms the strong HGS-PyVRP with a carefully manual design on large-scale CVRPB instances. These extensive results demonstrate URS's strong scalability and generalization ability to large-scale and realistic problems.

# 6   ABLATION STUDY

In this section, we conduct an ablation study across 11 seen VRP variants to validate the effect of key components of URS, mainly including: (1) without node-type indicator $\boldsymbol{\xi}$ of UDR; (2) without problem state $C_t$; (3) effects of different priors in MBM ; (4) effects of alternative decoder architectures. As detailed in Table 4. URS maintains its competitive performance without $C_t$. And its effectiveness is further improved by the inclusion of a node-type indicator $\boldsymbol{\xi}$. For MBM, the results reveal that three priors $\{\boldsymbol{D}, \boldsymbol{D}^{\mathbf{T}}, \boldsymbol{R}\}$ are complementary, and their combined use allows MBM to effectively learn the geometric and relational biases inherent in various problems, resulting in superior overall performance. Furthermore, replacing the decoder with POMO or ReLD confirms the strong cross-problem robustness of URS, while the integration of the decoder in ICAM (Zhou et al., 2024a) leads to a further improvement. More detailed results and analyses can be found in Appendix K.

Table 4: Effects of different components of URS.

| Method | Avg.gap |
|---|---|
| w/o $\boldsymbol{\xi}$ | 3.78% |
| w/o $C_t$ | 2.97% |
| $\{\boldsymbol{D}\}$ | 6.33% |
| $\{\boldsymbol{D}, \boldsymbol{D}^{\mathbf{T}}\}$ | 4.07% |
| $\{\boldsymbol{D}, \boldsymbol{R}\}$ | 7.34% |
| POMO-Dec. | 3.39% |
| ReLD-Dec. | 3.87% |
| URS | **2.74%** |

To empirically evaluate the necessity of our proposed components (MBM, WEIGHT($\lambda$), and BIAS($\lambda$)), we have conducted tests on the 11 seen problems as well as 11 additional unseen CVRP variants widely investigated in current multi-task research (e.g., MVMoE (Zhou et al., 2024b)). As shown in Appendix K.4, the full URS model achieves the lowest average gap and provides the best solution on the majority of tasks (19/22). This demonstrates that the synergy of these components is essential for robust cross-problem generalization.

# 7   CONCLUSION, LIMITATION, AND FUTURE WORK

In this work, we propose a novel RL-based Unified Routing Solver (URS) for cross-problem zero-shot generalization. The core of URS is a Unified Data Representation (UDR), which broadens problem coverage by substituting problem enumeration with data unification. Its generalization capability is further enhanced by a Mixed Bias Module (MBM) and an adaptive parameter generator. Moreover, an LLM-based constraint satisfaction mechanism guarantees solution feasibility and reduces dependency on domain expertise. Extensive experiments demonstrate that URS can produce high-quality solutions for more than 100 distinct VRP variants composed of 10 different constraints without any fine-tuning, more than 90 of which are unseen during training.

**Limitation and Future Work**   While URS achieves impressive results on a wide range of VRP variants, its current training scheme does not accommodate their inherent differences in complexity and randomly selects one problem at each training step. In addition, the UDR can produce highly similar representations for problems that share similar attributes but involve significantly different dynamic constraints (e.g., CVRP vs. CVRPL). This ambiguity may challenge the parameter generator and impair zero-shot performance on such variants. In the future, we aim to explore more effective training schemes to further improve training efficiency and design more robust unified data representation.

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

# A   RELATED WORK

## A.1   NEURAL ROUTING SOLVERS WITH SINGLE-TASK LEARNING

The NCO methods have emerged as a promising paradigm for tackling various routing problems, achieving impressive empirical performance. Early seminal work based on Pointer Networks ignited interest in learning end-to-end construction policies for TSP and CVRP (Vinyals et al., 2015; Bello et al., 2016; Nazari et al., 2018). Subsequently, the Transformer-style heavy-encoder and light-decoder auto-regressive architecture has become the dominant paradigm (Kool et al., 2019), while POMO (Kwon et al., 2020) exploits permutation symmetry to enhance exploration diversity. Many advancements within this paradigm have pushed promising results on small instances (e.g., 100-node TSPs) (Xin et al., 2020; Kim et al., 2022; Xiao et al., 2024a;b; Sun et al., 2024). However, models trained on a fixed instance scale often degrade sharply when evaluated out of scale. To mitigate the poor generalization performance, recent studies introduce auxiliary local policies (Gao et al., 2024), adaptive perception across instance scales (Zhou et al., 2024a), or different search space reduction methods (Fang et al., 2024; Zhou et al., 2025; Chen et al., 2025). In parallel, "heavy decoder" designs that dynamically recompute embeddings at each construction step (Drakulic et al., 2023; Luo et al., 2023) demonstrate strong generalization on large-scale instances. To improve the performance on large-scale routing instances, complementary directions are reducing solving difficulty via problem decomposition (Zheng et al., 2024b; Ye et al., 2024b; Li et al., 2025b) or adopting non-autoregressive generation augmented with additional searches (e.g., Monte Carlo tree search (MCTS) and 2-opt) (Fu et al., 2021; Sun & Yang, 2023; Qiu et al., 2022; Li et al., 2023; Ye et al., 2023). Moreover, heterogeneous vehicle routing (Li et al., 2022b), asymmetric distance settings (Kwon et al., 2021), multiple VRPs (Zheng et al., 2024a), and pickup–delivery problems (Li et al., 2021; Kong et al., 2024) have each motivated tailored neural designs, which have yielded competitive results. Despite these advancements, they still train a specialized model per problem type, limiting cross-problem transfer. We instead focus on a unified neural routing solver that aims to generalize across multiple routing problems without retraining from scratch for each specification.

## A.2   NEURAL ROUTING SOLVERS WITH MULTI-TASK LEARNING

To address the challenge of cross-problem generalization, growing attention has shifted to multi-task learning capable of adapting to diverse routing problems. Existing efforts can broadly fall into two categories: (1) constraint combination-based multi-task learning and (2) adapter-based fine-tuning frameworks. The former line (e.g., MTPOMO (Liu et al., 2024a)) treats VRP variants as combinations of a predefined set of attributes and trains a shared backbone, achieving promising results on up to 16 VRP variants. Follow-up work extends this paradigm by introducing mixture-of-experts (MoE) modules (Zhou et al., 2024b), heavy decoder designs plus knowledge distillation (Zheng et al., 2025a), and constraint-aware dual-attention (Li et al., 2025a) or updated Transformer structure (Berto et al., 2024) to improve model performance. While this enables knowledge sharing across problems, its coverage is inherently bounded by the hand-specified attribute set. The second family trains a shared backbone and performs adaptation to each variant via lightweight adapters. For instance, Lin et al. (2024) pretrain a TSP backbone and adapt to new variants through adapter-based fine-tuning. Drakulic et al. (2025) integrate edge features and adopt supervised multi-task pretraining for obtaining a powerful backbone before fine-tuning on novel tasks. Wang et al. (2025a) propose a multi-armed bandit strategy to realize more efficient multi-task training. Although these reduce retraining cost, they fail to achieve zero-shot generalization. In contrast, we focus on a constructive neural routing solver supported with zero-shot capability: once trained, it can directly generate high-quality solutions for previously unseen problems without any fine-tuning.

## A.3   LLM-BASED ALGORITHM GENERATION IN VRPs

Employing LLMs to solve VRPs is a new research direction. As an early attempt, Yang et al. (2023) utilizes the LLM as a direct optimizer, prompting an LLM to directly generate solutions for TSP instances. A more predominant stream leverages LLMs as automated algorithm designers to generate algorithms for solving VRPs. These methods often integrate LLMs with a search procedure. EoH (Liu et al., 2024b) combined an LLM with Evolutionary Computation to design heuristic components for a Guided Local Search (GLS) framework. ReEvo (Ye et al., 2024a) incorporated a reflection mechanism to refine heuristics for iterative solution construction. HSEvo (Dat et al.,

2025) additionally introduced a diversity mechanism for designing heuristics solving TSP and OP. And AHD-MCTS (Zheng et al., 2025b) employed Monte Carlo Tree Search (MCTS) to guide the design of heuristics for solving TSP and CVRP. Despite their promise, these methods share common limitations rooted in their reliance on the LLM's implicit, pre-trained knowledge as the core design logic. The process of discovering effective algorithms demands extensive search procedures and is computationally expensive. Furthermore, the heuristics generated by the LLM are susceptible to overfitting to the specific instances used during the design phase, limiting their generalization capabilities. In contrast, our URS overcomes these challenges by establishing a clear division of labor: the LLM is tasked with satisfying complex constraints, while our URS model is responsible for ensuring solution quality. This approach significantly reduces the reasoning burden on the LLM, mitigating the risk of overfitting and enabling URS to effectively solve a more extensive range of VRPs.

## B  SETUPS OF VRP VARIANTS

In this paper, we consider over 100 VRP variants which may simultaneously have one or more constraints from the following categories: (1) Capacity (C); (2) Open Route (O); (3) Backhaul (B); (4) Backhaul and Priority (BP); (5) Duration Limit (L); (6) Time Windows (TW); (7) Multi-Depot (MD); (8) Prize Collecting (PC); (9) Asymmetric (A); (10) Pickup and Delivery (PD). The node coordinates are randomly sampled from the unit square when available (in symmetric geometric cases, i.e., $d_{ij} = d_{ji}$), following AM (Kool et al., 2019). Below, we provide a comprehensive description of the data generation process for each constraint.

**Capacity (C)**   The selection of the vehicle capacity $\mathcal{C}$ is critical, as prior work (Luo et al., 2025c) demonstrates its significant impact on the solution structure. For PDVRP variants (e.g., PDCVRP, OPDCVRP, APDCVRP, and AOPDCVRP), an excessively large $\mathcal{C}$ causes the solution to approximate that of a PDTSP instance. Conversely, an overly restrictive $\mathcal{C}$ promotes a simplistic, repetitive Pickup-Delivery sequence. To ensure structurally sound solutions, we set $\mathcal{C}$=20 for PDCVRP variants. For all other VRP variants, we adopt the convention from AM (Kool et al., 2019), setting $\mathcal{C}$=50.

**Open Route (O)**   The open route setting implies that vehicles are not required to return to the depot after completing their service tours. This does not alter the data generation, and it specifically affects the final cost evaluation by excluding the travel distance from the last customer back to the depot.

**Backhaul (B)**   The standard CVRP involves vehicles delivering goods from a depot to a set of customers with positive demands. The introduction of backhauls extends this formulation to the CVRP with Backhauls (CVRPB), where vehicles must also load goods from backhaul customers and return them to the depot, in addition to serving the standard linehaul (delivery) customers. Following the experimental setup of MTPOMO (Liu et al., 2024a), we generate customer demands by sampling from a discrete uniform distribution over the integers $\{1, \ldots, 9\}$. Subsequently, 20% of these nodes are randomly designated as backhaul customers, and their corresponding demand values are negative, while the remainder are classified as linehaul customers. Notably, no precedence constraints are imposed between the servicing of linehaul and backhaul customers.To ensure the feasibility of the solutions constructed by our neural solver, we adhere to the MVMoE (Zhou et al., 2024b) framework's policy that the first customer visited must be a linehaul node.

**Backhaul and Priority (BP)**   This constraint imposes a strict precedence requirement on the standard CVRPB, dictating that all linehaul customers must be serviced prior to any backhaul customers. Consequently, once a vehicle serves a backhaul customer, it is prohibited from visiting any subsequent linehaul nodes within the same route. If the first customer served is a backhaul node, the vehicle's load is initialized to zero. This ensures the neural solver can construct a feasible solution.

**Duration Limit (L)**   Following MTPOMO (Liu et al., 2024a), we impose a duration limit (i.e., the maximum length of each vehicle route) of 3 in symmetric cases. Since the node coordinates are randomly sampled from the unit square, the maximum possible distance between any two nodes is $\sqrt{2}$. The setting is sufficient to ensure a vehicle can deliver goods to at least one customer and return to the depot, thus guaranteeing the feasibility of the solution. For the setting of asymmetric cases, we provide details in the description of the following asymmetric instances.

**Time Windows (TW)**   Consistent with the MVMoE (Zhou et al., 2024b), we set time windows of the depot to $[e_0, l_0]$=[0,3] and its service time to 0. The service time $s_i$ for each customer node is set to 0.2. For the time windows for each customer node, we generate them following the methodology outlined in MVMoE (Zhou et al., 2024b).

**Multi-Depot (MD)**   The generation of coordinates for the multiple depots aligns with that of other nodes, with the number of depots set to 3 (Berto et al., 2024). A vehicle is required to return to the same depot it departed from. To guarantee that each starting node is considered in this multi-depot variant, we set the number of trajectories for an instance to be the product of the number of depots and the number of customer nodes, maximizing the exploration of high-quality solutions.

**Prize Collecting (PC)**    Under the Prize Collecting condition, each node (excluding the depot) is assigned both a prize and a penalty. A prize is collected for each visited node, while a penalty is incurred for each unvisited node. The objective is to minimize the total travel distance plus the sum of penalties for all unvisited nodes. A key constraint is that a minimum amount of total prize must be collected before the vehicle is permitted to return to the depot and terminate its route. Consistent with the AM (Kool et al., 2019) framework, the prizes and penalties are generated as follows: $prize \sim \text{Uniform}\left(0, \frac{4}{n}\right)$, and $penalty \sim \text{Uniform}\left(0, 3 \cdot \frac{k_n}{n}\right)$, where $n = 100$ and $k_n = 4$ in our experiments.

**Asymmetric (A)**    We adopt the data generation method from MatNet (Kwon et al., 2021) to create an asymmetric distance matrix. Due to the small pairwise distances in the generated distance matrix, the parameters for duration limit and time windows set for symmetric cases are not applicable to the asymmetric problems. To ensure the constraints remain valid, we adjust the duration limit and depot end time to 0.6 and 1.0, respectively.

**Pickup and Delivery (PD)**    For PDP variants, we adhere to the methodology in Li et al. (2021). For a problem of size $N$, node 0 is set to the depot, nodes 1 to $N/2$ are designated as pickup nodes, while nodes from $N/2 + 1$ to $N$ are designated as delivery nodes. For each pickup node $v_i$, its corresponding delivery node is $v_{i+N/2}$. A strict precedence constraint is imposed in PDP cases, requiring each vehicle to visit a pickup node before its corresponding delivery node. For PDCVRP variants, the demand for each delivery node is randomly sampled from a discrete uniform distribution over $\{1, 2, \ldots, 9\}$. The demand for each pickup node is then set to the negative of its corresponding delivery node's demand. To ensure the feasibility of the constraints, the vehicle capacity is set to 20.

## C UNIFIED DATA REPRESENTATION

In this section, we provide a detailed description of our UDR $\boldsymbol{U} = \{\mathbf{u}_i\}_{i=0}^n$ (see Table 5). Each node $\mathbf{u}_i$ consists of three components, i.e., $\mathbf{u}_i = \{\boldsymbol{\rho}_i, \boldsymbol{\omega}_i, \boldsymbol{\xi}_i\}$. where $\boldsymbol{\rho}_i, \boldsymbol{\omega}_i, \boldsymbol{\xi}_i$ are a position identifier, unified attribute set, and node-type indicator, respectively. To distinguish linehaul/delivery from backhaul/pickup nodes, we assign negative demand values to backhaul/pickup nodes (positive for linehaul/delivery), making demand the only signed component, following (Liu et al., 2024a; Zhou et al., 2024b).

Table 5: Detailed description of each component in UDR.

| Attribute Name | Symbol | Range | Description | Examples |
|---|---|---|---|---|
| Random Identifier | $\eta$ | $[0,1]$ | Random scalar for asymmetric problems. | ATSP |
| Node Coordinates | $\{x, y\}$ | $[0,1]^2$ | 2D coordinates of the node. | TSP, CVRP |
| Demand | $\delta$ | $[-1,1]$ | Node demand ($+$ for delivery, $-$ for pickup). | CVRP, PDCVRP |
| Prize | $\epsilon$ | $[0,1]$ | Prize collected for each node. | OP, PCTSP |
| Penalty | $\mu$ | $[0,1]$ | Penalty incurred for each node. | PCTSP |
| Earliest Arrival Time | $e$ | $[0,1]$ | Earliest permissible arrival time. | CVRPTW |
| Latest Arrival Time | $l$ | $[0,1]$ | Latest permissible arrival time (deadline). | CVRPTW |
| Service Time | $s$ | $[0,1]$ | Time required for service at the node. | CVRPTW |
| Depot | $-$ | $\{0,1\}$ | Indicates if the node is a depot. | CVRP |
| Pickup Node | $-$ | $\{0,1\}$ | Indicates if the node is a pickup location. | PDTSP |
| Delivery Node | $-$ | $\{0,1\}$ | Indicates if the node is a delivery location. | CVRP, PDTSP |
| Node in Sub-routes | $-$ | $\{0,1\}$ | The solution $\pi$ can have sub-routes. | CVRP |
| Node in Open Route | $-$ | $\{0,1\}$ | Vehicles need not return to the depot. | OCVRP |

## D MULTI-HOT PROBLEM REPRESENTATION

Leveraging the UDR, each problem instance activates only a subset of the unified feature space. Thus, for any instance $\mathcal{G}_i$ from different problems, we simply mark active (non-zero) feature slots with 1 to obtain a corresponding multi-hot problem representation $\boldsymbol{\lambda}_i$. Different from raw feature values, it encodes presence only and provides conditional signals to the following position bias weight (see Equation (6)) and adaptive decoder (see Equation (8) and Equation (9)), helping the model distinguish variants and refine node selection. Here, we provide detailed representations for each problem in our experiments, and each problem is a 13-dimensional multi-hot vector.

As shown in Table 6, we train URS on eleven classic and varied routing problems to ensure that each feature in the UDR has been seen at least once, i.e., $\mathcal{P} = \{P_i\}_{i=1}^{11}$. Then, we evaluate generalization on unseen problems on different combinations of ten constraint attributes (see Appendix B), and the detailed representations of each unseen problem can be found in Table 7 (documented VRP variants) and Table 8 (undocumented VRP variants).

Table 6: The representations for seen routing problems in our experiments. The total number of problems is 11. In training, each feature in the UDR has been seen at least once. Here, RI represents the random identifier, and EAT, LAT, and ST indicate the earliest arrival time, the latest arrival time, and the service time, respectively.

| Problem | Oracle | RI | Coord. | Demand | Prize | Penalty | EAT | LAT | ST | Depot | Pickup | Delivery | Sub-routes | Open Route |
|---|---|---|---|---|---|---|---|---|---|---|---|---|---|---|
| ATSP | LKH-3 | ✓ | | | | | | | | | | | | |
| TSP | LKH-3 | | ✓ | | | | | | | | | | | |
| OP | Compass | | ✓ | | ✓ | | | | | ✓ | | | | |
| PCTSP | ILS | | ✓ | | ✓ | ✓ | | | | ✓ | | | | |
| PDTSP | LKH-3 | | ✓ | | | | | | | ✓ | ✓ | ✓ | | |
| ACVRP | OR-Tools | ✓ | | ✓ | | | | | | ✓ | | ✓ | ✓ | |
| CVRP | PyVRP | | ✓ | ✓ | | | | | | ✓ | | ✓ | ✓ | |
| CVRPTW | PyVRP | | ✓ | ✓ | | | ✓ | ✓ | ✓ | ✓ | | ✓ | ✓ | |
| CVRPB | OR-Tools | | ✓ | ✓ | | | | | | ✓ | ✓ | ✓ | ✓ | |
| OCVRP | LKH-3 | | ✓ | ✓ | | | | | | ✓ | | ✓ | ✓ | ✓ |
| OCVRPTW | OR-Tools | | ✓ | ✓ | | | ✓ | ✓ | ✓ | ✓ | | ✓ | ✓ | ✓ |

Table 7: The problem representations of 44 VRP variants that have been extensively explored in current literature (Liu et al., 2024a; Berto et al., 2024). Here, RI represents the random identifier, and EAT, LAT, and ST indicate the earliest arrival time, the latest arrival time, and the service time, respectively.

| Problem | Oracle | RI | Coord. | Demand | Prize | Penalty | EAT | LAT | ST | Depot | Pickup | Delivery | Sub-routes | Open Route |
|---|---|---|---|---|---|---|---|---|---|---|---|---|---|---|
| CVRPL | LKH-3 | | ✓ | ✓ | | | | | | ✓ | | ✓ | ✓ | |
| OCVRPB | OR-Tools | | ✓ | ✓ | | | | | | ✓ | ✓ | ✓ | ✓ | ✓ |
| CVRPBL | OR-Tools | | ✓ | ✓ | | | | | | ✓ | ✓ | ✓ | ✓ | |
| CVRPLTW | OR-Tools | | ✓ | ✓ | | | ✓ | ✓ | ✓ | ✓ | | ✓ | ✓ | |
| OCVRPBTW | OR-Tools | | ✓ | ✓ | | | ✓ | ✓ | ✓ | ✓ | ✓ | ✓ | ✓ | ✓ |
| CVRPBLTW | OR-Tools | | ✓ | ✓ | | | ✓ | ✓ | ✓ | ✓ | ✓ | ✓ | ✓ | |
| OCVRPL | OR-Tools | | ✓ | ✓ | | | | | | ✓ | | ✓ | ✓ | ✓ |
| CVRPBTW | OR-Tools | | ✓ | ✓ | | | ✓ | ✓ | ✓ | ✓ | ✓ | ✓ | ✓ | |
| OCVRPBL | OR-Tools | | ✓ | ✓ | | | | | | ✓ | ✓ | ✓ | ✓ | ✓ |
| OCVRPLTW | OR-Tools | | ✓ | ✓ | | | ✓ | ✓ | ✓ | ✓ | | ✓ | ✓ | ✓ |
| OCVRPBLTW | OR-Tools | | ✓ | ✓ | | | ✓ | ✓ | ✓ | ✓ | ✓ | ✓ | ✓ | ✓ |
| CVRPBP | OR-Tools | | ✓ | ✓ | | | | | | ✓ | ✓ | ✓ | ✓ | |
| OCVRPBP | OR-Tools | | ✓ | ✓ | | | | | | ✓ | ✓ | ✓ | ✓ | ✓ |
| CVRPBPL | OR-Tools | | ✓ | ✓ | | | | | | ✓ | ✓ | ✓ | ✓ | |
| OCVRPBPTW | OR-Tools | | ✓ | ✓ | | | ✓ | ✓ | ✓ | ✓ | ✓ | ✓ | ✓ | ✓ |
| CVRPBLTW | OR-Tools | | ✓ | ✓ | | | ✓ | ✓ | ✓ | ✓ | ✓ | ✓ | ✓ | |
| CVRPBPTW | OR-Tools | | ✓ | ✓ | | | ✓ | ✓ | ✓ | ✓ | ✓ | ✓ | ✓ | |
| OCVRPBPL | OR-Tools | | ✓ | ✓ | | | | | | ✓ | ✓ | ✓ | ✓ | ✓ |
| OCVRPBLTW | OR-Tools | | ✓ | ✓ | | | ✓ | ✓ | ✓ | ✓ | ✓ | ✓ | ✓ | ✓ |
| MDCVRP | PyVRP | | ✓ | ✓ | | | | | | ✓ | | ✓ | ✓ | |
| MDCVRPTW | PyVRP | | ✓ | ✓ | | | ✓ | ✓ | ✓ | ✓ | | ✓ | ✓ | |
| MDOCVRP | PyVRP | | ✓ | ✓ | | | | | | ✓ | | ✓ | ✓ | ✓ |
| MDCVRPL | PyVRP | | ✓ | ✓ | | | | | | ✓ | | ✓ | ✓ | |
| MDCVRPB | PyVRP | | ✓ | ✓ | | | | | | ✓ | ✓ | ✓ | ✓ | |
| MDOCVRPTW | PyVRP | | ✓ | ✓ | | | ✓ | ✓ | ✓ | ✓ | | ✓ | ✓ | ✓ |
| MDOCVRPB | PyVRP | | ✓ | ✓ | | | | | | ✓ | ✓ | ✓ | ✓ | ✓ |
| MDCVRPBL | PyVRP | | ✓ | ✓ | | | | | | ✓ | ✓ | ✓ | ✓ | |
| MDCVRPLTW | PyVRP | | ✓ | ✓ | | | ✓ | ✓ | ✓ | ✓ | | ✓ | ✓ | |
| MDOCVRPBTW | PyVRP | | ✓ | ✓ | | | ✓ | ✓ | ✓ | ✓ | ✓ | ✓ | ✓ | ✓ |
| MDCVRPBLTW | PyVRP | | ✓ | ✓ | | | ✓ | ✓ | ✓ | ✓ | ✓ | ✓ | ✓ | |
| MDOCVRPL | PyVRP | | ✓ | ✓ | | | | | | ✓ | | ✓ | ✓ | ✓ |
| MDCVRPBTW | PyVRP | | ✓ | ✓ | | | ✓ | ✓ | ✓ | ✓ | ✓ | ✓ | ✓ | |
| MDOCVRPBL | PyVRP | | ✓ | ✓ | | | | | | ✓ | ✓ | ✓ | ✓ | ✓ |
| MDOCVRPLTW | PyVRP | | ✓ | ✓ | | | ✓ | ✓ | ✓ | ✓ | | ✓ | ✓ | ✓ |
| MDOCVRPBLTW | PyVRP | | ✓ | ✓ | | | ✓ | ✓ | ✓ | ✓ | ✓ | ✓ | ✓ | ✓ |
| MDCVRPBP | PyVRP | | ✓ | ✓ | | | | | | ✓ | ✓ | ✓ | ✓ | |
| MDOCVRPBP | PyVRP | | ✓ | ✓ | | | | | | ✓ | ✓ | ✓ | ✓ | ✓ |
| MDCVRPBPL | PyVRP | | ✓ | ✓ | | | | | | ✓ | ✓ | ✓ | ✓ | |
| MDOCVRPBPTW | PyVRP | | ✓ | ✓ | | | ✓ | ✓ | ✓ | ✓ | ✓ | ✓ | ✓ | ✓ |
| MDCVRPBLTW | PyVRP | | ✓ | ✓ | | | ✓ | ✓ | ✓ | ✓ | ✓ | ✓ | ✓ | |
| MDCVRPBTW | PyVRP | | ✓ | ✓ | | | ✓ | ✓ | ✓ | ✓ | ✓ | ✓ | ✓ | |
| MDOCVRPBPL | PyVRP | | ✓ | ✓ | | | | | | ✓ | ✓ | ✓ | ✓ | ✓ |
| MDOCVRPBLTW | PyVRP | | ✓ | ✓ | | | ✓ | ✓ | ✓ | ✓ | ✓ | ✓ | ✓ | ✓ |
| SPCTSP | ILS | | ✓ | | ✓ | ✓ | | | | ✓ | | | | |

Table 8: The problem representations of 52 VRP variants that have not yet been explored in current literature. Here, RI represents the random identifier, and EAT, LAT, and ST indicate the earliest arrival time, the latest arrival time, and the service time, respectively.

| Problem | Oracle | RI | Coord. | Demand | Prize | Penalty | EAT | LAT | ST | Depot | Pickup | Delivery | Sub-routes | Open Route |
|---|---|---|---|---|---|---|---|---|---|---|---|---|---|---|
| ACVRPTW | OR-Tools | ✓ | | ✓ | | | ✓ | ✓ | ✓ | ✓ | | ✓ | ✓ | |
| AOCVRP | OR-Tools | ✓ | | ✓ | | | | | | ✓ | | ✓ | ✓ | ✓ |
| ACVRPL | OR-Tools | ✓ | | ✓ | | | | | | ✓ | | ✓ | ✓ | |
| ACVRPB | OR-Tools | ✓ | | ✓ | | | | | | ✓ | ✓ | ✓ | ✓ | |
| AOCVRPTW | OR-Tools | ✓ | | ✓ | | | ✓ | ✓ | ✓ | ✓ | | ✓ | ✓ | ✓ |
| AOCVRPB | OR-Tools | ✓ | | ✓ | | | | | | ✓ | ✓ | ✓ | ✓ | ✓ |
| ACVRPBL | OR-Tools | ✓ | | ✓ | | | | | | ✓ | ✓ | ✓ | ✓ | |
| ACVRPLTW | OR-Tools | ✓ | | ✓ | | | ✓ | ✓ | ✓ | ✓ | | ✓ | ✓ | |
| AOCVRPBTW | OR-Tools | ✓ | | ✓ | | | ✓ | ✓ | ✓ | ✓ | ✓ | ✓ | ✓ | ✓ |
| ACVRPBLTW | OR-Tools | ✓ | | ✓ | | | ✓ | ✓ | ✓ | ✓ | ✓ | ✓ | ✓ | |
| AOCVRPL | OR-Tools | ✓ | | ✓ | | | | | | ✓ | | ✓ | ✓ | ✓ |
| ACVRPBTW | OR-Tools | ✓ | | ✓ | | | ✓ | ✓ | ✓ | ✓ | ✓ | ✓ | ✓ | |
| AOCVRPBL | OR-Tools | ✓ | | ✓ | | | | | | ✓ | ✓ | ✓ | ✓ | ✓ |
| AOCVRPLTW | OR-Tools | ✓ | | ✓ | | | ✓ | ✓ | ✓ | ✓ | | ✓ | ✓ | ✓ |
| AOCVRPBLTW | OR-Tools | ✓ | | ✓ | | | ✓ | ✓ | ✓ | ✓ | ✓ | ✓ | ✓ | ✓ |
| ACVRPBP | OR-Tools | ✓ | | ✓ | | | | | | ✓ | ✓ | ✓ | ✓ | |
| AOCVRPBP | OR-Tools | ✓ | | ✓ | | | | | | ✓ | ✓ | ✓ | ✓ | ✓ |
| ACVRPBPL | OR-Tools | ✓ | | ✓ | | | | | | ✓ | ✓ | ✓ | ✓ | |
| AOCVRPBPTW | OR-Tools | ✓ | | ✓ | | | ✓ | ✓ | ✓ | ✓ | ✓ | ✓ | ✓ | ✓ |
| ACVRPBLTW | OR-Tools | ✓ | | ✓ | | | ✓ | ✓ | ✓ | ✓ | ✓ | ✓ | ✓ | |
| ACVRPBTW | OR-Tools | ✓ | | ✓ | | | ✓ | ✓ | ✓ | ✓ | ✓ | ✓ | ✓ | |
| AOCVRPBPL | OR-Tools | ✓ | | ✓ | | | | | | ✓ | ✓ | ✓ | ✓ | ✓ |
| AOCVRPBLTW | OR-Tools | ✓ | | ✓ | | | ✓ | ✓ | ✓ | ✓ | ✓ | ✓ | ✓ | ✓ |
| AMDCVRP | OR-Tools | ✓ | | ✓ | | | | | | ✓ | | ✓ | ✓ | |
| AMDCVRPTW | OR-Tools | ✓ | | ✓ | | | ✓ | ✓ | ✓ | ✓ | | ✓ | ✓ | |
| AMDOCVRP | OR-Tools | ✓ | | ✓ | | | | | | ✓ | | ✓ | ✓ | ✓ |
| AMDCVRPL | OR-Tools | ✓ | | ✓ | | | | | | ✓ | | ✓ | ✓ | |
| AMDCVRPB | OR-Tools | ✓ | | ✓ | | | | | | ✓ | ✓ | ✓ | ✓ | |
| AMDOCVRPTW | OR-Tools | ✓ | | ✓ | | | ✓ | ✓ | ✓ | ✓ | | ✓ | ✓ | ✓ |
| AMDOCVRPB | OR-Tools | ✓ | | ✓ | | | | | | ✓ | ✓ | ✓ | ✓ | ✓ |
| AMDCVRPBL | OR-Tools | ✓ | | ✓ | | | | | | ✓ | ✓ | ✓ | ✓ | |
| AMDCVRPLTW | OR-Tools | ✓ | | ✓ | | | ✓ | ✓ | ✓ | ✓ | | ✓ | ✓ | |
| AMDOCVRPBTW | OR-Tools | ✓ | | ✓ | | | ✓ | ✓ | ✓ | ✓ | ✓ | ✓ | ✓ | ✓ |
| AMDCVRPBLTW | OR-Tools | ✓ | | ✓ | | | ✓ | ✓ | ✓ | ✓ | ✓ | ✓ | ✓ | |
| AMDOCVRPL | OR-Tools | ✓ | | ✓ | | | | | | ✓ | | ✓ | ✓ | ✓ |
| AMDCVRPBTW | OR-Tools | ✓ | | ✓ | | | ✓ | ✓ | ✓ | ✓ | ✓ | ✓ | ✓ | |
| AMDOCVRPBL | OR-Tools | ✓ | | ✓ | | | | | | ✓ | ✓ | ✓ | ✓ | ✓ |
| AMDOCVRPLTW | OR-Tools | ✓ | | ✓ | | | ✓ | ✓ | ✓ | ✓ | | ✓ | ✓ | ✓ |
| AMDOCVRPBLTW | OR-Tools | ✓ | | ✓ | | | ✓ | ✓ | ✓ | ✓ | ✓ | ✓ | ✓ | ✓ |
| AMDCVRPBP | OR-Tools | ✓ | | ✓ | | | | | | ✓ | ✓ | ✓ | ✓ | |
| AMDOCVRPBP | OR-Tools | ✓ | | ✓ | | | | | | ✓ | ✓ | ✓ | ✓ | ✓ |
| AMDCVRPBPL | OR-Tools | ✓ | | ✓ | | | | | | ✓ | ✓ | ✓ | ✓ | |
| AMDOCVRPBPTW | OR-Tools | ✓ | | ✓ | | | ✓ | ✓ | ✓ | ✓ | ✓ | ✓ | ✓ | ✓ |
| AMDCVRPBLTW | OR-Tools | ✓ | | ✓ | | | ✓ | ✓ | ✓ | ✓ | ✓ | ✓ | ✓ | |
| AMDCVRPBTW | OR-Tools | ✓ | | ✓ | | | ✓ | ✓ | ✓ | ✓ | ✓ | ✓ | ✓ | |
| AMDOCVRPBPL | OR-Tools | ✓ | | ✓ | | | | | | ✓ | ✓ | ✓ | ✓ | ✓ |
| AMDOCVRPBLTW | OR-Tools | ✓ | | ✓ | | | ✓ | ✓ | ✓ | ✓ | ✓ | ✓ | ✓ | ✓ |
| APDTSP | OR-Tools | ✓ | | | | | | | | ✓ | ✓ | ✓ | | |
| PDCVRP | OR-Tools | | ✓ | ✓ | | | | | | ✓ | ✓ | ✓ | ✓ | |
| OPDCVRP | OR-Tools | | ✓ | ✓ | | | | | | ✓ | ✓ | ✓ | ✓ | ✓ |
| APDCVRP | OR-Tools | ✓ | | ✓ | | | | | | ✓ | ✓ | ✓ | ✓ | |
| AOPDCVRP | OR-Tools | ✓ | | ✓ | | | | | | ✓ | ✓ | ✓ | ✓ | ✓ |

# E   MODEL ARCHITECTURE AND TRAINING

## E.1   ENCODER

Similar to previous work (Kool et al., 2019; Kwon et al., 2020), we also use the encoder composed of $L$ stacked attention layers, where each attention layer consists of two sub-layers: an attention sub-layer and a Feed-Forward (FF) sub-layer, both of which use Instance Normalization (Ulyanov et al., 2016) and skip-connection (He et al., 2016).

Let $H^{(\ell-1)} = \{\mathbf{h}_i^{(\ell-1)}\}_{i=0}^n$ denote the input to the $\ell$-th attention layer for $\ell = 1, \ldots, L$. The outputs for the $i$-th node are computed as follows:

$$\widetilde{\mathbf{h}}_i^{(\ell)} = \mathrm{IN}^{(\ell)}\left(\mathbf{h}_i^{(\ell-1)} + \mathrm{MBM}^{(\ell)}\left(\mathbf{h}_i^{(\ell-1)}, H^{(\ell-1)}, \boldsymbol{D}, \boldsymbol{D}^{\mathrm{T}}, \boldsymbol{R}\right)\right), \tag{10}$$

$$\mathbf{h}_i^{(\ell)} = \mathrm{IN}^{(\ell)}\left(\widetilde{\mathbf{h}}_i^{(\ell)} + \mathrm{FF}^{(\ell)}\left(\widetilde{\mathbf{h}}_i^{(\ell)}\right)\right), \tag{11}$$

where $\mathrm{IN}(\cdot)$ denotes instance normalization (Ulyanov et al., 2016), MBM represents the adopted mixed bias module (see Equation (5) for details), and $\mathrm{FF}(\cdot)$ in Equation (11) corresponds to a fully connected neural network with ReLU activation.

After $L$ attention layers, the final node embeddings $H^{(L)} = \{\mathbf{h}_i^{(L)}\}_{i=0}^n$ encapsulate the advanced feature representations of each node.

Notably, to ensure dimensional consistency across bias channels, we normalize $\boldsymbol{D}$ and $\boldsymbol{D}^{\mathrm{T}}$ by dividing their maximum value before passing the MBM.

## E.2   DECODER

In URS, all parameters $\boldsymbol{\theta}_{dec}(\boldsymbol{\lambda})$ are adaptively generated conditioned on the multi-hot problem representation $\boldsymbol{\lambda}$ (see Appendix D for detailed problem representations). $\boldsymbol{\theta}_{dec}(\boldsymbol{\lambda})$ includes two MLP networks, which are $\mathrm{WEIGHT}(\boldsymbol{\lambda})$ and $\mathrm{BIAS}(\boldsymbol{\lambda})$. $\mathrm{WEIGHT}(\boldsymbol{\lambda})$ is used to generate linear projection matrices $[W_Q(\boldsymbol{\lambda}), W_K(\boldsymbol{\lambda}), W_V(\boldsymbol{\lambda})]$. $\mathrm{BIAS}(\boldsymbol{\lambda})$ in Equation (6) is used to generate bias weights for $\mathrm{Attention}(\cdot)$(see Equation (8)) and compatibility (see Equation (9)).

For $\mathrm{WEIGHT}(\boldsymbol{\lambda})$, we use a multi-layer MLP hypernetwork conditioned on a 13-dimensional problem representation $\boldsymbol{\lambda}$ to generate problem-specific adaptive parameters. Note that we factor $W_Q(\boldsymbol{\lambda}) \in \mathbb{R}^{(2d+1) \times d}$ into three vertically concatenated blocks $W_{\mathrm{first}}(\boldsymbol{\lambda}) \in \mathbb{R}^{d \times d}, W_{\mathrm{last}}(\boldsymbol{\lambda}) \in \mathbb{R}^{d \times d}$ and $W_C(\boldsymbol{\lambda}) \in \mathbb{R}^{1 \times d}$, applied respectively to the first visited node $\mathbf{h}_{\pi_1}^{(L)}$, the last visited node $\mathbf{h}_{\pi_{t-1}}^{(L)}$, and a scalar state feature $C_t \in \mathbb{R}^1$ (see Appendix E.3 for more details). Thus $\mathrm{WEIGHT}(\boldsymbol{\lambda})$ outputs the linear projection matrices $\{W_{\mathrm{first}}(\boldsymbol{\lambda}), W_{\mathrm{last}}(\boldsymbol{\lambda}), W_C(\boldsymbol{\lambda}), W_K(\boldsymbol{\lambda}), W_V(\boldsymbol{\lambda})\}$.

Furthermore, we adopt the parameter compression technique of Ha et al. (2016) to control model size. Specifically, we first generate an advanced embedding matrix $\boldsymbol{H}_{\mathrm{dec}}(\boldsymbol{\lambda}) = [\boldsymbol{h}_{\mathrm{dec}}^i(\boldsymbol{\lambda})|i = 1, 2, \ldots, 5] \in \mathbb{R}^{5 \times |\boldsymbol{\lambda}|}$ whose five row embeddings each parameterize one target projection matrix. It can be computed via three consecutive linear projections:

$$\boldsymbol{H}_{\mathrm{dec}}(\boldsymbol{\lambda}) = ((\boldsymbol{\lambda} W_1 + \mathbf{b}_1)W_2 + \mathbf{b}_2)W_3 + b_3, \tag{12}$$

where $W_1 \in \mathbb{R}^{|\lambda| \times d_h}$, $W_2 \in \mathbb{R}^{d_h \times d_h}$, $W_3 \in \mathbb{R}^{d_h \times (5 \times |\lambda|)}$, $\mathbf{b}_1 \in \mathbb{R}^{d_h}$, $\mathbf{b}_2 \in \mathbb{R}^{d_h}, \mathbf{b}_2 \in \mathbb{R}^{(5 \times |\lambda|)}$ are learnable parameters, $d_h$ is 256 in this work, following Lin et al. (2022). Next, we use five different linear projections to map the new representation $\boldsymbol{h}_{\mathrm{dec}}^i(\boldsymbol{\lambda})$ of each row to the corresponding decoder parameters. The computation can be expressed as:

$$W_{\mathrm{first}}(\boldsymbol{\lambda}) = \boldsymbol{h}_{\mathrm{dec}}^1(\boldsymbol{\lambda})W_{\mathbf{hyper}}^1, \ W_{\mathrm{last}}(\boldsymbol{\lambda}) = \boldsymbol{h}_{\mathrm{dec}}^2(\boldsymbol{\lambda})W_{\mathbf{hyper}}^2, \ W_C(\boldsymbol{\lambda}) = \boldsymbol{h}_{\mathrm{dec}}^3(\boldsymbol{\lambda})W_{\mathbf{hyper}}^3, \tag{13}$$

$$W_K(\boldsymbol{\lambda}) = \boldsymbol{h}_{\mathrm{dec}}^4(\boldsymbol{\lambda})W_{\mathbf{hyper}}^4, \ W_V(\boldsymbol{\lambda}) = \boldsymbol{h}_{\mathrm{dec}}^5(\boldsymbol{\lambda})W_{\mathbf{hyper}}^5, \tag{14}$$

where $W_{\mathbf{hyper}}^1 \in \mathbb{R}^{|\lambda| \times (d \times d)}$, $W_{\mathbf{hyper}}^2 \in \mathbb{R}^{|\lambda| \times (d \times d)}$, $W_{\mathbf{hyper}}^3 \in \mathbb{R}^{|\lambda| \times (1 \times d)}$, $W_{\mathbf{hyper}}^4 \in \mathbb{R}^{|\lambda| \times (d \times d)}$, $W_{\mathbf{hyper}}^5 \in \mathbb{R}^{|\lambda| \times (d \times d)}$ are learnable parameters. In this way, we use a simple MLP hypernetwork $\mathrm{WEIGHT}(\boldsymbol{\lambda})$ to generate used linear projection matrices $\{W_Q(\boldsymbol{\lambda}), W_K(\boldsymbol{\lambda}), W_V(\boldsymbol{\lambda})\}$ conditioned on the multi-hot problem representation $\boldsymbol{\lambda}$, enabling the adaptive construction of high-quality solutions for different problems.

### E.3 PROBLEM STATE $C_t$

In this section, we detail the problems that require additional state signals. If a problem is not mentioned in this section, it means we will not apply any extra state to it, i.e., $C_t = 0$.

For all problems with the capacity constraint, we explicitly impose **only remaining load**, while diverse additional constraints (e.g., time windows, duration limit, and backhaul) are enforced implicitly as $\mathcal{M}_t$ masks infeasible candidate nodes to guarantee valid solutions, instead of being fully enumerated in the input of the decoder.

For OP, we keep track of the remaining maximum length at time $t$, and $C_t$ represents the remaining routing length that can be moved, and we normalize it to $[0, 1]$ by dividing by the maximum length (i.e., 4.0 in OP100). The setting is the same as AM (Kool et al., 2019).

For PCTSP and SPCTSP, $C_t$ is the remaining prize to collect, and we do not provide any information about the penalties, as this is irrelevant for the remaining decisions, following Kool et al. (2019).

For the problem state, we provide an ablation study in Appendix K.2.

### E.4 TRAINING

Following Kwon et al. (2020), we use $n$ trajectories with distinct starting nodes in training. URS is trained by the REINFORCE (Williams, 1992) algorithm with a shared baseline (Kwon et al., 2020):

$$\nabla_\theta \mathcal{L}(\boldsymbol{\theta}) = \mathbb{E}_{p(\boldsymbol{\pi}|\mathcal{G},\boldsymbol{\theta})}[(f(\boldsymbol{\pi}|\mathcal{G}) - \frac{1}{n}\sum_{i=1}^{n} f(\boldsymbol{\pi}^i|\mathcal{G}))\nabla_\theta \log p_\theta(\boldsymbol{\pi}|\mathcal{G})], \quad (15)$$

$$p_\theta(\boldsymbol{\pi} \mid \mathcal{G}) = \prod_{t=2}^{n} p_\theta(\pi_t \mid \mathcal{G}, \pi_{1:t-1}), \quad (16)$$

where the objective function $f(\boldsymbol{\pi}|\mathcal{G})$ represents the total reward (e.g., the negative value of tour length) of instance $\mathcal{G}$ given a specific solution $\boldsymbol{\pi}$.

## F ADAPTATION ATTENTION FREE MODULE

Following Zhou et al. (2024a), we implement $\mathrm{Attention}(\cdot)$ using an adaptation attention free module (AAFM) to enhance geometric pattern recognition for routing problems. Given the input $X$, AAFM first transforms it into $Q$, $K$, and $V$ through corresponding linear projection operations:

$$Q = XW^Q, \quad K = XW^K, \quad V = XW^V, \quad (17)$$

where $W^Q$, $W^K$, and $W^V$ are learnable matrices. The AAFM computation is then expressed as:

$$\mathrm{Attention}(Q, K, V, A) = \sigma(Q) \odot \frac{\exp(A)(\exp(K) \odot V)}{\exp(A)\exp(K)}, \quad (18)$$

where $\sigma$ denotes the sigmoid function, $\odot$ represents the element-wise product, and $A = \{a_{ij}\}$ denotes the pair-wise adaptation bias. For distance matrices, the corresponding adaptation bias $f(\alpha, N, d_{ij}) = -\alpha \cdot \log_2 N \cdot d_{i,j}$. Here, $N$ denotes the total number of nodes (i.e., problem size), and $\alpha$ is generated via a lightweight network $\mathrm{BIAS}(\boldsymbol{\lambda})$ in our paper (see Equation (6)). Notably, for the relation matrix, we remove the scale $N$ in the calculation of adaptation bias because the relation is scale-independent.

Compared to multi-head attention (MHA) (Vaswani et al., 2017), AAFM enables the model to explicitly capture instance-specific knowledge by updating pair-wise adaptation biases while exhibiting lower computational overhead. Further details are provided in the related work section mentioned above.

# G  LLM-DRIVEN CONSTRAINT SATISFACTION MECHANISM

To operationalize the UDR framework for a vast spectrum of routing problems, we introduce an automated constraint satisfaction mechanism based on LLMs. This mechanism obviates the need for manual, problem-specific masks by automatically designing executable **mask generators**. This approach follows the generate-and-validate pipeline, transforming constraint specifications into reliable code.

## G.1  MASK GENERATOR DESIGN

This process begins by prompting an LLM with a prompt that comprises two key elements: 1) a natural language **task description** requires explicit and precise specification of both the optimization objective for the task and all associated operational constraints, as this component delineates the core problem to resolve; and 2) a standardized Python **code template** formally defines the environment by outlining the precise semantics, shapes, and data types of all input and output tensors, thereby specifying how the solution should interface with the existing code.

Our generation process relies on the clarity and accuracy of these components. When the problem is well-posed with clear objectives and precise I/O, the mask generator design process is highly efficient. Conversely, failures usually result from incomplete or ambiguous specification of the core components, not minor phrasing variations.

Here is an example for generating a mask for the CVRPTW that enforces constraints: 1) no revisiting nodes except the depot, 2) no exceeding current vehicle capacity, 3) arrival before node time windows, 4) only the depot being revisitable, and 5) no consecutive depot visits unless all other nodes are visited.

**Task Description:**   Given the current node, visited nodes, coordinates, demands, service times, available time windows, current capacity, current time, and the number of nodes, design a function to mask nodes that are either already visited, have demands exceeding capacity, or cannot be reached before their time window closed, considering the travel time from the current node (where travel time equals distance due to unit speed) plus the required service time at that node. Only the depot may be visited more than once. However, you can't visit the depot twice in a row unless all other nodes have already been visited.

**Code Template:**

```python
import torch
def mask_illegal_nodes(current_node: torch.Tensor, visited_nodes: torch.
    Tensor, coordinates: torch.Tensor,
                        demands: torch.Tensor, service_time: torch.Tensor,
     time_windows: torch.Tensor,
                        current_capacity: torch.Tensor, current_time:
    torch.Tensor, num_nodes: int) -> torch.Tensor:
    """
    Design a function to mask nodes that are either already visited, have
     demands exceeding capacity, or cannot be reached within their time
    window in each step.
    The operation processes batches in parallel, so no loops over the
    batch dimension are allowed-inputs must use batch-first shapes.

    Args:
    current_node: ID of the current node. Shape: [batch_size, 1]
    visited_nodes: Array of IDs of visited nodes and depot. Shape: [
    batch_size, num_visited_nodes]
    coordinates: Coordinates of the deopt and all nodes. Shape: [
    batch_size, num_nodes+1, 2]
    demands: Array of demands of the deopt and all nodes. Shape: [
    batch_size, num_nodes+1]
    service_time: Array of service time of the deopt and all nodes. Shape
    : [batch_size, num_nodes+1]
    time_windows: Array of available time window of the deopt and all
    nodes. Shape: [batch_size, num_nodes+1, 2]
```

```
16      current_capacity: Value of current capacity. Shape: [batch_size, 1]
17      current_time: Value of current time. Shape: [batch_size, 1]
18      num_nodes: The number of nodes.
19
20      Return:
21      mask: Boolean mask indicating valid deopt or nodes (True = selectable
        , False = masked). Shape: [batch_size, num_nodes+1]
22      """
23      return mask
```

The LLM completes the function body with the prompt guidance. The candidate mask generator is
an executable Python snippet implementing the described logic:

```
1  def mask_illegal_nodes(current_node: torch.Tensor, visited_nodes: torch.
       Tensor, coordinates: torch.Tensor, demands: torch.Tensor,
       service_time: torch.Tensor, time_windows: torch.Tensor,
       current_capacity: torch.Tensor, current_time: torch.Tensor, num_nodes
       : int) -> torch.Tensor:
2      """
3      Design a function to mask nodes that are either already visited, have
        demands exceeding capacity, or cannot be reached within their time
       window in each step.
4      The operation processes batches in parallel, so no loops over the
       batch dimension are allowed\u2014inputs must use batch-first shapes.
5      Args:
6      current_node: ID of the current node. Shape: [batch_size, 1]
7      visited_nodes: Array of IDs of visited nodes and depot. Shape: [
       batch_size, num_visited_nodes]
8      coordinates: Coordinates of the deopt and all nodes. Shape: [
       batch_size, num_nodes+1, 2]
9      demands: Array of demands of the deopt and all nodes. Shape: [
       batch_size, num_nodes+1]
10     service_time: Array of service time of the deopt and all nodes. Shape
       : [batch_size, num_nodes+1]
11     time_windows: Array of available time window of the deopt and all
       nodes. Shape: [batch_size, num_nodes+1, 2]
12     current_capacity: Value of current capacity. Shape: [batch_size, 1]
13     current_time: Value of current time. Shape: [batch_size, 1]
14     num_nodes: The number of nodes.
15     Return:
16     mask: Boolean mask indicating valid deopt or nodes (True = selectable
       , False = masked). Shape: [batch_size, num_nodes+1]
17     """
18     batch_size = current_node.size(0)
19     device = current_node.device
20     # Shape helpers
21     N = num_nodes + 1  # Including depot at idx 0
22     # 1) Mask already visited nodes (except depot handled later)
23     # Create visited mask: True if visited, False otherwise
24     visited_mask = torch.zeros((batch_size, N), dtype=torch.bool, device=
       device)
25     # visited_nodes: [batch_size, num_visited_nodes], scatter True for
       each node visited
26     # Use one-hot scatter add to build visited_mask
27     idx = visited_nodes.unsqueeze(-1)  # [batch_size, num_visited_nodes,
       1]
28     # For scatter we reshape visited_mask to 1D first per batch later
29     # Instead do scatter_ at dim=1
30     visited_mask = visited_mask.scatter_(1, visited_nodes, True)
31     # 2) Mask nodes with demands exceeding capacity
32     # demands: [batch_size, N], current_capacity: [batch_size,1]
33     demand_mask = demands <= current_capacity  # True if demand <=
       capacity
34     # 3) Mask nodes that cannot be reached within their time window
```

```python
35      # Calculate travel time = Euclidean distance current_node -> all
        nodes
36      # coordinates: [batch_size, N, 2]
37      # current_node: [batch_size, 1]
38      current_coords = torch.gather(coordinates, 1, current_node.unsqueeze
        (-1).expand(-1, -1, 2)).squeeze(1)  # [batch_size, 2]
39      # All nodes coords: coordinates [batch_size, N, 2]
40      # Compute distance = Euclidean norm
41      diff = coordinates - current_coords.unsqueeze(1)  # [batch_size, N,
        2]
42      travel_time = diff.norm(dim=2)  # [batch_size, N]
43      # Arrival time at each candidate node (without waiting): current_time
         + travel_time
44      arrival_time = current_time + travel_time  # [batch_size, N]
45      # Time windows: [batch_size, N, 2]
46      window_start = time_windows[..., 0]
47      window_end = time_windows[..., 1]
48      # We can arrive early but must wait for window start if arrival_time
        < window_start
49      start_service_time = torch.max(arrival_time, window_start)  # [
        batch_size, N]
50      # Departure time from node = start_service_time + service_time
51      departure_time = start_service_time + service_time  # [batch_size, N]
52      # Node feasible if arrival_time <= window_end (i.e., can start
        service before window closes)
53      time_mask = arrival_time <= window_end  # can arrive on time to start
         service
54      # 4) Mask depot revisits constraints:
55      # Depot is node 0
56      depot_idx = torch.zeros(batch_size, dtype=torch.long, device=device)
57      # Check if current_node == depot (node 0) to mask depot twice in a
        row
58      current_is_depot = (current_node.squeeze(1) == 0)  # [batch_size]
59      # Check if all other nodes are visited (excluding depot)
60      # visited_mask: True for visited nodes incl depot
61      # Count number of visited non-depot nodes:
62      visited_non_depot = visited_mask[:, 1:]  # [batch_size, N-1]
63      all_others_visited = visited_non_depot.all(dim=1)  # [batch_size]
64      # Mask depot if current node is depot and last visited node is depot
        (i.e., depot twice in a row)
65      # but allow depot if all other nodes have been visited
66      # So depot_masked_forbidden = True if depot is forbidden, False if
        allowed
67      # By problem statement: \"Only the depot may be visited multiple
        times. However, you can\u2019t visit the depot twice in a row unless
        all other nodes have already been visited.\"
68      # Check if previous node == depot (as we have current_node only, we
        assume last visit is current_node)
69      # So if current_node == depot and next node is also depot, it is
        forbidden unless all other have been visited
70      # Initially depot_mask lets depot be selected
71      depot_mask = torch.ones((batch_size,), dtype=torch.bool, device=
        device)
72      # If current node is depot and not all others visited => mask depot =
         False (mask it)
73      mask_depot = current_is_depot & (~all_others_visited)  # batches
        where depot revisit is forbidden
74      depot_mask[mask_depot] = False  # mask depot for these batches
75      # Build full mask of depot for all batches:
76      # depot_mask shape: [batch_size], need to broadcast to [batch_size, N
        ]
77      # depot_mask[:, None] for expand
78      depot_mask_full = torch.ones((batch_size, N), dtype=torch.bool,
        device=device)
79      depot_mask_full[:, 0] = depot_mask  # depot node mask per batch
```

```
80      # 5) Depot can be visited more than once, so remove depot from
        visited_mask
81      visited_mask[:, 0] = False  # depot can be visited multiple times (
        masking will depend on depot_mask)
82      # Combine all masks
83      # True = selectable, False = masked
84      final_mask = (~visited_mask) & demand_mask & time_mask &
        depot_mask_full
85      return final_mask
```

## G.2 AUTOMATED VALIDATION

Each candidate generator is then validated. We execute it within the URS solver or a simple simulated solver on a set of problem instances and use a problem-specific **checker** function to verify if the resulting solutions are constraint-compliant. For the CVRPTW example, the checker validates fundamental feasibility constraints, including capacity limits, time window adherence, and unique customer visitation. The restriction on consecutive depot visits is excluded from validation, serving instead as a mechanism to prevent algorithmic stagnation.

```
1  import torch
2
3  def cvrptw_checker(env, solutions):
4      batch_size, num_steps = solutions.shape
5      # all nodes visited once
6      all_nodes_visited = (torch.arange(env.problem_size + 1).unsqueeze(0).
       expand(batch_size, -1) == sorted[:, -env.problem_size - 1:]).all(1)
7      if not all_nodes_visited.any():
8          return -float('inf')
9      for i in range(num_steps):
10          crt_state = env.update(solutions[:, i])
11          # time window
12          if (env.current_time > crt_state.selected_time[:, -1]).any():
13              return -float('inf')
14          # capacity
15          if (env.available_capacity < crt_state.selected_demand).any():
16              return -float('inf')
17      return -1.0
```

## G.3 AHD FRAMEWORK INTEGRATION

For most common VRP variants, manually writing a masking function for each problem is possible by decoupling constraints. However, it is fundamentally more challenging than writing a post-hoc checker. While a checker simply validates a static, completed solution, a masking function must proactively guarantee feasibility by determining whether a current action would render the future route infeasible. Even if constraints can be checked individually, implementing this look-ahead logic correctly and efficiently is not a trivial task, which requires significant domain expertise for each constraint. Our mechanism automates this manual engineering task for the vast majority of variants, thereby lowering the barrier for users.

In our proposed framework, each masking function is validated by a problem-specific checker on a set of test instances. Finding the valid masking function is an offline generate-and-validate process, rather than relying on an LLM to perform online reasoning during the inference phase of the neural solver. We leverage automatic heuristic design frameworks (such as EoH Liu et al. (2024b)) to enable the LLM to iteratively refine the code of the masking function based on the checker's feedback. Any candidate masking function that produces even one infeasible solution in the test instances (i.e., validity rate $< 100\%$) is discarded. Once a valid masking function is obtained, it is used in the subsequent NCO process, and the offline process is terminated. In other words, every final valid masking function produced by the LLM is reliable, and it will generate valid masks for each solution construction step of the NCO process. The cost of the offline process is a one-time amortized cost per problem variant, as the generated masking function can be reused indefinitely for any instance of that variant.

### G.4 Consistency Across Different LLMs

To evaluate the influence of our framework on different LLMs, we benchmarked its performance using both GPT-3.5-turbo and GPT-4o-mini. The results, presented in Table 9 and Table 10, show that while the query efficiency varies with the underlying model's capability, the core outcome remains consistent: our framework successfully generates the necessary masking functions with both models. This finding demonstrates that our approach doesn't rely on a specific proprietary model but rather provides a robust methodology adaptable to various powerful LLMs.

Table 9: Cost analysis of generating mask functions for 11 training tasks using GPT-3.5-turbo.

|        | ATSP   | TSP   | OP     | PCTSP | PDTSP | OCVRPTW |
| ------ | ------ | ----- | ------ | ----- | ----- | ------- |
| #Query | 1      | 1     | 1      | 4     | 1     | 5       |
| #Token | 468    | 514   | 1359   | 3403  | 1277  | 11054   |
| Time   | <1min  | <1min | <1min  | 1min  | <1min | 1min    |

|        | ACVRP | CVRP  | CVRPTW | CVRPB | OCVRP | Avg.  |
| ------ | ----- | ----- | ------ | ----- | ----- | ----- |
| #Query | 1     | 1     | 5      | 2     | 1     | 2     |
| #Token | 1253  | 1371  | 9935   | 1920  | 1264  | 3074  |
| Time   | <1min | <1min | <1min  | <1min | <1min | <1min |

Table 10: Cost analysis of generating mask functions for 11 training tasks using GPT-4o-mini.

|        | ATSP  | TSP   | OP    | PCTSP | PDTSP | OCVRPTW |
| ------ | ----- | ----- | ----- | ----- | ----- | ------- |
| #Query | 1     | 1     | 3     | 1     | 10    | 17      |
| #Token | 459   | 387   | 2031  | 650   | 9877  | 18031   |
| Time   | <1min | <1min | <1min | <1min | 2min  | 3min    |

|        | ACVRP | CVRP  | CVRPTW | CVRPB | OCVRP | Avg.  |
| ------ | ----- | ----- | ------ | ----- | ----- | ----- |
| #Query | 1     | 1     | 15     | 3     | 2     | 5     |
| #Token | 1273  | 1245  | 15103  | 1920  | 1264  | 4749  |
| Time   | <1min | <1min | 3min   | <1min | <1min | 1min  |

## H Training and Model Settings

### H.1 Multi-Task Version

The detailed information about the hyperparameter settings can be found in Table 11. The training of URS is conducted on a single NVIDIA GeForce RTX 4090 GPU (24GB memory). The training procedure requires only about 95 hours ($\approx 4$ days) in total (500 epochs at approximately 12 minutes each) to obtain a unified parameter set capable of addressing a broad class of routing problems.

### H.2 Single-Task Version

URS-STL is trained on data from only a single problem type (e.g., "URS-STL (CVRP)" is trained only on CVRP). The URS-STL variant uses exactly the same model architecture, hyperparameters (e.g., embedding dimension, number of layers, optimizer), and training setup (e.g., initial learning rate, batch size) as the multi-task URS described in Section 5.1 and Appendix H.1. The only modification we made is to the training duration. We observed that the single-task models converge significantly faster than the multi-task model. Therefore, we train URS-STL models for 300 epochs, with a learning rate decay applied at the 251st epoch. This contrasts with the multi-task URS, which is trained for 500 epochs with decay applied at the 451st epoch. This adjustment is made solely to reflect the faster convergence of the single-task training. All other hyperparameters and model architecture remain identical.

Table 11: The hyperparameter settings of URS.

| Hyperparameter | Value |
|---|---|
| Optimizer | AdamW |
| Clipping parameter $\zeta$ | 50 |
| Initial learning rate | $10^{-4}$ |
| Epochs of learning rate decay | [451] |
| Factor of learning rate decay | 0.1 |
| Weight decay | $10^{-6}$ |
| The number of encoder layers $L$ | 12 |
| Embedding dimension $d$ | 128 |
| Feed forward dimension | 512 |
| Dimension $d_h$ in Equation (12) | 256 |
| Training scale | 100 |
| Batches of each epoch | $2,000$ |
| Batch size | 128 |
| Training Epochs | 500 |

## I GENERALIZATION TO UNSEEN VRPs

To comprehensively evaluate zero-shot generalization of URS, we test its performance on a diverse set of 96 unseen problems. This set is designed to cover both established and novel challenges, including 44 well-studied VRP variants from the NCO literature and 52 newly introduced variants that have not been explored before. The detailed results are presented in Table 12 and Table 13. On widely-studied constraints (i.e., C, O, L, TW, MD, B, BP), URS continues to deliver high-quality solutions. Notably, URS merely selects the next node from a given feasible candidates without any additional domain knowledge, making its performance especially meaningful. Benefiting from our UDR, URS can directly address previously unexplored variants, which compound asymmetry and PD relations. Without any fine-tuning, URS still constructs high-quality solutions for them, strengthening the broad problem generality and deployment efficiency.

Table 12: Zero-shot generalization performance across 44 VRP variants that have been explored in current literature (Liu et al., 2024a; Berto et al., 2024) ($N = 100$).

| Method | CVRPBP | | OCVRPBP | | CVRPBPL | | OCVRPBPTW | | CVRPBPLTW | |
|---|---|---|---|---|---|---|---|---|---|---|
| | Gap | Time | Gap | Time | Gap | Time | Gap | Time | Gap | Time |
| Oracle Solver | 0.00% | 6.6m | 0.00% | 6.6m | 0.00% | 6.6m | 0.00% | 6.6m | 0.00% | 6.6m |
| MVMoE | 13.95% | 14s | 17.36% | 14s | 13.38% | 14s | 2.29% | 15s | 10.46% | 16s |
| MTPOMO | 13.71% | 7s | 16.94% | 8s | 13.44% | 8s | 3.26% | 9s | 11.35% | 10s |
| ReLD-MTL | 13.57% | 9s | 14.83% | 9s | 12.96% | 10s | 2.27% | 11s | 10.80% | 12s |
| URS | **12.95%** | 7s | **14.30%** | 7s | **12.65%** | 8s | **-3.47%** | 8s | **9.74%** | 10s |

| Method | CVRPBPTW | | OCVRPBPL | | OCVRPBPLTW | | CVRPL | | CVRPBL | |
|---|---|---|---|---|---|---|---|---|---|---|
| | Gap | Time | Gap | Time | Gap | Time | Gap | Time | Gap | Time |
| Oracle Solver | 0.00% | 6.6m | 0.00% | 6.6m | 0.00% | 6.6m | 0.00% | 16m | 0.00% | 3.5h |
| MVMoE | 10.85% | 15s | 16.88% | 14s | 1.84% | 16s | 0.26% | 12s | 1.35% | 12s |
| MTPOMO | 11.53% | 9s | 16.39% | 8s | 2.77% | 10s | 0.48% | 6s | 1.79% | 6s |
| ReLD-MTL | 11.07% | 11s | 14.22% | 10s | 1.79% | 11s | **0.02%** | 8s | **1.01%** | 7s |
| URS | **9.73%** | 9s | **13.81%** | 8s | **-3.77%** | 9s | 0.43% | 7s | 1.65% | 6s |

| Method | OCVRPB | | CVRPLTW | | OCVRPBTW | | CVRPBLTW | | OCVRPL | |
|---|---|---|---|---|---|---|---|---|---|---|
| | Gap | Time | Gap | Time | Gap | Time | Gap | Time | Gap | Time |
| Oracle Solver | 0.00% | 3.5h | 0.00% | 3.5h | 0.00% | 3.5h | 0.00% | 3.5h | 0.00% | 3.5h |
| MVMoE | 7.09% | 12s | 1.47% | 16s | 9.95% | 15s | 7.33% | 16s | 3.15% | 13s |
| MTPOMO | 7.34% | 5s | 1.92% | 9s | 10.45% | 6s | 7.75% | 8s | 3.44% | 6s |
| ReLD-MTL | **5.36%** | 6s | **1.17%** | 11s | **9.29%** | 8s | **6.94%** | 9s | **2.31%** | 8s |
| URS | 9.35% | 6s | 2.67% | 9s | 13.77% | 7s | 9.18% | 8s | 3.22% | 7s |

| Method | CVRPBTW | | OCVRPBL | | OCVRPLTW | | OCVRPBLTW | | MDCVRPBP | |
|---|---|---|---|---|---|---|---|---|---|---|
| | Gap | Time | Gap | Time | Gap | Time | Gap | Time | Gap | Time |
| Oracle Solver | 0.00% | 3.5h | 0.00% | 3.5h | 0.00% | 3.5h | 0.00% | 3.5h | 0.00% | 6.6m |
| MVMoE | 7.08% | 15s | 7.12% | 12s | 3.90% | 15s | 10.01% | 15s | 64.91% | 33s |
| MTPOMO | 7.41% | 7s | 7.34% | 7s | 4.37% | 8s | 10.50% | 7s | 45.66% | 24s |
| ReLD-MTL | **6.74%** | 8s | **5.41%** | 7s | **3.16%** | 9s | **9.22%** | 8s | 40.28% | 27s |
| URS | 8.94% | 8s | 9.47% | 7s | 5.12% | 9s | 13.72% | 8s | **34.91%** | 24s |

| Method | MDOCVRPBP | | MDCVRPBPL | | MDCVRPBPTW | | MDCVRPBPLTW | | MDOCVRPBPTW | |
|---|---|---|---|---|---|---|---|---|---|---|
| | Gap | Time | Gap | Time | Gap | Time | Gap | Time | Gap | Time |
| Oracle Solver | 0.00% | 6.6m | 0.00% | 6.6m | 0.00% | 6.6m | 0.00% | 6.6m | 0.00% | 6.6m |
| MVMoE | 52.16% | 33s | 64.79% | 36s | 46.18% | 38s | 54.23% | 42s | 54.76% | 39s |
| MTPOMO | 40.15% | 24s | 45.84% | 26s | 34.19% | 29s | 40.26% | 32s | 40.32% | 30s |
| ReLD-MTL | 34.53% | 27s | 39.06% | 29s | 34.08% | 34s | 36.59% | 37s | 37.16% | 35s |
| URS | **19.13%** | 22s | **34.49%** | 26s | **11.17%** | 26s | **25.57%** | 32s | **25.77%** | 30s |

| Method | MDOCVRPBPL | | MDOCVRPBPLTW | | MDCVRP | | MDCVRPTW | | MDOCVRP | |
|---|---|---|---|---|---|---|---|---|---|---|
| | Gap | Time | Gap | Time | Gap | Time | Gap | Time | Gap | Time |
| Oracle Solver | 0.00% | 6.6m | 0.00% | 6.6m | 0.00% | 6.6m | 0.00% | 6.6m | 0.00% | 6.6m |
| MVMoE | 51.72% | 34s | 45.23% | 40s | 31.90% | 31s | 53.64% | 41s | 29.08% | 31s |
| MTPOMO | 39.34% | 27s | 33.37% | 32s | 23.75% | 23s | 36.10% | 31s | 27.46% | 23s |
| ReLD-MTL | 34.48% | 29s | 33.83% | 36s | 17.37% | 25s | **30.90%** | 34s | 22.32% | 25s |
| URS | **18.90%** | 24s | **10.96%** | 28s | **9.65%** | 23s | 37.26% | 32s | **22.01%** | 23s |

| Method | MDCVRPL | | MDCVRPB | | MDOCVRPTW | | MDOCVRPB | | MDOCVRPBL | |
|---|---|---|---|---|---|---|---|---|---|---|
| | Gap | Time | Gap | Time | Gap | Time | Gap | Time | Gap | Time |
| Oracle Solver | 0.00% | 6.6m | 0.00% | 6.6m | 0.00% | 6.6m | 0.00% | 6.6m | 0.00% | 6.6m |
| MVMoE | 33.42% | 34s | 21.70% | 26s | 51.57% | 40s | 28.21% | 26s | 20.35% | 29s |
| MTPOMO | 24.32% | 25s | 12.89% | 19s | 35.71% | 29s | 25.06% | 20s | 11.10% | 20s |
| ReLD-MTL | 18.48% | 28s | **9.38%** | 21s | 33.55% | 33s | 18.50% | 21s | **8.26%** | 23s |
| URS | **15.32%** | 26s | 13.14% | 19s | **26.05%** | 25s | **15.17%** | 18s | 11.19% | 21s |

| Method | MDCVRPLTW | | MDOCVRPBTW | | MDOCVRPBLTW | | MDOCVRPL | | MDCVRPBTW | |
|---|---|---|---|---|---|---|---|---|---|---|
| | Gap | Time | Gap | Time | Gap | Time | Gap | Time | Gap | Time |
| Oracle Solver | 0.00% | 6.6m | 0.00% | 6.6m | 0.00% | 6.6m | 0.00% | 6.6m | 0.00% | 6.6m |
| MVMoE | 55.55% | 44s | 62.47% | 35s | 63.18% | 37s | 29.33% | 33s | 65.90% | 35s |
| MTPOMO | 37.34% | 32s | 45.55% | 25s | 45.07% | 27s | 27.21% | 25s | 45.94% | 25s |
| ReLD-MTL | **31.56%** | 36s | 43.36% | 28s | 40.57% | 30s | **22.05%** | 27s | 41.28% | 28s |
| URS | 38.04% | 35s | **30.32%** | 22s | **37.70%** | 27s | 22.15% | 25s | **37.78%** | 25s |

| Method | MDOCVRPBL | | MDOCVRPLTW | | MDOCVRPBLTW | | SPCTSP | | | |
|---|---|---|---|---|---|---|---|---|---|---|
| | Gap | Time | Gap | Time | Gap | Time | Gap | Time | – | |
| Oracle Solver | 0.00% | 6.6m | 0.00% | 6.6m | 0.00% | 6.6m | 0.00% | 1.6h | – | |
| MVMoE | 28.70% | 28s | 51.09% | 43s | 62.10% | 37s | – | | – | |
| MTPOMO | 25.01% | 21s | 35.39% | 31s | 45.48% | 27s | – | | – | |
| ReLD-MTL | 18.50% | 22s | 33.26% | 35s | 43.27% | 30s | – | | – | |
| AM | – | | – | | – | | 2.69% | 5s | – | |
| URS | **14.90%** | 20s | **25.59%** | 27s | **30.41%** | 24s | **-2.37%** | 5s | – | |

Table 13: Zero-shot generalization performance across 52 VRP variants that have not yet been explored in the current literature ($N = 100$).

| Method | ACVRPL | | ACVRPBL | | ACVRPLTW | | ACVRPBLTW | | AOCVRPL | |
|---|---|---|---|---|---|---|---|---|---|---|
| | Gap | Time | Gap | Time | Gap | Time | Gap | Time | Gap | Time |
| Oracle Solver | 0.00% | 6.6m | 0.00% | 6.6m | 0.00% | 6.6m | 0.00% | 6.6m | 0.00% | 6.6m |
| URS | -2.24% | 1.9m | 6.18% | 2.4m | 12.22% | 2.7m | 10.18% | 2.4m | 37.36% | 3.1m |

| Method | AOCVRPBL | | AOCVRPLTW | | AOCVRPBLTW | | ACVRPBPL | | ACVRPBPLTW | |
|---|---|---|---|---|---|---|---|---|---|---|
| | Gap | Time | Gap | Time | Gap | Time | Gap | Time | Gap | Time |
| Oracle Solver | 0.00% | 6.6m | 0.00% | 6.6m | 0.00% | 6.6m | 0.00% | 6.6m | 0.00% | 6.6m |
| URS | 17.23% | 2.7m | 16.35% | 2.8m | 14.32% | 2.5m | 21.69% | 2.6m | -0.58% | 2.8m |

| Method | AOCVRPBPL | | AOCVRPBPLTW | | ACVRPB | | ACVRPTW | | AOCVRP | |
|---|---|---|---|---|---|---|---|---|---|---|
| | Gap | Time | Gap | Time | Gap | Time | Gap | Time | Gap | Time |
| Oracle Solver | 0.00% | 6.6m | 0.00% | 6.6m | 0.00% | 6.6m | 0.00% | 6.6m | 0.00% | 6.6m |
| URS | 28.25% | 2.7m | 2.54% | 2.7m | 7.10% | 2.1m | 11.28% | 2.5m | 37.63% | 2.9m |

| Method | AOCVRPTW | | AOCVRPB | | AOCVRPBTW | | ACVRPBTW | | ACVRPBP | |
|---|---|---|---|---|---|---|---|---|---|---|
| | Gap | Time | Gap | Time | Gap | Time | Gap | Time | Gap | Time |
| Oracle Solver | 0.00% | 6.6m | 0.00% | 6.6m | 0.00% | 6.6m | 0.00% | 6.6m | 0.00% | 6.6m |
| URS | 17.47% | 2.6m | 18.05% | 2.5m | 14.45% | 2.3m | 9.95% | 2.2m | 21.94% | 2.4m |

| Method | AOCVRPBP | | AOCVRPBPTW | | ACVRPBPTW | | APDTSP | | AMDCVRPL | |
|---|---|---|---|---|---|---|---|---|---|---|
| | Gap | Time | Gap | Time | Gap | Time | Gap | Time | Gap | Time |
| Oracle Solver | 0.00% | 6.6m | 0.00% | 6.6m | 0.00% | 6.6m | 0.00% | 6.6m | 0.00% | 6.6m |
| URS | 28.27% | 2.5m | 3.28% | 2.5m | 0.12% | 2.6m | 6.21% | 1m | 7.15% | 6.1m |

| Method | AMDCVRPBL | | AMDCVRPLTW | | AMDCVRPBLTW | | AMDOCVRPL | | AMDOCVRPBL | |
|---|---|---|---|---|---|---|---|---|---|---|
| | Gap | Time | Gap | Time | Gap | Time | Gap | Time | Gap | Time |
| Oracle Solver | 0.00% | 6.6m | 0.00% | 6.6m | 0.00% | 6.6m | 0.00% | 6.6m | 0.00% | 6.6m |
| URS | 17.51% | 6.5m | 19.98% | 8.8m | 16.73% | 7.5m | 47.80% | 8.6m | 27.23% | 7.1m |

| Method | AMDOCVRPLTW | | AMDOCVRPBLTW | | AMDCVRPBPL | | AMDCVRPBPLTW | | AMDOCVRPBPL | |
|---|---|---|---|---|---|---|---|---|---|---|
| | Gap | Time | Gap | Time | Gap | Time | Gap | Time | Gap | Time |
| Oracle Solver | 0.00% | 6.6m | 0.00% | 6.6m | 0.00% | 6.6m | 0.00% | 6.6m | 0.00% | 6.6m |
| URS | 24.90% | 7.9m | 22.84% | 6.7m | 29.07% | 7.2m | 4.45% | 8.3m | 29.30% | 7.3m |

| Method | AMDOCVRPBPLTW | | AMDCVRP | | AMDCVRPTW | | AMDOCVRP | | AMDCVRPB | |
|---|---|---|---|---|---|---|---|---|---|---|
| | Gap | Time | Gap | Time | Gap | Time | Gap | Time | Gap | Time |
| Oracle Solver | 0.00% | 6.6m | 0.00% | 6.6m | 0.00% | 6.6m | 0.00% | 6.6m | 0.00% | 6.6m |
| URS | 8.63% | 7.5m | 3.56% | 5.3m | 20.96% | 8.1m | 48.02% | 7.8m | 17.35% | 5.8m |

| Method | AMDOCVRPTW | | AMDOCVRPB | | AMDOCVRPBTW | | AMDCVRPBTW | | AMDOCVRPBP | |
|---|---|---|---|---|---|---|---|---|---|---|
| | Gap | Time | Gap | Time | Gap | Time | Gap | Time | Gap | Time |
| Oracle Solver | 0.00% | 6.6m | 0.00% | 6.6m | 0.00% | 6.6m | 0.00% | 6.6m | 0.00% | 6.6m |
| URS | 26.02% | 7.3m | 27.78% | 6.5m | 23.67% | 6.2m | 17.52% | 6.8m | 30.15% | 6.5m |

| Method | AMDOCVRPBP | | AMDOCVRPBPTW | | AMDCVRPBPTW | | PDCVRP | | OPDCVRP | |
|---|---|---|---|---|---|---|---|---|---|---|
| | Gap | Time | Gap | Time | Gap | Time | Gap | Time | Gap | Time |
| Oracle Solver | 0.00% | 6.6m | 0.00% | 6.6m | 0.00% | 6.6m | 0.00% | 6.6m | 0.00% | 6.6m |
| URS | 29.80% | 6.7m | 9.61% | 6.9m | 4.63% | 7.6m | -1.47% | 4.3s | 4.93% | 4.3s |

| Method | APDCVRP | | AOPDCVRP | | – | | – | | – | |
|---|---|---|---|---|---|---|---|---|---|---|
| | Gap | Time | Gap | Time | | | | | | |
| Oracle Solver | 0.00% | 6.6m | 0.00% | 6.6m | – | | – | | – | |
| URS | 7.03% | 1.2m | 11.13% | 1.2m | – | | – | | – | |

# J  MORE EXPERIMENTS

## J.1  TRAINING PROBLEM SELECTION

As shown in Table 6, our selection of the 11 training problems is guided by a "minimum redundancy" principle: we ensure that most attributes in UDR appeared in at least two distinct problem types during training, which helps prevent the model from overfitting a specific attribute to a single problem. The only exception is the Penalty attribute, as preliminary experiments have shown that exposure in a single problem type is sufficient for generalization to variants such as PCTSP and SPCTSP.

To empirically validate this selection strategy, we train two additional variants and compare them with our original URS: URS-Diff (Different Selection) and URS-Expand (Broader Problems). The first variant, URS-Diff, uses a different problem set where OCVRPTW is replaced by attribute-independent CVRPL. This modification means that the Open Route and Time Window attributes each appear in only one training problem. The second variant, URS-Expand, is trained on a broader set of 22 problems, incorporating all variants evaluated in MTPOMO (Liu et al., 2024a) and MVMoE (Zhou et al., 2024b).

As shown in Table 14, URS-Diff performs significantly worse on unseen problems involving Open Route and Time Window constraints. Meanwhile, URS-Expand achieves a slightly better performance than URS due to seeing more problem variants during training. However, the original URS remains highly competitive. Remarkably, despite being trained on only half the number of problem types (11 vs. 22), URS achieves the best solution on 12 out of the 22 evaluated problems.

These results demonstrate that the choice of our selected 11 problems offers a balanced trade-off, providing sufficient attribute coverage for strong generalization while maintaining training efficiency. Expanding the training set yields only marginal gains, while a poorly selected set leads to poor cross-problem generalization. In future work, we plan to explore more effective training problem selection strategies to further enhance model performance and training efficiency.

Table 14: Comparison under different training problem settings. The symbol * indicates the original problem selection, and $\Delta$ represents a different problem selection of equal number. All test problems are encompassed within URS-Expand.

| Method | ATSP (* $\Delta$) | TSP(* $\Delta$) | OP(* $\Delta$) | PCTSP(* $\Delta$) | PDTSP(* $\Delta$) | ACVRP(* $\Delta$) |
|---|---|---|---|---|---|---|
| URS-Diff | 4.52% | 1.05% | 0.82% | 1.31% | 6.09% | 5.10% |
| URS-Expand | 4.77% | 0.93% | 0.99% | 1.41% | 5.85% | 5.48% |
| URS | **2.26%** | **0.57%** | **0.45%** | **1.06%** | **4.98%** | **3.06%** |

| | CVRP(* $\Delta$) | CVRPTW(* $\Delta$) | CVRPB(* $\Delta$) | OCVRP(* $\Delta$) | OCVRPTW(*) | CVRPL($\Delta$) |
|---|---|---|---|---|---|---|
| URS-Diff | 1.97% | 6.26% | 1.80% | 3.75% | 15.45% | 0.58% |
| URS-Expand | 2.04% | **5.62%** | 1.77% | 3.45% | **4.34%** | 0.65% |
| URS | **1.81%** | 6.13% | **1.46%** | **3.24%** | 5.07% | **0.43%** |

| | OCVRPB | CVRPBL | CVRPLTW | OCVRPBTW | CVRPBLTW | OCVRPL |
|---|---|---|---|---|---|---|
| URS-Diff | 7.55% | 2.02% | 2.76% | 27.48% | 10.25% | 3.73% |
| URS-Expand | **5.39%** | 1.88% | **2.03%** | **9.13%** | **6.29%** | 3.41% |
| URS | 9.35% | **1.65%** | 2.67% | 13.77% | 9.18% | **3.22%** |

| | CVRPBTW | OCVRPBL | OCVRPLTW | OCVRPBLTW | Avg.gap | Best Sol. |
|---|---|---|---|---|---|---|
| URS-Diff | 10.00% | 7.56% | 15.61% | 27.80% | 7.43% | 0/22 |
| URS-Expand | **6.07%** | **5.47%** | **4.38%** | **9.23%** | **4.12%** | 10/22 |
| URS | 8.94% | 9.47% | 5.12% | 13.72% | 4.89% | **12/22** |

## J.2  RESULTS ON DIFFERENT EVALUATION SCENARIOS

Verifying performance under different evaluation scenarios is important for testing a model. To address this, we conducted a systematic evaluation using the exact scenarios from Sym-NCO (Kim et al., 2022). We utilize the test datasets directly from the official Sym-NCO GitHub repository, which comprises 10,000 instances for each problem variant (i.e., TSP, CVRP, PCTSP, and OP). Following

Table 15: Performance evaluation for different problems under different evaluation scenarios.

| Method | TSP100 | | CVRP100 | | OP100 | | PCTSP100 | |
|---|---|---|---|---|---|---|---|---|
| | Gap | Time | Gap | Time | Gap | Time | Gap | Time |
| *The evaluation scenario of URS, the number of instances is 1000.* | | | | | | | | |
| *Multi-start rollout and instance augmentation×8 are both applied.* | | | | | | | | |
| Oracle Solver | 0.00% | 6m | 0.00% | 9.1m | 0.00% | 1.5m | 0.00% | 1.2h |
| Sym-NCO | 0.14% | 5s | **1.46%** | 6s | 0.68% | 15s | 0.14% | 16s |
| URS-STL | **0.08%** | 6s | 1.63% | 6s | **0.11%** | 4s | **-0.09%** | 5s |
| URS | 0.57% | 6s | 1.81% | 6s | 0.45% | 4s | 1.06% | 5s |
| *In the evaluation scenario of Sym-NCO, the number of instances is 10000.* | | | | | | | | |
| *Only multi-start rollout is applied.* | | | | | | | | |
| Oracle Solver | 0.00% | 21m | 0.00% | 13h | 0.00% | 15m | 0.00% | 12h |
| Sym-NCO | 0.39% | 8s | **1.46%** | 10s | 0.45% | 2.6m | -0.02% | 2.7m |
| URS-STL | **0.30%** | 7s | 1.80% | 8s | **-0.02%** | 10.6s | **-0.06%** | 11.7s |
| URS | 1.24% | 7s | 2.03% | 8s | 0.50% | 10.6s | 1.30% | 11.7s |

Table 16: Performance comparison with POMO and Sym-NCO in real-world instances in TSPLIB with $50 < N < 250$.

| Instance | Opt. | POMO aug×20 | | Sym-NCO aug×20 | | URS-STL aug×8 | | URS aug×8 | |
|---|---|---|---|---|---|---|---|---|---|
| | | Obj. | Gap | Obj. | Gap | Obj. | Gap | Obj. | Gap |
| eil51 | 426 | 429 | 0.82% | 432 | 1.39% | 430 | 0.94% | 427 | **0.23%** |
| berlin52 | 7,542 | 7,545 | 0.04% | 7,544 | 0.03% | 7,542 | **0.00%** | 7,543 | 0.01% |
| st70 | 675 | 677 | 0.31% | 677 | 0.31% | 675 | **0.00%** | 677 | 0.30% |
| pr76 | 108,159 | 108,681 | 0.48% | 108,388 | 0.21% | 108,159 | **0.00%** | 109,089 | 0.86% |
| eil76 | 538 | 544 | 1.18% | 544 | 1.18% | 538 | **0.00%** | 543 | 0.93% |
| rat99 | 1,211 | 1,270 | 4.90% | 1,261 | 4.17% | 1,212 | **0.08%** | 1,252 | 3.39% |
| rd100 | 7,910 | 7,912 | 0.03% | 7,911 | 0.02% | 7,910 | **0.00%** | 7,927 | 0.21% |
| KroA100 | 21,282 | 21,486 | 0.96% | 21,397 | 0.54% | 21,282 | **0.11%** | 21,572 | 1.36% |
| KroB100 | 22,141 | 22,285 | 0.65% | 22,378 | 1.07% | 22,146 | **0.02%** | 22,667 | 2.38% |
| KroC100 | 20,749 | 20,755 | 0.03% | 20,930 | 0.87% | 20,749 | **0.00%** | 20,772 | 0.11% |
| KroD100 | 21,294 | 21,488 | 0.91% | 21,696 | 1.89% | 21,294 | **0.00%** | 21,704 | 1.93% |
| KroE100 | 22,068 | 22,196 | 0.58% | 22,313 | 1.11% | 22,121 | **0.24%** | 22,192 | 0.56% |
| eil101 | 629 | 641 | 1.84% | 641 | 1.84% | 630 | **0.16%** | 640 | 1.75% |
| lin105 | 14,379 | 14,690 | 2.16% | 14,358 | 0.54% | 14,379 | **0.00%** | 14,754 | 2.61% |
| pr124 | 59,030 | 59,353 | 0.55% | 59,202 | 0.29% | 59,030 | **0.00%** | 59,942 | 1.54% |
| bier127 | 118,282 | 125,331 | 5.96% | 122,664 | 3.70% | 119,975 | **1.43%** | 122,941 | 3.94% |
| ch130 | 6,110 | 6,112 | **0.03%** | 6,118 | 0.14% | 6,116 | 0.10% | 6,184 | 1.21% |
| pr136 | 96,772 | 97,481 | 0.73% | 97,579 | 0.83% | 97,447 | **0.70%** | 98,197 | 1.47% |
| pr144 | 58,537 | 59,197 | 1.13% | 58,930 | 0.67% | 58,673 | **0.23%** | 58,981 | 0.76% |
| kroA150 | 26,524 | 26,833 | 1.16% | 26,865 | 1.28% | 26,591 | **0.25%** | 27,368 | 3.18% |
| kroB150 | 26,130 | 26,596 | 1.78% | 26,648 | 1.98% | 26,237 | **0.41%** | 26,900 | 2.95% |
| pr152 | 73,682 | 74,372 | 0.94% | 75,292 | 2.18% | 74,056 | **0.51%** | 75,453 | 2.40% |
| u159 | 42,080 | 42,567 | 1.16% | 42,602 | 1.24% | 42,427 | **0.82%** | 42,775 | 1.65% |
| rat195 | 2,323 | 2,546 | 9.58% | 2,502 | 7.70% | 2,379 | **2.41%** | 2,420 | 4.18% |
| kroA200 | 29,368 | 29,937 | 1.94% | 29,816 | 1.53% | 29,677 | **1.05%** | 30,419 | 3.58% |
| ts225 | 126,643 | 131,811 | 4.08% | 127,742 | **0.87%** | 128,522 | 1.48% | 129,233 | 2.05% |
| tsp225 | 3,919 | 4,149 | 5.87% | 4,126 | 5.27% | 4,000 | **2.15%** | 4,058 | 3.63% |
| pr226 | 80,369 | 82,428 | 2.56% | 82,337 | 2.45% | 81,707 | **1.66%** | 82,049 | 2.09% |
| Avg.Gap | | 1.87% | | 1.62% | | **0.52%** | | 1.83% | |

Sym-NCO, we employ an N-start rollout without applying instance augmentation for TSP and CVRP. For PCTSP and OP, the AM-based Sym-NCO employs a 200-sample approach. To ensure a fair comparison, we apply an additional coordinate transformation $(1 - x, 1 - y)$ to URS to match this sampling budget. Like Sym-NCO, we also tested the solvers on small-scale TSPLib data ($50 < N < 250$).

The comparison results under Sym-NCO's evaluation settings are presented in Table 15 and Table 16. According to the results, URS delivers impressive performance that is comparable to or even surpasses Sym-NCO, under Sym-NCO's own evaluation scenarios. This confirms that the performance of URS is robust and not dependent on a specific evaluation setting. Unlike Sym-NCO, which is a specialized solver trained individually for each problem using distinct models (POMO for TSP and CVRP; AM for PCTSP and OP) and specific hyperparameter tuning, URS achieves competitive results across a wide range of problems with a single set of parameters, highlighting its strong generalization capability.

### J.3 RESULTS ON BENCHMARK DATASET

Our test results on Benchmark datasets are presented in Table 17 and Table 18. All models are trained on instances of size $N = 100$.

In Set-X dataset, some results for comparative models are sourced from MVMoE (Zhou et al., 2024b) and RouteFinder (Berto et al., 2024). As shown in Table 17, URS maintains its position as the best overall performance across instances of varying scales. Notably, RL-based URS surpasses the SL-based representative specialized model LEHD (8.678% vs. 12.836%), underscoring its practical applicability in real-world scenarios.

For CVRPLib XXL benchmark. The results in Table 18 clearly demonstrate that URS can significantly outperform existing multi-task neural routing solvers, especially on large-scale instances. Notably, URS also greatly surpasses many representative specialized models (e.g., LEHD, BQ, and ELG). These results further underscore the practical applicability of URS in real-world scenarios.

Table 17: Results on large-scale CVRPLIB instances (Set-X) (Uchoa et al., 2017) ($500 \leq N \leq 1000$). All models are trained on instances of size $N = 100$.

| Set-X Instance | Opt. | POMO Obj. | POMO Gap | LEHD Obj. | LEHD Gap | MTPOMO Obj. | MTPOMO Gap | MVMoE/4E Obj. | MVMoE/4E Gap | MVMoE/4E-L Obj. | MVMoE/4E-L Gap | RF-MVMoE Obj. | RF-MVMoE Gap | RF-TE Obj. | RF-TE Gap | URS Obj. | URS Gap |
|---|---|---|---|---|---|---|---|---|---|---|---|---|---|---|---|---|---|
| X-n502-k39 | 69226 | 75617 | 9.232% | 71438 | 3.195% | 77284 | 11.640% | 73533 | 6.222% | 74429 | 7.516% | 76338 | 10.274% | 71791 | 3.705% | 71281 | 2.969% |
| X-n513-k21 | 24201 | 30518 | 26.102% | 25624 | 5.880% | 28510 | 17.805% | 32102 | 32.647% | 31231 | 29.048% | 32639 | 34.866% | 28465 | 17.619% | 26166 | 8.119% |
| X-n524-k153 | 154593 | 201877 | 30.586% | 280556 | 81.480% | 192249 | 24.358% | 186540 | 20.665% | 182392 | 17.982% | 170999 | 10.612% | 174381 | 12.800% | 175250 | 13.362% |
| X-n536-k96 | 94846 | 106073 | 11.837% | 103785 | 9.425% | 106514 | 12.302% | 109581 | 15.536% | 108543 | 14.441% | 105847 | 11.599% | 103272 | 8.884% | 102969 | 8.564% |
| X-n548-k50 | 86700 | 103093 | 18.908% | 90644 | 4.549% | 94562 | 9.068% | 95894 | 10.604% | 95917 | 10.631% | 104289 | 20.287% | 100956 | 16.443% | 89768 | 3.539% |
| X-n561-k42 | 42717 | 49370 | 15.575% | 44728 | 4.708% | 47846 | 12.007% | 56008 | 31.114% | 51810 | 21.287% | 53383 | 24.969% | 49454 | 15.771% | 45964 | 7.601% |
| X-n573-k30 | 50673 | 83545 | 64.871% | 53482 | 5.543% | 60913 | 20.208% | 59473 | 17.366% | 57042 | 12.569% | 61524 | 21.414% | 55952 | 10.418% | 54361 | 7.278% |
| X-n586-k159 | 190316 | 229887 | 20.792% | 232867 | 22.358% | 208893 | 9.761% | 215668 | 13.321% | 214577 | 12.748% | 212151 | 11.473% | 205575 | 8.018% | 202645 | 6.478% |
| X-n599-k92 | 108451 | 150572 | 38.839% | 115377 | 6.386% | 120333 | 10.956% | 128949 | 18.901% | 125279 | 15.517% | 126578 | 16.714% | 116560 | 7.477% | 114423 | 5.507% |
| X-n613-k62 | 59535 | 68451 | 14.976% | 62484 | 4.953% | 67984 | 14.192% | 82586 | 38.718% | 74945 | 25.884% | 73456 | 23.383% | 67267 | 12.987% | 65901 | 10.693% |
| X-n627-k43 | 62164 | 84434 | 35.825% | 67568 | 8.693% | 73060 | 17.528% | 70987 | 14.193% | 70905 | 14.061% | 70414 | 13.271% | 67572 | 8.700% | 66499 | 6.973% |
| X-n641-k35 | 63682 | 75573 | 18.672% | 68249 | 7.172% | 72643 | 14.071% | 75329 | 18.289% | 72655 | 14.090% | 71975 | 13.023% | 70831 | 11.226% | 67005 | 5.218% |
| X-n655-k131 | 106780 | 127211 | 19.134% | 117532 | 10.069% | 116988 | 9.560% | 117678 | 10.206% | 118475 | 10.952% | 119057 | 11.497% | 112202 | 5.078% | 110237 | 3.237% |
| X-n670-k130 | 146332 | 208079 | 42.197% | 220927 | 50.977% | 190118 | 29.922% | 197695 | 35.100% | 183447 | 25.364% | 168226 | 14.962% | 168999 | 15.490% | 184010 | 25.748% |
| X-n685-k75 | 68205 | 79482 | 16.534% | 72946 | 6.951% | 80892 | 18.601% | 97388 | 42.787% | 89441 | 31.136% | 82269 | 20.620% | 77847 | 14.137% | 75942 | 11.344% |
| X-n701-k44 | 81923 | 97843 | 19.433% | 86327 | 5.376% | 92075 | 12.392% | 98469 | 20.197% | 94924 | 15.870% | 90189 | 10.090% | 89932 | 9.776% | 86038 | 5.023% |
| X-n716-k35 | 43373 | 51381 | 18.463% | 46502 | 7.214% | 52709 | 21.525% | 56773 | 30.895% | 52305 | 20.593% | 52250 | 20.467% | 49669 | 14.516% | 46496 | 7.200% |
| X-n733-k159 | 136187 | 159098 | 16.823% | 149115 | 9.493% | 161961 | 18.925% | 178322 | 30.939% | 167477 | 22.976% | 156387 | 14.833% | 148463 | 9.014% | 147743 | 8.485% |
| X-n749-k98 | 77269 | 87786 | 13.611% | 83439 | 7.985% | 90582 | 17.229% | 100438 | 29.985% | 94497 | 22.296% | 92147 | 19.255% | 85171 | 10.227% | 83759 | 8.399% |
| X-n766-k71 | 114417 | 135464 | 18.395% | 131487 | 14.919% | 144041 | 25.891% | 152352 | 33.155% | 136255 | 19.086% | 130505 | 14.061% | 129935 | 13.563% | 139371 | 21.810% |
| X-n783-k48 | 72386 | 90289 | 24.733% | 76766 | 6.051% | 83169 | 14.897% | 100383 | 38.677% | 92960 | 28.423% | 96336 | 33.087% | 83185 | 14.919% | 77437 | 6.978% |
| X-n801-k40 | 73305 | 124278 | 69.536% | 77546 | 5.785% | 85077 | 16.059% | 91560 | 24.903% | 87662 | 19.585% | 87118 | 18.843% | 86164 | 17.542% | 77369 | 5.544% |
| X-n819-k171 | 158121 | 193451 | 22.344% | 178558 | 12.925% | 177157 | 12.039% | 183599 | 16.113% | 185832 | 17.525% | 179596 | 13.581% | 174441 | 10.321% | 171024 | 8.160% |
| X-n837-k142 | 193737 | 237884 | 22.787% | 207709 | 7.212% | 214207 | 10.566% | 229526 | 18.473% | 221286 | 14.220% | 230362 | 18.904% | 208528 | 7.635% | 203457 | 5.017% |
| X-n856-k95 | 88965 | 152528 | 71.447% | 92936 | 4.464% | 101774 | 14.398% | 99129 | 11.425% | 106816 | 20.065% | 105801 | 18.924% | 98291 | 10.483% | 94547 | 6.274% |
| X-n876-k59 | 99299 | 119764 | 20.609% | 104183 | 4.918% | 116617 | 17.440% | 119619 | 20.463% | 114333 | 15.140% | 114016 | 14.821% | 107416 | 8.174% | 105417 | 6.161% |
| X-n895-k37 | 53860 | 70245 | 30.421% | 58028 | 7.739% | 65587 | 21.773% | 79018 | 46.710% | 64310 | 19.402% | 69099 | 28.294% | 64871 | 20.444% | 58137 | 7.941% |
| X-n916-k207 | 329179 | 399372 | 21.324% | 385208 | 17.021% | 361719 | 9.885% | 383681 | 16.557% | 374016 | 13.621% | 373600 | 13.494% | 352998 | 7.236% | 346556 | 5.279% |
| X-n936-k151 | 132715 | 237625 | 79.049% | 196547 | 48.097% | 186262 | 40.347% | 220926 | 66.466% | 190407 | 43.471% | 161343 | 21.571% | 163162 | 22.942% | 172675 | 30.110% |
| X-n957-k87 | 85465 | 130850 | 53.104% | 90295 | 5.651% | 98198 | 14.898% | 113882 | 33.250% | 105629 | 23.593% | 123633 | 44.659% | 102689 | 20.153% | 90485 | 5.874% |
| X-n979-k58 | 118976 | 147687 | 24.132% | 127972 | 7.561% | 138092 | 16.067% | 146347 | 23.005% | 139682 | 17.404% | 131754 | 10.740% | 129952 | 9.225% | 125353 | 5.360% |
| X-n1001-k43 | 72355 | 100399 | 38.759% | 76689 | 5.990% | 87660 | 21.153% | 114441 | 58.176% | 94734 | 30.929% | 88969 | 22.962% | 85929 | 18.760% | 77739 | 7.441% |
| Avg. Gap | | | 29.658% | | 12.836% | | 16.796% | | 26.408% | | 19.607% | | 18.795% | | 12.303% | | 8.678% |

Table 18: Comparison on CVRPLib Set-XXL with scale $\in [3000, 7000]$. All models are trained on instances of size N = 100.

| Method | Type | Leuven1 (N=3000) | Leuven2 (N=4000) | Antwerp1 (N=6000) | Antwerp2 (N=7000) | Total N $\in$ [3000,7000] |
|---|---|---|---|---|---|---|
| LEHD greedy | Specialized | 16.60% | 34.86% | 14.66% | 22.77% | 22.22% |
| BQ greedy | Specialized | 18.53% | 30.70% | 16.48% | 27.67% | 23.34% |
| POMO aug×8 | Specialized | 460.32% | 202.17% | OOM | OOM | – |
| Sym-NCO aug×8 | Specialized | 196.82% | 328.17% | OOM | OOM | – |
| ELG aug×8 | Specialized | 12.12% | 21.52% | OOM | OOM | – |
| MTPOMO aug×8 | General | 67.72% | 87.31% | OOM | OOM | – |
| MVMoE/4E aug×8 | General | 299.10% | 170.10% | OOM | OOM | – |
| MVMoE/4E-L aug×8 | General | 182.28% | 127.07% | OOM | OOM | – |
| RF-MVMoE aug×8 | General | 57.30% | 160.27% | OOM | OOM | – |
| RF-TF aug×8 | General | 26.90% | 45.56% | OOM | OOM | – |
| CaDA aug×8 | General | 1035.51% | OOM | OOM | OOM | – |
| ReLD-MTL aug×8 | General | 17.00% | 30.59% | OOM | OOM | – |
| GOAL greedy | General | OOM | OOM | OOM | OOM | – |
| URS aug×8 | General | **11.57%** | **17.80%** | **9.23%** | **14.95%** | **13.39%** |

### J.4 SCALABILITY TO LARGER-SCALE REALISTIC PROBLEMS

Due to the limited benchmark datasets currently available for VRP variants, we have conducted further experiments on large-scale synthetic problem instances to validate the large-scale cross-problem generalization ability of URS. We generate a new large-scale (1K-5K nodes) synthetic dataset for a diverse set of problems (CVRPTW, CVRPB, and OCVRPTW). Each dataset contains 16 instances. For all instances, we adhere to the capacity constraints specified in TAM (Hou et al., 2022). To ensure a fair comparison, HGS-PyVRP is executed with multi-core (16) parallelism enabled, which allows it to process all instances of a specific size concurrently. According to the results, our proposed URS consistently achieves the best solution quality, complemented by remarkably fast inference times, across various problem instances among all comparable neural solvers. The advantages of URS over existing multi-task neural solvers are significant. Remarkably, URS already outperforms the strong HGS-PyVRP with a carefully manual design on large-scale CVRPB instances. These extensive new results demonstrate URS's strong scalability and generalization ability to large-scale and realistic problems.

Table 19: Generalization performance on large-scale instances of different VRP variants. All models are trained on instances of size N = 100 and employ instance augmentation $\times$ 8. Each dataset includes 16 instances. Note that "Time" is the total running time, and HGS-PyVRP is executed with multi-core (16) parallelism enabled.

| Problem | Method | N = 1000 Obj.(Gap) | Time | N = 2000 Obj.(Gap) | Time | N = 3000 Obj.(Gap) | Time | N = 4000 Obj.(Gap) | Time | N = 5000 Obj.(Gap) | Time |
|---|---|---|---|---|---|---|---|---|---|---|---|
| CVRPTW | HGS-PyVRP | 159.35(0.00%) | 20m | 337.15(0.00%) | 40m | 516.97(0.00%) | 1h | 618.58(0.00%) | 2.7h | 787.25(0.00%) | 3.3h |
| | MTPOMO | 219.22(37.57%) | 1.7m | 488.40(44.86%) | 14.6m | 773.00(49.53%) | 49.5m | 975.61(57.72%) | 1.9h | OOM | |
| | MVMoE | 233.53(46.55%) | 1.7m | 624.73(85.30%) | 14.6m | 1094.25(111.67%) | 51.9m | 1486.34(140.28%) | 2h | OOM | |
| | ReLD-MTL | 183.31(15.036%) | 1.6m | 396.28(17.54%) | 13.4m | 613.58(18.69%) | 45.7m | 748.65(21.03%) | 1.8h | OOM | |
| | URS-MTL | **172.58(8.30%)** | **1.3m** | **365.27(8.34%)** | **11.1m** | **559.03(8.14%)** | **30.1m** | **673.63(8.90%)** | **1.2h** | **853.79(8.45%)** | **2.3h** |
| CVRPB | HGS-PyVRP | 36.52(0.00%) | 20m | 51.80(0.00%) | 40m | 75.12(0.00%) | 1h | 93.52(0.00%) | 2.7h | 112.21(0.00%) | 3.3h |
| | MTPOMO | 45.92(25.77%) | 1.1m | 83.50(61.20%) | 8.8m | 136.52(81.73%) | 31.4m | 197.99(111.70%) | 1.2h | OOM | |
| | MVMoE | 98.60(170.03%) | 1m | 250.47(383.57%) | 8.1m | 332.93(343.19%) | 28.9m | 423.19(352.49%) | 1.1h | OOM | |
| | ReLD-MTL | 38.52(5.48%) | 1m | 59.21(14.30%) | 8m | 88.36(17.62%) | 28.3m | 114.88(22.83%) | 1.1h | OOM | |
| | URS-MTL | **34.86(-4.54%)** | **40s** | **50.39(-2.72%)** | **5.1m** | **72.01(-4.15%)** | **17.3m** | **91.74(-1.91%)** | **41.2m** | **111.93(-0.24%)** | **1.3h** |
| OCVRPTW | HGS-PyVRP | 90.91(0.00%) | 20m | 164.00(0.00%) | 40m | 224.16(0.00%) | 1h | 299.45(0.00%) | 2.7h | 367.86(0.00%) | 3.3h |
| | MTPOMO | 147.14(61.85%) | 1.6m | 297.53(81.43%) | 12.8m | 437.73(95.28%) | 44.3m | 601.63(100.91%) | 1.7h | OOM | |
| | MVMoE | 147.35(62.09%) | 1.6m | 389.60(137.57%) | 12.9m | 634.11(182.88%) | 44.7m | 899.18(200.28%) | 1.7h | OOM | |
| | ReLD-MTL | 110.00(21.01%) | 1.5m | 208.60(27.20%) | 12.3m | 292.70(30.57%) | 42m | 394.42(31.72%) | 1.6h | OOM | |
| | URS-MTL | **98.76(8.63%)** | **1m** | **178.07(8.58%)** | **7.6m** | **243.88(8.80%)** | **25.4m** | **325.64(8.74%)** | **1h** | **398.72(8.39%)** | **1.9h** |

# K  ABLATION STUDY

In this section, we conduct a detailed ablation study and analysis to demonstrate the effectiveness and robustness of URS. Please note that, unless stated otherwise, the results presented in the ablation study reflect the best result from multiple trajectories, and we employ instance augmentation to improve performance in the ablation study. We adopt the widely used 100-node setting as our primary evaluation scenario for URS.

## K.1  EFFECTS OF NODE-TYPE INDICATOR

Table 20: The ablation study of the node-type indicator on trained VRPs

| Method | ATSP | TSP | OP | PCTSP | PDTSP | ACVRP | CVRP | CVRPTW | CVRPB | OCVRP | OCVRPTW | Avg.gap |
|---|---|---|---|---|---|---|---|---|---|---|---|---|
| w/o $\xi_i$ | **2.14%** | **0.48%** | 0.45% | 1.06% | 9.28% | **2.84%** | 3.66% | 6.61% | 1.57% | 7.87% | 5.66% | 3.78% |
| w/ $\xi_i$ | 2.26% | 0.57% | **0.45%** | **1.06%** | **4.98%** | 3.06% | **1.81%** | **6.13%** | **1.46%** | **3.24%** | **5.07%** | **2.74%** |

To evaluate the effectiveness of node-type indicator $\xi$, we conduct an ablation study about including and excluding $\xi$. Table 20 shows that adding the node-type indicator reduces the average optimality gap from 3.78% to 2.74%, with large gains on the routing problem with relatively rich instance node roles, such as PDTSP, where the coexistence of a depot and paired pickup–delivery nodes (with inherent precedence coupling) allows the explicit node-role encoding to further boost performance. Meanwhile, URS maintains a competitive performance in addressing the problem of single-node roles.

## K.2  EFFECTS OF PROBLEM STATE

Table 21: Performance impact of removing the problem state feature $C_t$

| Method | ATSP | TSP | OP | PCTSP | PDTSP | ACVRP | CVRP | CVRPTW | CVRPB | OCVRP | OCVRPTW | Avg.gap |
|---|---|---|---|---|---|---|---|---|---|---|---|---|
| w/o $C_t$ | **2.09%** | **0.45%** | 0.60% | 1.20% | **4.59%** | 4.03% | 2.70% | **5.71%** | 2.64% | 4.02% | **4.62%** | 2.97% |
| w/ $C_t$ | 2.26% | 0.57% | **0.45%** | **1.06%** | 4.98% | **3.06%** | **1.81%** | 6.13% | **1.46%** | **3.24%** | 5.07% | **2.74%** |

As described in Appendix E.3, the majority of VRP variants are already solved without supplying an additional problem state $C_t$. To test whether the remaining use of $C_t$ matters, we conduct an ablation in which we eliminate it entirely, enforcing $C_t \equiv 0$ for all problems. The results in Table 21 show that URS maintains its competitive performance even in the absence of $C_t$. This indicates that explicit constraint-state features are not essential for the effectiveness of URS. Interestingly, on (O)CVRPTW instances, the model without any explicit constraint features outperforms the version that supplies only capacity information. We guess that capacity and time windows information are equally important, so enforcing only capacity creates a bias toward load considerations.

## K.3  EFFECTS OF MIXED BIAS MODULE

Table 22: Ablation of prior components in the Mixed Bias Module (MBM).

| $D$ | $D^{\mathbf{T}}$ | $R$ | ATSP | TSP | OP | PCTSP | PDTSP | ACVRP | CVRP | CVRPTW | CVRPB | OCVRP | OCVRPTW | Avg.gap |
|---|---|---|---|---|---|---|---|---|---|---|---|---|---|---|
| ✓ | × | × | 16.64% | **0.44%** | **0.25%** | **0.67%** | 7.33% | 16.62% | 2.42% | 6.44% | 3.96% | 9.44% | 5.47% | 6.33% |
| ✓ | ✓ | × | **2.09%** | 0.45% | 0.48% | 0.82% | 7.66% | 9.11% | 2.52% | 6.89% | 2.68% | 5.70% | 6.36% | 4.07% |
| ✓ | × | ✓ | 16.60% | 0.46% | 0.49% | 0.94% | **4.49%** | 26.22% | 4.16% | 6.57% | 3.07% | 12.23% | 5.53% | 7.34% |
| ✓ | ✓ | ✓ | 2.26% | 0.57% | 0.45% | 1.06% | 4.98% | **3.06%** | **1.81%** | **6.13%** | **1.46%** | **3.24%** | **5.07%** | **2.74%** |

To further assess the effectiveness of MBM, we conduct an ablation study that incorporates different combinations of problem-specific priors into the MBM. Owing to its flexibility, the MBM permits imposing multiple priors on the same shared attention layer. The results in Appendix K.3 underscore the complementary roles of the three prior components: (1) The transposed distance matrix $D^{\mathbf{T}}$ captures directional asymmetry absent from the raw distance matrix, and its inclusion markedly stabilizes optimization on asymmetric variants; (2) Removing the pickup–delivery relation matrix $R$ degrades performance on PDTSPs, confirming that explicit relational coupling is indispensable

for these instances; (3) When all priors are fused, MBM can simultaneously learn the geometric and relational biases inherent in various problems, yielding the best overall performance.

## K.4 Effects of Different Decoder Architectures

Table 23: Effect of different decoder architectures on URS cross-problem performance

| Decoder | ATSP | TSP | OP | PCTSP | PDTSP | ACVRP | CVRP | CVRPTW | CVRPB | OCVRP | OCVRPTW | Avg.gap |
|---------|------|-----|-----|-------|-------|-------|------|--------|-------|-------|---------|---------|
| URS-POMO | 4.83% | 0.74% | 0.72% | 1.29% | 5.45% | 5.09% | 1.99% | 6.42% | 1.83% | 3.62% | 5.31% | 3.39% |
| URS-ReLD | 6.48% | 0.90% | 0.86% | 1.51% | 5.81% | 6.51% | 2.33% | 6.56% | 2.33% | 3.62% | 5.64% | 3.87% |
| URS | 2.26% | 0.57% | 0.45% | 1.06% | 4.98% | 3.06% | 1.81% | 6.13% | 1.46% | 3.24% | 5.07% | 2.74% |

We adopt an AM (Kool et al., 2019) as our basic encoder-decoder model. Multiple decoder refinements have been proposed around AM (Kwon et al., 2020; Huang et al., 2025; Zhou et al., 2024a). To assess how decoder structure impacts multi-task performance, we additionally train two URS variants whose decoders follow POMO and ReLD, denoted URS-POMO and URS-ReLD. For stronger and fairer baselines, we apply Distance-aware Attention Reshaping (DAR) (Wang et al., 2025b) to both, and all decoder parameters are generated by a hypernetwork conditioned on the problem representation $\lambda$ (see Appendix E.2). As shown in Table 23, URS-POMO and URS-ReLD still exhibit strong cross-problem robustness, while replacing the decoder with ICAM (Zhou et al., 2024a) yields a further overall improvement.

## K.5 Effects of Different Components on Zero-shot Generalization

Table 24: Ablation of different components in the model architecture, which includes MBM, WEIGHT($\lambda$), and BIAS($\lambda$). The symbol (*) indicates the seen problems in training.

| MBM | WEIGHT($\lambda$) | BIAS($\lambda$) | ATSP (*) | TSP (*) | OP (*) | PCTSP (*) | PDTSP (*) | ACVRP (*) |
|-----|------------------|-----------------|----------|---------|--------|-----------|-----------|-----------|
| ✗ | ✓ | ✓ | 16.64% | 0.44% | 0.25% | 0.67% | 7.33% | 16.62% |
| ✓ | ✗ | ✓ | 3.24% | 1.10% | 1.04% | 1.42% | 6.38% | 4.07% |
| ✓ | ✓ | ✗ | 3.83% | 1.16% | 1.09% | 1.52% | 6.65% | 4.27% |
| ✓ | ✗ | ✗ | 4.79% | 1.58% | 1.39% | 1.92% | 7.05% | 5.39% |
| ✓ | ✓ | ✓ | 2.26% | 0.57% | 0.45% | 1.06% | 4.98% | 3.06% |

| MBM | WEIGHT($\lambda$) | BIAS($\lambda$) | CVRP (*) | CVRPTW (*) | CVRPB (*) | OCVRP (*) | OCVRPTW (*) | CVRPL |
|-----|------------------|-----------------|----------|------------|-----------|-----------|-------------|-------|
| ✗ | ✓ | ✓ | 2.42% | 6.44% | 3.96% | 9.44% | 5.47% | 0.98% |
| ✓ | ✗ | ✓ | 4.23% | 6.52% | 3.00% | 3.99% | 5.41% | 2.92% |
| ✓ | ✓ | ✗ | 9.94% | 6.18% | 5.00% | 4.61% | 5.21% | 4.79% |
| ✓ | ✗ | ✗ | 2.90% | 6.60% | 3.12% | 4.47% | 5.76% | 1.46% |
| ✓ | ✓ | ✓ | 1.81% | 6.13% | 1.46% | 3.24% | 5.07% | 0.43% |

| MBM | WEIGHT($\lambda$) | BIAS($\lambda$) | OCVRPB | CVRPBL | CVRPLTW | OCVRPBTW | CVRPBLTW | OCVRPL |
|-----|------------------|-----------------|--------|--------|---------|----------|----------|--------|
| ✗ | ✓ | ✓ | 15.95% | 4.18% | 2.89% | 18.78% | 10.93% | 9.51% |
| ✓ | ✗ | ✓ | 9.54% | 5.30% | 2.73% | 14.37% | 9.80% | 4.64% |
| ✓ | ✓ | ✗ | 9.58% | 3.25% | 3.07% | 14.16% | 9.91% | 3.88% |
| ✓ | ✗ | ✗ | 9.92% | 3.34% | 3.08% | 14.02% | 9.92% | 4.52% |
| ✓ | ✓ | ✓ | 9.35% | 1.65% | 2.67% | 13.77% | 9.18% | 3.22% |

| MBM | WEIGHT($\lambda$) | BIAS($\lambda$) | CVRPBTW | OCVRPBL | OCVRPLTW | OCVRPBLTW | Avg.gap | Best Sol. |
|-----|------------------|-----------------|---------|---------|----------|-----------|---------|-----------|
| ✗ | ✓ | ✓ | 10.74% | 16.14% | 5.57% | 18.80% | 8.37% | 3/22 |
| ✓ | ✗ | ✓ | 9.67% | 10.67% | 5.23% | 14.49% | 5.90% | 0/22 |
| ✓ | ✓ | ✗ | 9.63% | 10.42% | 5.51% | 13.99% | 6.26% | 0/22 |
| ✓ | ✗ | ✗ | 9.79% | 10.02% | 5.81% | 13.98% | 5.95% | 0/22 |
| ✓ | ✓ | ✓ | 8.94% | 9.47% | 5.12% | 13.72% | 3.88% | 19/22 |

To empirically evaluate the necessity of these components, we have conducted tests on the 11 seen problems as well as 11 additional unseen CVRP variants widely investigated in current multi-task research (e.g., MVMoE (Zhou et al., 2024b)). The results are presented in Table 24. We can observe that removing MBM leads to a significant performance drop on problems with complex geometric properties, such as ATSP, ACVRP, and PDTSP, which confirms MBM is critical for perceiving specific geometric biases. In addition, removing WEIGHT($\lambda$) or BIAS($\lambda$) also results in performance degradation across most tasks. In terms of overall performance, URS significantly outperforms any model where a single module is removed. The full URS model achieves the lowest average gap and

provides the best solution on the majority of tasks (19/22). This demonstrates that the synergy of these components is essential for robust cross-problem generalization.

## L LICENSES FOR USED RESOURCES

Table 25: List of licenses for the codes and datasets we used in this work.

| Resource | Type | Link | License |
|---|---|---|---|
| HGS-PyVRP (Wouda et al., 2024) | Code | https://github.com/PyVRP/PyVRP | MIT License |
| LKH3 (Helsgaun, 2017) | Code | http://webhotel4.ruc.dk/~keld/research/LKH-3/ | Available for academic research use |
| OR-Tools (Perron & Furnon, 2023) | Code | https://github.com/google/or-tools | Apache-2.0 License |
| POMO (Kwon et al., 2020) | Code | https://github.com/yd-kwon/POMO/tree/master/NEW_py_ver | MIT License |
| Sym-NCO (Kim et al., 2022) | Code | https://github.com/alstn12088/Sym-NCO | Available for any non-commercial use |
| LEHD (Luo et al., 2023) | Code | https://github.com/CIAM-Group/NCO_code/tree/main/single_objective/LEHD | Available for any non-commercial use |
| BQ (Drakulic et al., 2023) | Code | https://github.com/naver/bq-nco | CC BY-NC-SA 4.0 license |
| ICAM (Zhou et al., 2024a) | Code | https://github.com/CIAM-Group/ICAM | MIT License |
| MatNet (Kwon et al., 2021) | Code | https://github.com/yd-kwon/MatNet/tree/main | MIT License |
| Heter-AM (Li et al., 2021) | Code | https://github.com/jingwenli0312/Heterogeneous-Attentions-PDP-DRL | MIT License |
| MTPOMO (Liu et al., 2024a) | Code | https://github.com/FeiLiu36/MTNCO | MIT License |
| MVMoE (Zhou et al., 2024b) | Code | https://github.com/RoyalSkye/Routing-MVMoE | MIT License |
| RouteFinder (Berto et al., 2024) | Code | https://github.com/ai4co/routefinder | MIT License |
| CaDA (Li et al., 2025a) | Code | https://github.com/CIAM-Group/CaDA | MIT License |
| ReLD (Huang et al., 2025) | Code | https://github.com/ziweileonhuang/reld-nco | MIT License |
| GOAL (Drakulic et al., 2025) | Code | https://github.com/naver/goal-co | Available for any non-commercial use |
| CVRPLIB Set-X (Uchoa et al., 2017) | Dataset | http://vrp.galgos.inf.puc-rio.br/index.php/en/ | Available for academic research use |

We list the used existing codes and datasets in Table 25, and all of them are open-sourced resources for academic usage.

## M LLM USAGE

During the preparation of this manuscript, we utilized a Large Language Model (LLM) as a writing assistance tool. The primary purpose of its use was to improve the linguistic quality of the paper. Specifically, the LLM is used to polish the writing for enhanced clarity and fluency and to identify and correct grammatical, spelling, and punctuation errors.

We confirm that all core research ideas, study design, data analysis, and conclusions are independently developed by the authors. The LLM does not contribute to the research ideation, methodology, or substantive academic content of the paper. Its function was strictly confined to language editing and proofreading to ensure the manuscript's professionalism and readability.

