# OpenReview forum: "URS: A Unified Neural Routing Solver for Cross-Problem Zero-Shot Generalization"
_ICLR.cc/2026/Conference — Submitted to ICLR 2026_

### Official Review · Reviewer_t66G · 2025-10-26

**Soundness:** 3
**Presentation:** 2
**Contribution:** 2
**Rating:** 4
**Confidence:** 3

**Summary:**

This paper introduces URS, a novel unified RL routing approach that is able to find high-quality solutions for over 100 VRP variants, of which 90 are done through zero-shot generalization. URS is able to achieve this through four novel proposed components. The first is a unified data representation, whereby each routing problem shares a unified representation. Secondly, URS introduces a Mixed Bias Module (MBM), an attention mechanism that learns the geometric and relational priors, such as distance and the relation between nodes. Thirdly, they utilize a parameter generator that adjusts the weights and biases of the MBM for each specific routing problem. Lastly, URS uses an LLM that generates a masking function for each routing variant, whereby it checks the mask against a predefined checker function to see if it is correct.

**Strengths:**

• The unified representation is well motivated, and the ablation study in Table 4 shows the cost of removing attributes, such as problem state and node-type indicator.

• URS is able to handle over 100 VRP variants, whereby 96 variants are unseen through training, and URS is able to achieve this through zero-shot learning. The results also indicate that URS is either able to achieve similar or outperform existing baselines on the seen VRP variants, and outperforms all baselines on the unseen variants.

• While the main results are URS using the AM model, the ablation study shows that URS is also able to work fine with POMO or ReLD, which both report a worse gap; however, the difference is only around 1%. This signifies that URS is not strongly dependent on the model used.

**Weaknesses:**

•	The most glaring weakness of this paper is the LLM masking. The paper explains that an LLM is used to create a masking function, whereby it iterates through prompting by checking if the masking function satisfies a checker function. The main issue is that the checker function needs to be predefined, and without it, URS would not function. However, if you need to predifine the checker function, why not define the masking yourself? The LLM feels unnecessary to me and also raises questions about the zero-shot capabilities of URS, if the checker function is predefined. This limitation is also not mentioned in the limitations section.

•	Section 3.2 about the model architecture is difficult to understand, and it is unclear why MBM, the WEIGHT($\lambda$), and BIAS($\lambda$) are needed. Especially, WEIGHT($\lambda$) and BIAS($\lambda$) are unclear to me, given that the ablation study does not include them. The paper would improve if it included a better explanation of why they are included and how they improve the performance of URS.

**Questions:**

•	Would it be possible to do the masking completely automatically, whereby the checker function is also written by an LLM?

•	How important is the LLM masking for URS to function? Is it needed for each VRP variant or only some?

•	How many iterations are needed on average for the LLM to find a valid masking?

---

> ### Author Response · Authors · 2025-11-22
> **Response to Reviewer t66G [1/3]**
>
> Thank you very much for your time and effort in reviewing our work.  Below, we address each concern point by point. All revisions and additions made to the manuscript have been marked in blue to aid in your review.
>
> > **W1. LLM masking**
>
> Thank you for this critical comment, which highlights a crucial distinction in our methodology between the final solution checker and the stepwise mask generator. We sincerely appreciate this opportunity to offer the following clarification:
>
> - **Checker Function vs. Masking Function:** We fully agree with you that our proposed method requires a predefined checker function to verify whether the LLM-produced masking function can correctly construct feasible solutions. However, we want to emphasize the significant difference in their implementation difficulty. Example codes for both the checker and masking function for CVRPTW are provided **in our revised Appendix G**. A checker function only needs to validate a completed solution through static and post-hoc verification. In contrast, a mask function must correctly handle dynamic constraints at every step of the solution construction to guarantee feasibility, which requires proactive look-ahead reasoning to ensure that adding a node will not lead to an infeasible state later. Because writing a validation checker is relatively easy, we utilize it as part of our automated generate-and-validate pipeline to create the masking function.
> - **Why LLM?** According to the discussion above, manually designing a mask function requires considerable domain expertise for each specific constraint to ensure solution feasibility. While specialists can certainly implement such a masking function themselves, a primary objective in NCO is to reduce dependence on these handcrafted and problem-specific rules. Our LLM-driven approach automates this process, significantly lowering the barrier for non-expert users to apply our solver to new problem variants by simply providing a natural language description and a straightforward checker.
>
>
> - **Clarification on Zero-Shot Capabilities:** Finding the valid masking function is an **offline generate-and-validate process**, rather than relying on an LLM to perform online reasoning during the inference phase of the neural solver. We leverage automatic heuristic design frameworks (EoH [2]) to enable the LLM to iteratively refine the code of the masking function based on the checker's feedback. Any candidate masking function that produces even one infeasible solution in the test instances (i.e., validity rate < 100\%) is discarded. Once a valid masking function is obtained, it is used in the subsequent NCO process, and the offline process is terminated. **In other words, every final valid masking function produced by the LLM is reliable, and it will generate valid masks for each solution construction step of the NCO process.** The cost of the offline process is a one-time amortized cost per problem variant, as the generated masking function can be reused indefinitely for any instance of that variant. In our context, **the term "zero-shot" applies to the neural solver (URS), as its model parameters $\theta$ are trained once and then fixed, without the need for retraining or fine-tuning on new problems.**
>
> We hope these clarifications can fully address your concerns about the masking function and the zero-shot capabilities of URS. We will carefully incorporate the above discussion into the revised paper.

---

> ### Author Response · Authors · 2025-11-22
> **Response to Reviewer t66G [2/3]**
>
> > **W2. model architecture**
>
> Thank you for your valuable feedback regarding Section 3.2. We apologize for the lack of clarity in our original description and will revise this section to better explain the motivation and role of each component.
>
> - **Rationale for MBM:** The Mixed Bias Module (MBM) is introduced to address the limitations of standard attention mechanisms in capturing geometric and relational biases inherent in diverse routing problems. For example, effectively representing asymmetric distances in AVRPs or pairing relationships in PDPs remains challenging, which hinders the achievement of high-quality solutions across a wide range of problems. While prior work, such as MatNet[1], has proposed addressing this limitation by incorporating distance biases from different directions in separate parallel layers, this design leads to high model complexity and is only applicable to ATSP. In contrast, our MBM innovatively integrates multiple geometric priors (e.g., $D$, $D^T$, relation matrix $R$) within a single shared attention layer. This integrated design enhances flexibility while reducing architectural redundancy.
>
> - **Adaptive Parameter Generators:** Using a model with a single static set of parameters to solve over 100 diverse VRP variants is challenging and prone to underfitting. To mitigate this, we introduce a problem-adaptive mechanism consisting of two components: (1) BIAS($\lambda$) dynamically adjusts how the model integrates different geometric priors in the attention mechanism based on the problem representation $\lambda$;  and (2) WEIGHT($\lambda$) generates adaptive decoder parameters for each problem. Unlike existing adapter-based fine-tuning methods, URS adaptively generates the decoder parameters for each problem, thereby enabling the generation of high-quality solutions without requiring retraining or fine-tuning.
>
>
> - **Additional Ablation Study:** To empirically evaluate the necessity of these components, we have conducted tests on the 11 seen problems as well as 11 additional unseen CVRP variants widely investigated in current multi-task research (e.g., MVMoE[2]). The results are presented in **Table 1**. We can observe that removing MBM leads to a significant performance drop on problems with complex geometric properties, such as ATSP, ACVRP, and PDTSP, which confirms MBM is critical for perceiving specific geometric biases. In addition, removing WEIGHT($ \lambda$) or BIAS($ \lambda$) also results in performance degradation across most tasks. In terms of overall performance, URS significantly outperforms any model where a single module is removed. The full URS model achieves the lowest average gap and provides the best solution on the majority of tasks (19/22). This demonstrates that the synergy of these components is essential for robust cross-problem generalization.
>
> We will incorporate these clarifications and the new ablation results into the revised manuscript. Thank you again for your constructive comment.
>
> Table 1. Ablation of different components in the model architecture, which includes MBM, WEIGHT($\lambda$), and BIAS($\lambda$). The symbol (*) indicates the seen problems in training.
> |MBM|WEIGHT($\lambda$)|BIAS($\lambda$)|ATSP(*)|TSP(*)|OP(*)|PCTSP(*)|PDTSP(*)|ACVRP(*)|
> |:---:|:---:|:---:|:---:|:---:|:---:|:---:|:---:|:---:|
> |✗|✓|✓|16.64\%|**0.44\%**|**0.25\%**|**0.67\%**|7.33\%|16.62\%|
> |✓|✗|✓|3.24\%|1.10\%|1.04\%|1.42\%|6.38\%|4.07\%|
> |✓|✓|✗|3.83\%|1.16\%|1.09\%|1.52\%|6.65\%|4.27\%|
> |✓|✗|✗|4.79\%|1.58\%|1.39\%|1.92\%|7.05\%|5.39\%|
> |✓|✓|✓|**2.26\%**|0.57\%|0.45\%|1.06\%|**4.98\%**|**3.06\%**|
> |MBM|WEIGHT($\lambda$)|BIAS($\lambda$)|CVRP(*)|CVRPTW(*)|CVRPB(*)|OCVRP(*)|OCVRPTW(*)|CVRPL|
> |✗|✓|✓|2.42\%|6.44\%|3.96\%|9.44\%|5.47\%|0.98\%|
> |✓|✗|✓|4.23\%|6.52\%|3.00\%|3.99\%|5.41\%|2.92\%|
> |✓|✓|✗|9.94\%|6.18\%|5.00\%|4.61\%|5.21\%|4.79\%|
> |✓|✗|✗|2.90\%|6.60\%|3.12\%|4.47\%|5.76\%|1.46\%|
> |✓|✓|✓|**1.81\%**|**6.13\%**|**1.46\%**|**3.24\%**|**5.07\%**|**0.43\%**|
> |MBM|WEIGHT($\lambda$)|BIAS($\lambda$)|OCVRPB|CVRPBL|CVRPLTW|OCVRPBTW|CVRPBLTW|OCVRPL|
> |✗|✓|✓|15.95\%|4.18\%|2.89\%|18.78\%|10.93\%|9.51\%|
> |✓|✗|✓|9.54\%|5.30\%|2.73\%|14.37\%|9.80\%|4.64\%|
> |✓|✓|✗|9.58\%|3.25\%|3.07\%|14.16\%|9.91\%|3.88\%|
> |✓|✗|✗|9.92\%|3.34\%|3.08\%|14.02\%|9.92\%|4.52\%|
> |✓|✓|✓|**9.35\%**|**1.65\%**|**2.67\%**|**13.77\%**|**9.18\%**|**3.22\%**|
> |MBM|WEIGHT($\lambda$)|BIAS($\lambda$)|CVRPBTW|OCVRPBL|OCVRPLTW|OCVRPBLTW|Avg.gap|Best Sol.|
> |✗|✓|✓|10.74\%|16.14\%|5.57\%|18.80\%|8.37\%|3/22|
> |✓|✗|✓|9.67\%|10.67\%|5.23\%|14.49\%|5.90\%|0/22|
> |✓|✓|✗|9.63\%|10.42\%|5.51\%|13.99\%|6.26\%|0/22|
> |✓|✗|✗|9.79\%|10.02\%|5.81\%|13.98\%|5.95\%|0/22|
> |✓|✓|✓|**8.94\%**|**9.47\%**|**5.12\%**|**13.72\%**|**3.88\%**|**19/22**|

---

> ### Author Response · Authors · 2025-11-22
> **Response to Reviewer t66G [3/3]**
>
> > **Q1. Would it be possible to do the masking completely automatically, whereby the checker function is also written by an LLM?**
>
> Thank you for raising this insightful question. Our LLM-driven constraint satisfaction mechanism employs a deliberate division of responsibilities to ensure reliability:
>
> (1) The masking function acts as a controller that proactively enforces dynamic constraints at each step of solution construction. It requires look-ahead reasoning to ensure that adding any node will not cause infeasibility later in the process. Since manually designing such a function requires considerable domain expertise to ensure solution feasibility across all constraints, we use the LLM to generate it, treating its output as tentative until verified.
>
> (2) The checker function serves as a trusted verifier to validate the correctness of the LLM-generated masking function. Since implementing a checker function is inherently simpler than designing the masking function, we keep the checker as a manually written component. This human-verified process is essential, and relying on the LLM for both components would make it impossible to distinguish between a faulty masking function and an incorrect checker, thereby compromising the robustness of the scheme.
>
> > **Q2. How important is the LLM masking for URS to function? Is it needed for each VRP variant or only some?**
>
> Thank you for raising this important question. We appreciate the opportunity to clarify the role of the LLM-driven masking mechanism in our framework.
>
> - **Masking Mechanism is Fundamental for Feasibility:** Consistent with prevailing neural solvers like AM[3] and LEHD[4], a valid masking mechanism is indispensable for a model to work. It ensures that the model only selects valid nodes at each step (e.g., satisfying capacity or time window constraints), thereby guaranteeing the generation of feasible solutions.
>
> - **Automated Masking Generator:** While the masking mechanism itself is mandatory, the use of an LLM to generate these masks aims specifically to automate the process and further reduce reliance on domain expertise, which is an optional step. Users may instead rely on their own substantial expertise to implement the masking function for each constraint. However, as discussed above, manually designing the masking function is a labor-intensive process. Our LLM-driven approach automates this process, significantly lowering the barrier for non-experts to apply our solver to new problem variants by simply supplying a natural language description and a straightforward checker.
> - **Applicability:** The LLM masking pipeline is utilized for **all variants** in our experiments. Even for simpler problems where manual coding is straightforward, we utilize the LLM to maintain a unified, fully automated workflow that does not rely on predefined rule sets.
>
> > **Q3. How many iterations are needed on average for the LLM to find a valid masking?**
>
> Thank you for this important question regarding the efficiency of our method.
>
> Across the 11 training tasks in our study, we found the generation process to be highly efficient. Specifically, as shown in **Table 2** and **Table 3**, the average number of queries required to generate a valid masking function was $5$ and $2$, respectively. In all cases, a valid function was successfully generated within $20$ queries, which confirms the robustness of our approach.
>
> Table 2. Cost analysis of generating mask functions for 11 training tasks using GPT-4o-mini.
> ||ATSP|TSP|OP|PCTSP|PDTSP|OCVRPTW|
> |---|:---:|:---:|:---:|:---:|:---:|:---:|
> |\#Query|1|1|3|1|10|17|
> |\#Token|459|387|2031|650|9877|18031|
> |Time|<1min|<1min|<1min|<1min|2min|3min|
> ||**ACVRP**|**CVRP**|**CVRPTW**|**CVRPB**|**OCVRP**|**Avg.**|
> |\#Query|1|1|15|3|2|5|
> |\#Token|1273|1245|15103|1920|1264|4749|
> |Time|<1min|<1min|3min|<1min|<1min|1min|
>
> Table 3. Cost analysis of generating mask functions for 11 training tasks using GPT-3.5-turbo.
> ||ATSP|TSP|OP|PCTSP|PDTSP|OCVRPTW|
> |---|:---:|:---:|:---:|:---:|:---:|:---:|
> |\#Query|1|1|1|4|1|5|
> |\#Token|468|514|1359|3403|1277|11054|
> |Time|<1min|<1min|<1min|1min|<1min|1min|
> ||**ACVRP**|**CVRP**|**CVRPTW**|**CVRPB**|**OCVRP**|**Avg.**|
> |\#Query|1|1|5|2|1|2|
> |\#Token|1253|1371|9935|1920|1264|3074|
> |Time|<1min|<1min|<1min|<1min|<1min|<1min|
>
> > **References**
>
> [1] Matrix Encoding Networks for Neural Combinatorial Optimization, NeurIPS 2021.
>
> [2] MVMoE: Multi-Task Vehicle Routing Solver with Mixture-of-Experts, ICML 2024.
>
> [3] Attention, learn to solve routing problems! ICLR 2019.
>
> [4] Neural combinatorial optimization with heavy decoder: Toward large scale generalization, NeurIPS 2023.

---

> > ### Comment · Reviewer_t66G · 2025-11-27
> > **Satisfied with additional experiments but still not convinced by LLM for mask generation**
> >
> > I would like to thank the authors for their response and the improved version of the paper. I am satisfied with the additional experiments and the clearer explanation of the Mixed Bias Module (MBM). However, I am not fully convinced by the rationale for using an LLM for mask generation. My primary concern is that, from my understanding, the checker function is defined for each individual problem instance rather than being universal. This implies that each checker function would still need to be manually verified to ensure its validity. I find the argument that mask creation is substantially more difficult unpersuasive, as it remains unclear how much more challenging the mask creation is than creating a checker function. If the checker function were universal, I would have increased my score.

---

> ### Author Response · Authors · 2025-12-02
> **Further Clarification: The Modularity of Checkers and the Substantial Challenge of Efficient Mask Generation**
>
> We sincerely thank you for the positive feedback on our additional experiments. We greatly appreciate this opportunity to further clarify the rationale behind using an LLM, specifically addressing your concerns regarding the universality of the checker and the comparison between designing checkers and mask generators.
>
> - **Modular Reusability of Checker:** While our proposed model can handle over 100 VRP variants, we do not need to design new verification logic for every case. **The validation logic for individual constraints (e.g., capacity, time windows) is highly modular and universally reusable**. We only need to design a standard and logic-simple verification function for each of the ten constraints (e.g., checking whether the total demand on a route exceeds vehicle capacity). This checker takes a complete solution as input, and directly verifies if all constraints are satisfied, which is a straightforward binary judgment on (Yes/No) a static sequence. Furthermore, **since the validation step is performed only once per solution, efficiency is not a bottleneck, which allows for simple and intuitive implementations (e.g., Python for-loops) to check constraints sequentially**. Consequently, designing a checker for a new variant is a relatively lightweight task that involves calling these existing modular functions, requiring minimal domain expertise and implementation effort.
>
> - **Substantial Challenges in Mask Generation:** We acknowledge that for most common VRP variants, it is theoretically feasible to manually write masking functions by decoupling constraints, similar to checker construction. However, implementing these masking functions is fundamentally more challenging than performing a binary Yes/No on a completed solution. Even if the constraints can be checked individually, implementing a correct and efficient proactive look-ahead logic is a non-trivial task. Unlike a single static verification at the end, a masking function must ensure the validity of the action space at each step during the whole solution construction process. **This requires a proactive simulation to guarantee that adding a new node will not lead to an infeasible state later in the process.** Furthermore, since masking is performed stepwise during solution construction, **it demands high efficiency and must be implemented using advanced tensor operations (e.g., PyTorch batching and broadcasting) rather than simple iterative loops**. Translating logical constraints into an efficient tensor-based masking function poses a significant challenge to the user's coding capabilities and deep domain expertise, which our LLM framework successfully automates.
>
> More importantly, checking the satisfaction of each constraint individually is sufficient to guarantee the validity of the solution, which makes checker implementation straightforward and fully decouplable. However, **for more complex VRPs (e.g., PDCVRPTW), a valid masking function cannot be achieved through such constraint decoupling.** Given the inherent irreversibility of constructive generation, a node selection that appears valid in the current state might render the solution infeasible several steps later. Writing a valid mask function that can handle such coupled constraints remains a major challenge in NCO. Our established generate-and-validate pipeline represents a promising direction that could potentially address even these highly coupled problems as LLM reasoning capabilities continue to evolve.
>
> - **The Benefit of Using LLM:** We would like to emphasize again that while the masking mechanism itself is mandatory, the use of an LLM to generate these masks aims specifically to automate the process and further reduce reliance on domain expertise, which is an **optional step**. **Users may instead rely on their own substantial expertise to design the masking function for each constraint**. However, as discussed above, manually designing the masking function is a labor-intensive process. Our LLM-driven mechanism automates this complex, expert-level look-ahead logic (masking), leaving the user with only the entry-level post-hoc verification logic (checking). This significantly lowers the barrier for applying a neural solver to new unseen VRP variants by simply supplying a natural language description and a straightforward checker, aligning with the core goal of zero-shot generalization.
>
> We hope the above discussion can further clarify the difference in difficulty between checkers and mask functions design and justify the value of our LLM-driven approach.

---

### Official Review · Reviewer_kuky · 2025-10-29

**Soundness:** 3
**Presentation:** 3
**Contribution:** 3
**Rating:** 6
**Confidence:** 4

**Summary:**

URS proposes a unified neural solver along with a unified data representation (UDR) scheme for 100+ VRP variants using a single model which possess strong zero-shot generalization capability to other unseen VRP variants, prominently elevating the performance and expanding solvability scope of NCO model to a wider range of VRPs especially on current rarely-studied VRP variants (over 52 variants). Since generalization and adaptability on a wider range of VRP tasks has been an very promising and necessary direction in this field, the proposed framework in the paper offers a solid foundation for wider real-world application of NCO solver in the field of vehicle routing tasks.

**Strengths:**

1) Realistic wider application: Solving 100+ VRP variants with one model is genuinely ambitious. Most prior work handles maybe 10-20 variants with a pre-defined basket of problem tag or problem-specific adapters, so this is a real step forward for broader deployment that requires less training and expert knowledge.

2) Concrete representation of VRPs: The design of UDR is an interesting thought and makes sense as shown by many experiments-nstead of hardcoding problem types, you define a shared feature space and let different problems "activate" different features. With concrete and thorough consideration of problem settings (e.g.  multi-hot representation λ is simple and effective), UDR feels like the right level of abstraction. The combination of mixed bias module, UDR-based parameter generator and LLM-driven constraints-satisfication checker generation also ensures practicability and effectiveness of the unified network.

3) Comprehensive experiments and ablations: Testing on 96 unseen variants (including 52 brand new ones) is thorough. The ablations clearly show each component matters - removing node-type indicators or the relation matrix hurts performance where you'd expect. While the performance degradation of URS compared to URS-STL remains at an acceptable range,  the generalization capability on unseen problems is well-established in tables 10-11. I appreciate seeing results on both seen and unseen problems.

**Weaknesses:**

Reproducibility and Missing Details: Several implementation details remain unclear, which raises concerns about reproducibility. For instance, it is not specified which large language model (LLM) was used in Section 4, nor which evolutionary algorithm was employed for evolving the mask generator and constraint checker. Additionally, the number of iterations and the typical runtime for the evolutionary process are not reported. Finally, the absence of a GitHub or code repository link further limits the reproducibility of the reported results.

limitation of the proposed method is not dicussed.

**Questions:**

1.	On the difficulty of Duration-Limited and Open-Route problems:
What factors make Duration-Limit (L) and Open-Route variants more challenging for URS? As shown in Table 10, URS consistently underperforms baselines on problems with Duration Limit and Time Window constraints—for instance, OCVRPB (9.35% vs. 5.36% for ReLD-MTL) and CVRPLTW (2.67% vs. 1.17% for ReLD-MTL). Similarly, Open-Route combinations also exhibit weaker performance. These variants appear to share specific constraint patterns that might expose weaknesses in URS’s design. Could the authors analyze why these variants are harder and discuss potential architectural modifications that could better handle such constraints?
2.	On the choice of training problems:
The paper reports training on 11 specific problem types. How would the performance change if the model were trained on a broader set (e.g., 20 problems) or on a different selection of the 11? A justification of the problem selection criteria, along with a brief sensitivity analysis, would strengthen the empirical validity of the results.
3.	On the single-task version (URS-STL):
Could the authors clarify what the single-task variant (URS-STL) entails? Does it refer to training on a single problem type using the same hyperparameters and training setup as URS, or does it involve any task-specific adaptation? A clearer explanation would help interpret the reported results.

---

> ### Author Response · Authors · 2025-11-22
> **Response to Reviewer kuky [1/3]**
>
> Thank you very much for your time and effort in reviewing our work.  Below, we address each concern point by point. All revisions and additions made to the manuscript have been marked in blue to aid in your review.
>
> > **W1. Reproducibility and Missing Details**
>
> We sincerely thank you for highlighting these omissions, and we totally agree that these details are crucial for reproducibility. We apologize for not including them in the initial submission. We will add all these details to the Appendix to ensure the work is fully reproducible.
>
> - **LLM Selection:** The large language model used for the LLM-driven constraint satisfaction mechanism is GPT-3.5-torbo. We will explicitly state this in the revised manuscript.
>
> - **Evolutionary Framework:** We employ the well-established Evolution of Heuristics (EoH) framework[1] to guide the mask generation process. We have cited this work in Section 4 and Appendix A.3 as a suitable automatic heuristics design framework, and we will revise the text to make it explicit that this specific framework is used.
>
> - **Iterations and Runtime:** We apologize for the omission of these metrics. They are not fixed hyperparameters but vary with the complexity of the target constraints. Our automated pipeline is formulated as an optimization task that terminates as soon as a fully reliable mask function is found. (i.e., one that achieves a 100\% validity rate on test instances). To quantify this process, we have measured the iterations and runtime required for the 11 typical VRP variants used in our training set, recording the average number of queries and running time, as well as the specific metrics for each of the 11 problems. As shown in **Table 1**, simple problems (e.g., TSP) require only one query, while more complex and coupled problems (e.g., CVRPTW) necessitate a more extensive search.
> - **Code Availability:** We are fully committed to open science and reproducibility, and **will make both our entire code repository and the trained models publicly available on GitHub upon the acceptance of the manuscript.** This repository will include the trained URS model weights, all 100+ generated mask generators, and all the scripts necessary to reproduce our experimental results.
>
> We hope these clarifications will fully address your concerns about reproducibility.
>
> Table 1. Cost analysis of generating mask functions for 11 training tasks using GPT-3.5-turbo.
> ||ATSP|TSP|OP|PCTSP|PDTSP|OCVRPTW|
> |---|:---:|:---:|:---:|:---:|:---:|:---:|
> |\#Query|1|1|1|4|1|5|
> |\#Token|468|514|1359|3403|1277|11054|
> |Time|<1min|<1min|<1min|1min|<1min|1min|
> ||**ACVRP**|**CVRP**|**CVRPTW**|**CVRPB**|**OCVRP**|**Avg.**|
> |\#Query|1|1|5|2|1|2|
> |\#Token|1253|1371|9935|1920|1264|3074|
> |Time|<1min|<1min|<1min|<1min|<1min|<1min|
>
> > **W2. limitation of the proposed method is not discussed.**
>
> Thank you for this valuable feedback. We would like to mention that an initial discussion on "Limitation and Future Work" was already included in **Section 7 (Page 9)**, where we noted that "its current training scheme does not accommodate their inherent differences in complexity" and proposed exploring "more effective training schemes" as future work.
>
> That said, we fully appreciate your suggestion to make this discussion more comprehensive. In the revised manuscript, we will expand Section 7 to include the following additional limitations:
>
> - **UDR Representation Ambiguity:** As also mentioned in our response to Q1 below, the UDR can produce highly similar representations for problems that share similar attributes but involve significantly different dynamic constraints (e.g., CVRP vs. CVRPL ). This ambiguity may challenge the parameter generator and impair zero-shot performance on such variants.
>
> - **Training Problem Selection:** Our current method randomly selects one problem at each training step. A more promising direction for future work would be to design more effective training problem strategies, which could further enhance model performance and training efficiency.
>
> Thank you again for prompting us to strengthen this important section.

---

> ### Author Response · Authors · 2025-11-22
> **Response to Reviewer kuky [2/3]**
>
> > **Q1. On the difficulty of Duration-Limited and Open-Route problems**
>
> Thank you for this insightful observation. We fully agree with your analysis regarding the performance gap on Duration-Limited and Open-Route variants.
>
> We believe that the core challenge lies in the fact that constraints such as Open-Route (O) and Duration-Limit (L) are enforced dynamically through the masking function $\mathcal{M}_t$ during solution construction. As a result, the input attributes (i.e., node coordinates and demands) for a standard CVRP, an OCVRP, and a CVRPL are largely identical within our Unified Data Representation (UDR) framework. This is problematic because the model receives very similar input signals for tasks with diametrically opposed solution structures: OCVRP expands the search space of customer nodes, while CVRPL restricts it.
>
> We are aware of this potential ambiguity during the design phase. This is precisely why we introduced the Node-type Indicator ($\xi_i$), which includes a specific "Node in Open Route" flag to help the model distinguish O-variants from standard CVRP. However, as you correctly pointed out, the experimental results demonstrate that this mitigation is not enough. The 'Open Route' flag alone does not fully resolve the conflicting policy, and L-variants (which currently lack a dedicated flag in $\xi_i$) suffer even more from this representation ambiguity.
>
> We acknowledge that this limitation points to an important direction for future work. We plan to develop more expressive and discriminative data representations that can better capture both the shared structure and the specific differences of different VRP variants, especially for dynamically enforced constraints like O and L. Such improvements would allow the parameter generator to produce more specialized and effective policies, thereby strengthening zero-shot generalization on these challenging problem types.
>
> We sincerely appreciate your valuable comment, which has greatly inspired meaningful future directions for enhancing the URS framework.
>
> > **Q2. On the choice of training problems**
>
> Thank you for this insightful question. We acknowledge that the choice of training problems is critical for zero-shot generalization.
>
> - **Justification of Selection Criteria:** As shown in **Table 2**, our selection of the 11 training problems is guided by a "minimum redundancy" principle: we ensure that most attributes in UDR appeared in **at least two distinct problem types** during training, which helps prevent the model from overfitting a specific attribute to a single problem. The only exception is the Penalty attribute, as preliminary experiments have shown that exposure in a single problem type is sufficient for generalization to variants such as PCTSP and SPCTSP.
>
> - **Experimental Setup:** To empirically validate this selection strategy, we train two additional variants and compare them with our original URS, which are URS-Diff (Different Selection) and URS-Expand (Broader Problems). The first variant, URS-Diff, uses a different problem set where OCVRPTW is replaced by attribute-independent CVRPL. This modification means that the Open Route and Time Window attributes each appear in only one training problem. The second variant, URS-Expand, is trained on a broader set of 22 problems, incorporating all variants evaluated in MTPOMO[2] and MVMoE[3].
>
>
> - **Sensitivity Analysis:** As shown in **Table 3**, URS-Diff performs significantly worse on unseen problems involving Open Route and Time Window constraints. Meanwhile, URS-Expand achieves a slightly better performance than URS due to seeing more problem variants during training. However, the original URS remains highly competitive. Remarkably, despite being trained on only half the number of problem types (11 vs. 22), URS achieves the best solution on 12 out of the 22 evaluated problems.
>
> These results demonstrate that the choice of our selected 11 problems offers a balanced trade-off, providing sufficient attribute coverage for strong generalization while maintaining training efficiency. Expanding the training set yields only marginal gains, while a poorly selected set leads to poor cross-problem generalization. In future work, we plan to explore more effective training problem selection strategies to further enhance model performance and training efficiency.

---

> ### Author Response · Authors · 2025-11-22
> **Response to Reviewer kuky [3/3]**
>
> Table 2. The representations for seen routing problems in our experiments. The total number of problems is 11. In training, each feature in the UDR has been seen at least once. Here, RI represents the random identifier, and EAT, LAT, and ST indicate the earliest arrival time, the latest arrival time, and the service time, respectively.
> |Problem|RI|Coord.|Demand|Prize|Penalty|EAT|LAT|ST|Depot|Pickup|Delivery|Sub-routes|OpenRoute|
> |---|:---:|:---:|:---:|:---:|:---:|:---:|:---:|:---:|:---:|:---:|:---:|:---:|:---:|
> |ATSP|✓|||||||||||||
> |TSP||✓||||||||||||
> |OP||✓||✓|||||✓|||||
> |PCTSP||✓||✓|✓||||✓|||||
> |PDTSP||✓|||||||✓|✓|✓|||
> |ACVRP|✓||✓||||||✓||✓|✓||
> |CVRP||✓|✓||||||✓||✓|✓||
> |CVRPTW||✓|✓|||✓|✓|✓|✓||✓|✓||
> |CVRPB||✓|✓||||||✓|✓|✓|✓||
> |OCVRP||✓|✓||||||✓||✓|✓|✓|
> |OCVRPTW||✓|✓|||✓|✓|✓|✓||✓|✓|✓|
>
> Table 3. Comparison under different training problem settings. The symbol * indicates the original problem selection, and $\Delta$ represents a different problem selection of equal number. All test problems are encompassed within URS-Expand.
> |Method|ATSP(*$\Delta$)|TSP(*$\Delta$)|OP(*$\Delta$)|PCTSP(*$\Delta$)|PDTSP(*$\Delta$)|ACVRP(*$\Delta$)|
> |:---:|:---:|:---:|:---:|:---:|:---:|:---:|
> |URS-Diff|4.52\%|1.05\%|0.82\%|1.31\%|6.09\%|5.10\%|
> |URS-Expand|4.77\%|0.93\%|0.99\%|1.41\%|5.85\%|5.48\%|
> |URS|**2.26\%**|**0.57\%**|**0.45\%**|**1.06\%**|**4.98\%**|**3.06\%**|
> ||CVRP(*$\Delta$)|CVRPTW(*$\Delta$)|CVRPB(*$\Delta$)|OCVRP(*$\Delta$)|OCVRPTW(*)|CVRPL($\Delta$)|
> |URS-Diff|1.97\%|6.26\%|1.80\%|3.75\%|15.45\%|0.58\%|
> |URS-Expand|2.04\%|**5.62\%**|1.77\%|3.45\%|**4.34\%**|0.65\%|
> |URS|**1.81\%**|6.13\%|**1.46\%**|**3.24\%**|5.07\%|**0.43\%**|
> ||OCVRPB|CVRPBL|CVRPLTW|OCVRPBTW|CVRPBLTW|OCVRPL|
> |URS-Diff|7.55\%|2.02\%|2.76\%|27.48\%|10.25\%|3.73\%|
> |URS-Expand|**5.39\%**|1.88\%|**2.03\%**|**9.13\%**|**6.29\%**|3.41\%|
> |URS|9.35\%|**1.65\%**|2.67\%|13.77\%|9.18\%|**3.22\%**|
> ||CVRPBTW|OCVRPBL|OCVRPLTW|OCVRPBLTW|Avg.gap|Best Sol.|
> |URS-Diff|10.00\%|7.56\%|15.61\%|27.80\%|7.43\%|0/22|
> |URS-Expand|**6.07\%**|**5.47\%**|**4.38\%**|**9.23\%**|**4.12\%**|10/22|
> |URS|8.94\%|9.47\%|5.12\%|13.72\%|4.89\%|**12/22**|
>
> > **Q3. On the single-task version (URS-STL)**
>
> Thank you for requesting this clarification, which is important for interpreting the results in Table 2 of the initial manuscript.
>
> URS-STL is trained on data from only a single problem type (e.g., "URS-STL (CVRP)" is trained only on CVRP). We confirm that the URS-STL variant uses **exactly the same model architecture, hyperparameters** (e.g., embedding dimension, number of layers, optimizer), and training setup (e.g., initial learning rate, batch size) as the multi-task URS described in Section 5.1 and Appendix H .
>
> The only modification we made is to the training duration. We observed that the single-task models converge significantly faster than the multi-task model. Therefore, we train URS-STL models for 300 epochs, with a learning rate decay applied at the 251st epoch. This contrasts with the multi-task URS, which is trained for 500 epochs with decay applied at the 451st epoch.
>
> This adjustment is made solely to reflect the faster convergence of the single-task training. All other hyperparameters and model architecture remain identical. We will clarify these details in the revised paper.
>
> > **References**
>
> [1] Evolution of Heuristics: Towards Efficient Automatic Algorithm Design Using Large Language Model, ICML 2024.
>
> [2] Multi-Task Learning for Routing Problem with Zero-Shot Generalization, KDD 2024.
>
> [3] MVMoE: Multi-Task Vehicle Routing Solver with Mixture-of-Experts, ICML 2024.

---

### Official Review · Reviewer_aQrG · 2025-10-31

**Soundness:** 2
**Presentation:** 2
**Contribution:** 3
**Rating:** 4
**Confidence:** 4

**Summary:**

The paper addresses to learn solvers for vehicle routing problems with various
combinations of 10 properties and constraints (capacity, open route, backhaul,
backhaul and priority, duration limit, time windows, multi-depot, price collecting,
asymmetric distances, pickup and delivery).  The authors propose to
represent all these problems as a graph of customer nodes where each nodes
has features for all the 10 problems, set to 0, if the problem instance does
not feature this property (called "unified data representation"). An attention-based
model is used to predict the next node to add to the current tour.
Attentions are  biased by the distance between nodes (in both directions) and
a problem-specific relation between them (e.g., a priority constraint in
pickup and delivery problems) as well as a learnable factor depending
on an indicator for different problem properties and constraints (called
"multi-hot problem representation"). To mask off infeasible candidates,
an LLM is tasked with writing the code to produce the masks based on
a natural language description of the constraints. In experiments it is shown
that the proposed approach for TSP type problems works better than
the existing multitask model GOAL-MTL, while for VRP type problems
it is on par with ReLD-MTL on the problem variants it has been trained
for. On the unseen problem variants, it performs better, and it is shown
to solve many combinations other multitask solvers cannot tackle.

**Strengths:**

- s1. common representation of 10 different VRP properties and constraints.
- s2. ample experiments on over 100 combinations

**Weaknesses:**

- w1. in the evaluation, it is not clear what percentage of feasible solutions the proposed
  model achieves.
- w2. the effort required to write the constraint checkers for the 10 different VRP properties
  and constraints seems exaggerated, and thus the approach to let an LLM write them questionable.
- w3. a comparison with published results in the literature is unclear, esp. for problem-specific
  solvers.

more details:

w1. in the evaluation, it is not clear what percentage of feasible solutions the proposed
model achieves.
- As the code for the masks is written by an LLM, there is no guarantee that the constraints
  actually are met in the end. Did you evaluate how many of the solutions actually are
  feasible?

w2. the effort required to write the constraint checkers for the 10 different VRP properties
and constraints seems exaggerated, and thus the approach to let an LLM write them questionable.
- writing 10 constraint checkers seems not so much effort. You talk about constraint
  checkers for the combinations of those constraints. Are there combinations of constraints
  that pose difficulties? I.e., are not just done by checking constraint 1 and 2 individually?

w3. a comparison with published results in the literature is unclear, esp. for problem-specific
solvers.
- For example, for VRP type problems, only CVRP is compared with problem specific solvers such
  as POMO, and here POMO wins by a good margin.
- Are there other VRP type problems with established neural solvers where URS gets close?
- Can you reproduce some experiments on the established methods such as Sym-NCO or BQ
  and compare your method on their evaluation scenarios?



smaller points:
- p1. on p.3, "a feasible solution is a tour" is only valid for TSP type problems, and then m is n.
- p2. the claim to look at "more than 100 distinct VRP variants" is slightly misguiding, as all these
  variants are different combinations of 10 properties and constraints.
- p3. on p.4 "multi-hot type tensor": usually called "binary vector".
- the relation matrix R (eq. 4) comes out of the blue, it was not introduced in sec. 3.1. But as it is
  part of the common representation of  the 10 different problem aspects, it should be mentioned
  in sec. 3.1.

**Questions:**

- q1. Can you report how many of the tours the LLM authored masking mechanism
  lets through, actually are feasible?
- q2. Are there combinations of constraints that pose difficulties? I.e., are not just done
  by checking constraint 1 and 2 individually?
- q3. Are there VRP type problems beyond CVRP with established neural solvers where
  URS gets close?

---

> ### Author Response · Authors · 2025-11-22
> **Response to Reviewer aQrG [1/4]**
>
> Thank you very much for your time and effort in reviewing our work.  Below, we address each concern point by point. All revisions and additions made to the manuscript have been marked in blue to aid in your review.
>
> > **W1. in the evaluation, it is not clear what percentage of feasible solutions the proposed model achieves.**
> >
> > **Q1. Can you report how many of the tours the LLM authored masking mechanism lets through, actually are feasible?**
>
> Thank you for raising this valuable concern. In short, the feasibility rate of all solutions generated by URS is **100\%**.
>
> - **Feasibility:** In our proposed framework, each masking function is validated by a problem-specific checker on a set of test instances. Finding the valid masking function is an offline generate-and-validate process, rather than relying on an LLM to perform online reasoning during the inference phase of the neural solver. We leverage automatic heuristic design frameworks (EoH [2]) to enable the LLM to iteratively refine the code of the masking function based on the checker's feedback. Any candidate masking function that produces even one infeasible solution in the test instances (i.e., validity rate < 100\%) is discarded. Once a valid masking function is obtained, it is used in the subsequent NCO process, and the offline process is terminated. **In other words, every final valid masking function produced by the LLM is reliable, and it will generate valid masks for each solution construction step of the NCO process.** The cost of the offline process is a one-time amortized cost per problem variant, as the generated masking function can be reused indefinitely for any instance of that variant.
>
> - **Consistency Across Different LLM Settings:** To evaluate the influence of our framework on different LLMs, we benchmarked its performance using both GPT-4o-mini and GPT-3.5-turbo. The results, presented in **Table 1** and **Table 2**, show that while the query efficiency varies with the underlying model's capability, the core outcome remains consistent: our framework successfully generates the necessary masking functions with both models. This finding demonstrates that our approach doesn't rely on a specific proprietary model but rather provides a robust methodology adaptable to various powerful LLMs.
>
> Table 1. Cost analysis of generating mask functions for 11 training tasks using GPT-4o-mini.
> ||ATSP|TSP|OP|PCTSP|PDTSP|OCVRPTW|
> |---|:---:|:---:|:---:|:---:|:---:|:---:|
> |\#Query|1|1|3|1|10|17|
> |\#Token|459|387|2031|650|9877|18031|
> |Time|<1min|<1min|<1min|<1min|2min|3min|
> ||**ACVRP**|**CVRP**|**CVRPTW**|**CVRPB**|**OCVRP**|**Avg.**|
> |\#Query|1|1|15|3|2|5|
> |\#Token|1273|1245|15103|1920|1264|4749|
> |Time|<1min|<1min|3min|<1min|<1min|1min|
>
> Table 2. Cost analysis of generating mask functions for 11 training tasks using GPT-3.5-turbo.
> ||ATSP|TSP|OP|PCTSP|PDTSP|OCVRPTW|
> |---|:---:|:---:|:---:|:---:|:---:|:---:|
> |\#Query|1|1|1|4|1|5|
> |\#Token|468|514|1359|3403|1277|11054|
> |Time|<1min|<1min|<1min|1min|<1min|1min|
> ||**ACVRP**|**CVRP**|**CVRPTW**|**CVRPB**|**OCVRP**|**Avg.**|
> |\#Query|1|1|5|2|1|2|
> |\#Token|1253|1371|9935|1920|1264|3074|
> |Time|<1min|<1min|<1min|<1min|<1min|<1min|

---

> ### Author Response · Authors · 2025-11-22
> **Response to Reviewer aQrG [2/4]**
>
> > **W2. the effort required to write the constraint checkers for the 10 different VRP properties and constraints seems exaggerated, and thus the approach to let an LLM write them questionable.**
> >
> > **Q2. Are there combinations of constraints that pose difficulties? I.e., are not just done by checking constraint 1 and 2 individually?**
>
> Thank you for this critical question, which highlights a crucial distinction in our methodology: the difference between a final solution checker and the stepwise mask generator. We sincerely appreciate this opportunity to offer the following clarification:
>
> - **Verification vs. Construction:** We totally agree with you that writing post-hoc solution checkers for each VRP variant (i.e., functions that verify a completed solution) is not a significant effort. Precisely because writing these validation checkers is relatively easy, we utilize them as part of our automated generate-and-validate pipeline. The true engineering bottleneck lies in designing the stepwise mask generator itself, which is the component our LLM mechanism actually designs. This generator is far more complex than a final checker, as it must correctly handle dynamic constraints at every construction step to proactively ensure feasibility. Example codes for both the checker and masking function for CVRPTW are provided **in our revised Appendix G**.
>
> - **Reducing Domain Expertise and Automation:** For most common VRP variants, manually writing a masking function for each problem is possible by decoupling constraints. However, it is fundamentally more challenging than writing a post-hoc checker. While a checker simply validates a static, completed solution, **a masking function must proactively guarantee feasibility by determining whether a current action would render the future route infeasible.** Even if constraints can be checked individually, implementing this look-ahead logic correctly and efficiently is not a trivial task, which requires significant domain expertise for each constraint. Our mechanism automates this manual engineering task for the vast majority of variants, thereby lowering the barrier for users.
>
> - **Coupled Constraints:** In response to your question about whether there are problems where feasibility cannot be guaranteed by individually checking constraints 1 and 2, **the answer is yes.** **For more complex VRPs, constraints are coupled and cannot be checked in isolation.**
>
> One example is the Pickup and Delivery CVRP with Time Windows (PDCVRPTW), which combines PD and TW constraints. Its core challenge stems from the strict coupling between pickup and delivery pairing and time window constraints. Specifically, the selection of the next feasible node is rigorously restricted by the pending delivery obligations of visited pickup nodes, and we must ensure that the vehicle can service the corresponding delivery nodes within their valid time windows on the same route. Given the inherent irreversibility of constructive generation, a node selection that appears valid in the current state may inadvertently make mandatory future delivery nodes unreachable due to time window constraint violations. Existing approaches often rely on iterative repair strategies to transform infeasible solutions into feasible ones, rather than attempting to construct a complete feasible solution directly. Attempting to decouple and check these constraints individually is highly likely to lead to dead ends and infeasible solutions.
>
> We have explicitly tried to utilize various leading LLMs to generate a proper masking function for PDCVRPTW. While our method can automate the generation of a masking function for all VRP variants considered in this work, we found that current LLMs are not yet capable of realizing the complex coupled logic for PDCVRPTW. We leave the challenge of properly handling such complex VRPs with coupled constraints in NCO as a crucial direction for future work. Importantly, our established generate-and-validate pipeline represents a promising direction that could potentially address even these highly coupled problems as LLM reasoning capabilities continue to evolve.
>
> In summary, we do not use the LLM to write the easy checkers. Instead, we use these easy checkers to validate the output of the complicated masking function, thereby ensuring solution feasibility through a process automated by the LLM. We believe this is a necessary component for further automating the complex task of generating feasible solutions in a vast and open-ended constraint space.

---

> ### Author Response · Authors · 2025-11-22
> **Response to Reviewer aQrG [3/4]**
>
> > **w3. a comparison with published results in the literature is unclear, esp. for problem-specific solvers.**
> >
> > **q3. Are there VRP type problems beyond CVRP with established neural solvers where URS gets close?**
>
> We sincerely appreciate your valuable feedback regarding our evaluation scenarios. First, we would like to clarify that for the more than 100 problems we tested, many of them do not have comparable specialized neural solvers available. Sym-NCO and BQ are both powerful specialized solvers capable of handling multiple routing problems. We had already included them as key baselines in Table 2 **(Page 8)** of our original submission to benchmark URS across these problems. On the other hand, we fully understand the importance of verifying performance under different evaluation scenarios. To address this, we conducted a systematic evaluation using the exact scenarios from Sym-NCO[2].
>
> - **Experimental Setup:** We utilize the test datasets directly from the official Sym-NCO GitHub repository, which comprises 10,000 instances for each problem variant (i.e., TSP, CVRP, PCTSP, and OP). Following Sym-NCO, we employ an N-start rollout without applying instance augmentation for TSP and CVRP. For PCTSP and OP, the AM-based Sym-NCO employs a 200-sample approach. To ensure a fair comparison, we apply an additional coordinate transformation ($1-x$, $1-y$) to URS to match this sampling budget. Like Sym-NCO, we also tested the solvers on small-scale TSPLib data (50 < N < 250).
>
> - **Results and Analysis:** The comparison results under Sym-NCO's evaluation settings are presented in **Table 3(synthetic instances)** and **Table 4 (TSPLib)**. According to the results, URS delivers impressive performance that is comparable to or even surpasses Sym-NCO, under Sym-NCO's own evaluation scenarios. This confirms that the performance of URS is robust and not dependent on a specific evaluation setting. Unlike Sym-NCO, which is a specialized solver trained individually for each problem using distinct models (POMO for TSP and CVRP; AM for PCTSP and OP) and specific hyperparameter tuning, URS achieves competitive results across a wide range of problems with a single set of parameters, highlighting its strong generalization capability.
>
> Table 3. Performance evaluation for different problems under different evaluation scenarios.
> |Method|TSP100||CVRP100||OP100||PCTSP100||
> |---|:---:|:---:|:---:|:---:|:---:|:---:|:---:|:---:|
> ||Gap|Time|Gap|Time|Gap|Time|Gap|Time|
> |_The evaluation scenario of URS, the number of instances is 1000._|||||||||
> |_Multi-start rollout and instance augmentation*8 are both applied._|||||||||
> |Oracle Solver|0.00\%|6m|0.00\%|9.1m|0.00\%|1.5m|0.00\%|1.2h|
> |Sym-NCO|0.14\%|5s|**1.46\%**|6s|0.68\%|15s|0.14\%|16s|
> |URS-STL|**0.08\%**|6s|1.63\%|6s|**0.11\%**|4s|**-0.09\%**|5s|
> |URS|0.57\%|6s|1.81\%|6s|0.45\%|4s|1.06\%|5s|
> |_In the evaluation scenario of Sym-NCO, the number of instances is 10000._|||||||||
> |_Only multi-start rollout is applied._|||||||||
> |Oracle Solver|0.00\%|21m|0.00\%|13h|0.00\%|15m|0.00\%|12h|
> |Sym-NCO|0.39\%|8s|**1.46\%**|10s|0.45\%|2.6m|-0.02\%|2.7m|
> |URS-STL|**0.30\%**|7s|1.80\%|8s|**-0.02\%**|10.6s|**-0.06\%**|11.7s|
> |URS|1.24\%|7s|2.03\%|8s|0.50\%|10.6s|1.30\%|11.7s|

---

> ### Author Response · Authors · 2025-11-22
> **Response to Reviewer aQrG [4/4]**
>
> Table 4. Performance comparison with POMO and Sym-NCO in real-world instances in TSPLIB with scale $< 250$.
> |Instance|Opt.|POMO aug*20||Sym-NCO aug*20||URS-STL aug*8||URS aug*8||
> |:---:|:---:|:---:|:---:|:---:|:---:|:---:|:---:|:---:|:---:|
> |||Obj.|Gap|Obj.|Gap|Obj.|Gap|Obj.|Gap|
> |eil51|426|429|0.82\%|432|1.39\%|430|0.94\%|427|**0.23\%**|
> |berlin52|7,542|7,545|0.04\%|7,544|0.03\%|7,542|**0.00\%**|7,543|0.01\%|
> |st70|675|677|0.31\%|677|0.31\%|675|**0.00\%**|677|0.30\%|
> |pr76|108,159|108,681|0.48\%|108,388|0.21\%|108,159|**0.00\%**|109,089|0.86\%|
> |eil76|538|544|1.18\%|544|1.18\%|538|**0.00\%**|543|0.93\%|
> |rat99|1,211|1,270|4.90\%|1,261|4.17\%|1,212|**0.08\%**|1,252|3.39\%|
> |rd100|7,910|7,912|0.03\%|7,911|0.02\%|7,910|**0.00\%**|7,927|0.21\%|
> |KroA100|21,282|21,486|0.96\%|21,397|0.54\%|21,282|**0.11\%**|21,572|1.36\%|
> |KroB100|22,141|22,285|0.65\%|22,378|1.07\%|22,146|**0.02\%**|22,667|2.38\%|
> |KroC100|20,749|20,755|0.03\%|20,930|0.87\%|20,749|**0.00\%**|20,772|0.11\%|
> |KroD100|21,294|21,488|0.91\%|21,696|1.89\%|21,294|**0.00\%**|21,704|1.93\%|
> |KroE100|22,068|22,196|0.58\%|22,313|1.11\%|22,121|**0.24\%**|22,192|0.56\%|
> |eil101|629|641|1.84\%|641|1.84\%|630|**0.16\%**|640|1.75\%|
> |lin105|14,379|14,690|2.16\%|14,358|0.54\%|14,379|**0.00\%**|14,754|2.61\%|
> |pr124|59,030|59,353|0.55\%|59,202|0.29\%|59,030|**0.00\%**|59,942|1.54\%|
> |bier127|118,282|125,331|5.96\%|122,664|3.70\%|119,975|**1.43\%**|122,941|3.94\%|
> |ch130|6,110|6,112|**0.03\%**|6,118|0.14\%|6,116|0.10\%|6,184|1.21\%|
> |pr136|96,772|97,481|0.73\%|97,579|0.83\%|97,447|**0.70\%**|98,197|1.47\%|
> |pr144|58,537|59,197|1.13\%|58,930|0.67\%|58,673|**0.23\%**|58,981|0.76\%|
> |kroA150|26,524|26,833|1.16\%|26,865|1.28\%|26,591|**0.25\%**|27,368|3.18\%|
> |kroB150|26,130|26,596|1.78\%|26,648|1.98\%|26,237|**0.41\%**|26,900|2.95\%|
> |pr152|73,682|74,372|0.94\%|75,292|2.18\%|74,056|**0.51\%**|75,453|2.40\%|
> |u159|42,080|42,567|1.16\%|42,602|1.24\%|42,427|**0.82\%**|42,775|1.65\%|
> |rat195|2,323|2,546|9.58\%|2,502|7.70\%|2,379|**2.41\%**|2,420|4.18\%|
> |kroA200|29,368|29,937|1.94\%|29,816|1.53\%|29,677|**1.05\%**|30,419|3.58\%|
> |ts225|126,643|131,811|4.08\%|127,742|**0.87\%**|128,522|1.48\%|129,233|2.05\%|
> |tsp225|3,919|4,149|5.87\%|4,126|5.27\%|4,000|**2.15\%**|4,058|3.63\%|
> |pr226|80,369|82,428|2.56\%|82,337|2.45\%|81,707|**1.66\%**|82,049|2.09\%|
> |Avg.Gap||1.87\%||1.62\%||**0.52\%**||1.83\%||
>
> > **W4. smaller points**
>
> We sincerely thank you for these detailed and constructive suggestions. They are very helpful for improving the clarity, precision, and logical flow of our manuscript. We totally agree with all these points and will revise our paper accordingly.
>
> - **Regarding p1:** You are absolutely correct. The definition "a feasible solution is a tour" is imprecise for the general VRP context. We apologize for this oversight. We will revise this definition in Section 2.1 to "a feasible solution of a VRP calls for the determination of a set of routes, each performed by a single vehicle that starts and ends at its own depot or starting node, all the operational constraints are satisfied[3]" to accurately reflect that VRP solutions typically involve multiple vehicles.
> - **Regarding p2:** We appreciate this feedback on our phrasing. We agree that the claim "more than 100 distinct VRP variants" is not very accurate. We will revise this statement throughout the paper to explicitly state that these variants are combinations of 10 different constraints. This provides a more precise description of our experiments.
>
> - **Regarding p3:** We will replace the term "multi-hot type tensor" with "binary vector" as suggested for better clarity and consistency.
>
> - **Regarding p4:** Regarding the paper's structure, this is an excellent point for the relation matrix $ R$, it is indeed part of the problem representation and should have been introduced in Section 3.1 alongside other instance inputs. We apologize for this omission and will move its definition from Section 3.2 to Section 3.1 to improve the logical flow.
>
> Thank you very much for helping us refine these important details.
>
> > **References**
>
> [1] Evolution of Heuristics: Towards Efficient Automatic Algorithm Design Using Large Language Model, ICML 2024.
>
> [2] Sym-NCO: Leveraging Symmetricity for Neural Combinatorial Optimization, NeurIPS 2022.
>
> [3] The vehicle routing problem. Society for Industrial and Applied Mathematics, 2002.

---

> > ### Comment · Reviewer_aQrG · 2025-11-27
> >
> > Thanks to the authors for their detailed answers to my questions.
> >
> > w1. feasibility rate.
> > - That the LLM constructed feasibility checker has 100% accuracy on the test set
> >   is a good indicator for that it actually is correct.
> > - However, it is not a guarantee.
> > - I read your answer in a way that you checked this explicitly for all the solutions
> >   constructed for the evaluation. This should be made very clear in the paper.
> >
> > w2. complex feasibility checkers for coupled constraints.
> > - Thanks a lot especially for this answer. This makes many aspects way more clearer.
> > - If you make the distinction between the two types of feasibility checkers and
> >   bring the example about coupled constraints early in the paper, this will help
> >   a lot.
> >
> > w3. comparison to Sym-NCO.
> > - These are very good results!
> >
> > You answers addressed several of my concerns. I raised my score accordingly.

---

> > > ### Author Response · Authors · 2025-11-28
> > >
> > > Thank you very much for the positive feedback and the increased score. We are glad our response resolved your concerns. We will finalize the manuscript based on your helpful suggestions:
> > >
> > > w1: We confirm that all solutions were explicitly checked. We will revise the text to unambiguously state that 100% accuracy implies that every solution has been verified.
> > >
> > > w2: We fully agree. We will move the coupled constraints example to the earlier sections to aid understanding and will explicitly distinguish between the two types of feasibility checkers.
> > >
> > > w3: Thank you for the endorsement. The Sym-NCO comparison results will definitely be included in the final version.
> > >
> > > Once again, we thank you for your time and for helping us improve the quality of our work.

---

### Official Review · Reviewer_nAJG · 2025-10-31

**Soundness:** 2
**Presentation:** 3
**Contribution:** 2
**Rating:** 4
**Confidence:** 4

**Summary:**

This paper proposes URS to enhance the zero-shot generalization of neural routing solvers across various unseen VRPs. It incorporates four components: unified data representation (UDR) to broaden problem coverage; mixed bias module (MBM) to learn the relational biases in various problems; a parameter generator to adjust model parameters for generalization enhancement; and  an LLM-driven constraint satisfaction mechanism to engender stepwise masking function for solution feasibility. Experimental results show the effectiveness of the proposed method across multiple variants.

**Strengths:**

1. The paper proposes a comprehensive framework to address the zero-shot cross-problem generalization issue for VRPs, allowing fully zero-shot inference without retraining.
2. The combination of UDR, MBM, and conditional hypernetwork forms a coherent and principled framework. The paper is well-written and easy to follow.
3. Experimental results show that the proposed method can efficiently solve over 100 problem variants, 90 of which are unseen during training.

**Weaknesses:**

1. The framework is well-constructed, but its components (UDR, MBM, conditional generator) mainly combine known ideas such as feature unification and attention biasing similar to existing works. The novelty lies more in integration and scale than in new learning principles.

2. The strong zero-shot results may largely stem from the LLM-based constraint module, yet its impact is not clearly isolated. The paper also lacks analysis of mask reliability, prompt sensitivity, or consistency across different LLM settings.

3. Experiments are limited to VRPs ( ≤1000-node), leaving scalability to larger or more realistic problems (>2000 nodes) uncertain.

**Questions:**

Please refer to the weakness part.

---

> ### Author Response · Authors · 2025-11-22
> **Response to Reviewer nAJG [1/4]**
>
> Thank you very much for your time and effort in reviewing our work. Below, we address each concern point by point. All revisions and additions made to the manuscript have been marked in blue to aid in your review.
>
> > **W1. The framework is well-constructed, but the novelty lies more in integration and scale than in new learning principles.**
>
> Thank you for the constructive feedback and for acknowledging our framework as well-constructed. We fully agree with you that our work builds upon established concepts from the research community. However, we would like to highlight the specific novelties in our proposed approaches that enable URS's unprecedented zero-shot generalization, which has not been achieved by any existing works:
>
> - **Unified Data Representation:** The key innovation of our proposed UDR lies in replacing problem enumeration with data unification, which is a conceptual shift from prior multi-task solvers. It decouples instance representation from problem constraints, which delegates problem-specific constraints to the model-agnostic masking function. This design enables URS to handle an open-ended and compositional constraint space, allowing it to generalize to unseen VRP variants.
>
> - **Mixed Bias Module:** While attention biasing is a known concept in the NCO community (e.g., MatNet[1]), our MBM introduces a novel and more efficient integration. Unlike MatNet, which processes different biases ($D$ and $D^T$) in parallel attention layers, our MBM uniquely combines multiple priors ($D$, $D^T$, and $R$) within a single shared attention layer. This approach is not only more flexible (e.g., easily integrating the relation matrix $R$) but also reduces architectural redundancy.
>
> - **Conditional Parameter Generator:** Our parameter generator achieves true zero-shot generalization, a critical difference from adapter-based methods that require fine-tuning. By linking the data-driven UDR ($\lambda$) to hypernetworks (BIAS($\lambda$) and WEIGHT($\lambda$)), we create a unified decoding entrance that maps representations of unseen problems directly to all decoder parameters without retraining.
>
> In summary, while the underlying concepts (feature unification, attention biasing, and conditional generation) are not entirely new, our specific design and synergistic integration of these components are novel. It is this integrated design that enables URS to be the first single model capable of successfully solving over 100 VRP variants with zero-shot generalization.

---

> ### Author Response · Authors · 2025-11-22
> **Response to Reviewer nAJG [2/4]**
>
> > **W2. The strong zero-shot results may largely stem from the LLM-based constraint module, yet its impact is not clearly isolated. The paper also lacks analysis of mask reliability, prompt sensitivity, or consistency across different LLM settings.**
>
> Thank you for this insightful comment, which allows us to clarify the respective contributions of the URS model and the LLM-based constraint module. **To be clear, the strong zero-shot solution quality stems from the URS model, while the LLM-driven constraint satisfaction mechanism contributes to solution feasibility rather than solution quality.** The role of LLM is limited to automatically generating the masking function $\mathcal{M}_t$. Although such masking functions can be manually written by the user, which is a common practice in existing NCO methods, their design might require significant domain knowledge. Therefore, we introduce an automated masking function design framework powered by LLMs. We would like to make the following remarks for the LLM-driven constraint satisfaction mechanism:
>
> - **Reducing Domain Expertise and Automation:** For most common VRP variants, manually writing a masking function for each problem is possible by decoupling constraints. However, it is fundamentally more challenging than writing a post-hoc checker. While a checker simply validates a static, completed solution, **a masking function must proactively guarantee feasibility by determining whether a current action would render the future route infeasible.** Even if constraints can be checked individually, implementing this look-ahead logic correctly and efficiently is not a trivial task, which requires significant domain expertise for each constraint. Our mechanism automates this manual engineering task for all used variants in this paper, thereby lowering the barrier for users.
>
> - **Offline Process and Reliability:** In our proposed framework, each masking function is validated by a problem-specific checker on a set of test instances. Finding the valid masking function is an offline generate-and-validate process, rather than relying on an LLM to perform online reasoning during the inference phase of the neural solver. We leverage automatic heuristic design frameworks (i.e., EoH [2]) to enable the LLM to iteratively refine the code of the masking function based on the checker's feedback. Any candidate masking function that produces even one infeasible solution in the test instances (i.e., validity rate < 100\%) is discarded. Once a valid masking function is obtained, it is used in the subsequent NCO process, and the offline process is terminated. **In other words, every final valid masking function produced by the LLM is reliable, and it will generate valid masks for each solution construction step of the NCO process.** The cost of the offline process is a one-time amortized cost per problem variant, as the generated masking function can be reused indefinitely for any instance of that variant.
>
> - **Prompt Sensitivity:** Regarding prompt sensitivity, our approach follows the structured EoH framework [2], which inherently minimizes ambiguity. The prompt consists of two components:
>
> 1. Natural Language Task Description: This component requires a clear and accurate articulation of the task's optimization objective and all relevant operational constraints. It defines what problem to solve.
>
> 2. Code Template and I/O Specification: This component formally defines the environment, including the precise semantics, shapes, and data types of all input and output tensors. It specifies how the solution should interface with the existing code.
>
> Our generation process relies on the clarity and accuracy of these components. When the problem is well-posed with clear objectives and precise I/O, the mask function generation process is highly efficient. Conversely, failures usually result from incomplete or ambiguous specification of the core components, not minor phrasing variations.
>
> - **Consistency Across Different LLM Settings:** To evaluate the influence of our framework on different LLMs, we benchmarked its performance using both GPT-4o-mini and GPT-3.5-turbo. The results, presented in **Table 1** and **Table 2**, show that while the query efficiency varies with the underlying model's capability, the core outcome remains consistent: our framework successfully generates the necessary masking functions with both models. This finding demonstrates that our approach doesn't rely on a specific proprietary model but rather provides a robust methodology adaptable to various powerful LLMs.
>
> In summary, the strong zero-shot results are a synergy of both: the LLM enables solution feasibility, while the URS model provides solution quality. We will revise Section 4 to make the role and offline nature of the LLM module more explicit, and also add the above analysis into the revised paper.

---

> ### Author Response · Authors · 2025-11-22
> **Response to Reviewer nAJG [3/4]**
>
> Table 1. Cost analysis of generating mask functions for 11 training tasks using GPT-4o-mini.
> ||ATSP|TSP|OP|PCTSP|PDTSP|OCVRPTW|
> |---|:---:|:---:|:---:|:---:|:---:|:---:|
> |\#Query|1|1|3|1|10|17|
> |\#Token|459|387|2031|650|9877|18031|
> |Time|<1min|<1min|<1min|<1min|2min|3min|
> ||**ACVRP**|**CVRP**|**CVRPTW**|**CVRPB**|**OCVRP**|**Avg.**|
> |\#Query|1|1|15|3|2|5|
> |\#Token|1273|1245|15103|1920|1264|4749|
> |Time|<1min|<1min|3min|<1min|<1min|1min|
>
> Table 2. Cost analysis of generating mask functions for 11 training tasks using GPT-3.5-turbo.
> ||ATSP|TSP|OP|PCTSP|PDTSP|OCVRPTW|
> |---|:---:|:---:|:---:|:---:|:---:|:---:|
> |\#Query|1|1|1|4|1|5|
> |\#Token|468|514|1359|3403|1277|11054|
> |Time|<1min|<1min|<1min|1min|<1min|1min|
> ||**ACVRP**|**CVRP**|**CVRPTW**|**CVRPB**|**OCVRP**|**Avg.**|
> |\#Query|1|1|5|2|1|2|
> |\#Token|1253|1371|9935|1920|1264|3074|
> |Time|<1min|<1min|<1min|<1min|<1min|<1min|
> > **W3. Experiments are limited to VRPs ( ≤1000-node), leaving scalability to larger or more realistic problems (>2000 nodes) uncertain.**
>
> Thank you for this valuable suggestion regarding scalability. We totally agree that it is a critical aspect for practical neural solvers. To thoroughly address this concern, we have conducted extensive new experiments on several large-scale benchmarks and synthetic datasets: (1) The large-scale CVRPLib XXL benchmark dataset (scale $\in [3000,7000]$); and (2) a new large-scale (1K-5K nodes) synthetic datasets for a diverse set of problems (CVRPTW, CVRPB, and OCVRPTW) to explicitly test the large-scale cross-problem generalization ability. As shown in the table below, we can observe:
>
> - **Large-scale CVRPLib Benchmark:** We evaluated URS on the CVRPLib XXL benchmark with scale $\in [3000,7000]$. The results (**Table 3**) clearly demonstrate that URS can significantly outperform existing multi-task neural routing solvers, especially on large-scale instances. Notably, URS also greatly surpasses many representative specialized models (e.g., LEHD, BQ, and ELG), underscoring its practical applicability in real-world scenarios.
>
> - **Large-scale VRP Variants:** Due to the limited benchmark datasets currently available for VRP variants, we have conducted further experiments on large-scale synthetic problem instances to validate the generalization performance of URS. For all instances, we adhere to the capacity constraints specified in TAM[3]. To ensure a fair comparison, HGS-PyVRP is executed with multi-core (16) parallelism enabled, which allows it to process all instances of a specific size concurrently. According to the results (**Table 4**),  our proposed URS consistently achieves the best solution quality, complemented by remarkably fast inference times, across various problem instances among all comparable neural solvers. The advantages of URS over existing multi-task neural solvers are significant. Remarkably, **URS already outperforms the strong HGS-PyVRP with carefully manual design on large-scale CVRPB instances.**
>
> These extensive new results demonstrate URS's strong scalability and generalization ability to large-scale and realistic problems. Thank you very much again for this suggestion, which substantially strengthens our paper's contribution.
>
> Table 3. Comparison on CVRPLib Set-XXL with scale $\in [3000,7000]$. All models are trained on instances of size N = 100.
> |Method||Type|Leuven1|Leuven2|Antwerp1|Antwerp2|Total||
> |---|---|:---:|:---:|:---:|:---:|:---:|:---:|:---:|
> ||||(N=3000)|(N=4000)|(N=6000)|(N=7000)|N∈[3000,7000]||
> |LEHD greedy||Specialized|16.60\%|34.86\%|14.66\%|22.77\%|22.22\%||
> |BQ greedy||Specialized|18.53\%|30.70\%|16.48\%|27.67\%|23.34\%||
> |POMO aug*8||Specialized|460.32\%|202.17\%|OOM|OOM|$-$||
> |Sym-NCOaug*8||Specialized|196.82\%|328.17\%|OOM|OOM|$-$||
> |ELG aug*8||Specialized|12.12\%|21.52\%|OOM|OOM|$-$||
> |MTPOMO aug*8||General|67.72\%|87.31\%|OOM|OOM|$-$||
> |MVMoE/4E aug*8||General|299.10\%|170.10\%|OOM|OOM|$-$||
> |MVMoE/4E-Laug*8||General|182.28\%|127.07\%|OOM|OOM|$-$||
> |RF-MVMoE aug*8||General|57.30\%|160.27\%|OOM|OOM|$-$||
> |RF-TF aug*8||General|26.90\%|45.56\%|OOM|OOM|$-$||
> |CaDA aug*8||General|1035.51\%|OOM|OOM|OOM|$-$||
> |ReLD-MTL aug*8||General|17.00\%|30.59\%|OOM|OOM|$-$||
> |GOAL greedy||General|OOM|OOM|OOM|OOM|$-$||
> |URS aug*8||General|**11.57\%**|**17.80\%**|**9.23\%**|**14.95\%**|**13.39\%**||

---

> ### Author Response · Authors · 2025-11-22
> **Response to Reviewer nAJG [4/4]**
>
> Table 4. Generalization performance on large-scale instances of different VRP variants. All models are trained on instances of size N = 100 and employ instance augmentation $ \times$ 8. Each dataset includes 16 instances. Note that "Time" is the total running time, and HGS-PyVRP is executed with multi-core (16) parallelism enabled.
> |Problem|Method|N=1000||N=2000||N=3000||N=4000||N=5000||
> |:---:|:---:|:---:|:---:|:---:|:---:|:---:|:---:|:---:|:---:|:---:|:---:|
> |||Obj.(Gap)|Time|Obj.(Gap)|Time|Obj.(Gap)|Time|Obj.(Gap)|Time|Obj.(Gap)|Time|
> |CVRPTW|HGS-PyVRP|159.35(0.00\%)|20m|337.15(0.00\%)|40m|516.97(0.00\%)|1h|618.58(0.00\%)|2.7h|787.25(0.00\%)|3.3h|
> ||MTPOMO|219.22(37.57\%)|1.7m|488.40(44.86\%)|14.6m|773.00(49.53\%)|49.5m|975.61(57.72\%)|1.9h|OOM||
> ||MVMoE|233.53(46.55\%)|1.7m|624.73(85.30\%)|14.6m|1094.25(111.67\%)|51.9m|1486.34(140.28\%)|2h|OOM||
> ||ReLD-MTL|183.31(15.036\%)|1.6m|396.28(17.54\%)|13.4m|613.58(18.69\%)|45.7m|748.65(21.03\%)|1.8h|OOM||
> ||URS-MTL|**172.58(8.30\%)**|**1.3m**|**365.27(8.34\%)**|**11.1m**|**559.03(8.14\%)**|**30.1m**|**673.63(8.90\%)**|**1.2h**|**853.79(8.45\%)**|**2.3h**|
> |CVRPB|HGS-PyVRP|36.52(0.00\%)|20m|51.80(0.00\%)|40m|75.12(0.00\%)|1h|93.52(0.00\%)|2.7h|112.21(0.00\%)|3.3h|
> ||MTPOMO|45.92(25.77\%)|1.1m|83.50(61.20\%)|8.8m|136.52(81.73\%)|31.4m|197.99(111.70\%)|1.2h|OOM||
> ||MVMoE|98.60(170.03\%)|1m|250.47(383.57\%)|8.1m|332.93(343.19\%)|28.9m|423.19(352.49\%)|1.1h|OOM||
> ||ReLD-MTL|38.52(5.48\%)|1m|59.21(14.30\%)|8m|88.36(17.62\%)|28.3m|114.88(22.83\%)|1.1h|OOM||
> ||URS-MTL|**34.86(-4.54\%)**|**40s**|**50.39(-2.72\%)**|**5.1m**|**72.01(-4.15\%)**|**17.3m**|**91.74(-1.91\%)**|**41.2m**|**111.93(-0.24\%)**|**1.3h**|
> |OCVRPTW|HGS-PyVRP|90.91(0.00\%)|20m|164.00(0.00\%)|40m|224.16(0.00\%)|1h|299.45(0.00\%)|2.7h|367.86(0.00\%)|3.3h|
> ||MTPOMO|147.14(61.85\%)|1.6m|297.53(81.43\%)|12.8m|437.73(95.28\%)|44.3m|601.63(100.91\%)|1.7h|OOM||
> ||MVMoE|147.35(62.09\%)|1.6m|389.60(137.57\%)|12.9m|634.11(182.88\%)|44.7m|899.18(200.28\%)|1.7h|OOM||
> ||ReLD-MTL|110.00(21.01\%)|1.5m|208.60(27.20\%)|12.3m|292.70(30.57\%)|42m|394.42(31.72\%)|1.6h|OOM||
> ||URS-MTL|**98.76(8.63\%)**|**1m**|**178.07(8.58\%)**|**7.6m**|**243.88(8.80\%)**|**25.4m**|**325.64(8.74\%)**|**1h**|**398.72(8.39\%)**|**1.9h**|
>
> > **References:**
>
> [1] Matrix Encoding Networks for Neural Combinatorial Optimization, NeurIPS 2021.
>
> [2] Evolution of Heuristics: Towards Efficient Automatic Algorithm Design Using Large Language Model, ICML 2024.
>
> [3] Generalize Learned Heuristics to Solve Large-scale Vehicle Routing Problems in Real-time, ICLR 2023.

---

### Meta-Review · Area_Chair_Xgfx · 2026-01-01

**Summary:**

This paper proposes a unified neural routing solver named URS for zero-shot generalization across a wide range of unseen VRPs using a single model without any fine-tuning. Extensive experiments show that URS can consistently produce high-quality solutions for more than 100 distinct VRP variants composed of 10 different constraints without any fine-tuning.

Reviewers acknowledged this work has some value, but also raised several major concerns, e.g., the limited novelty (as most components of the proposed method are the combination of known ideas), the necessity of the LLM masking, and the experiments limited to VRPs with no more than 1000 nodes. The authors gave detailed responses, and added additional experimental results. Some concerns have been addressed, but the novelty issue may remain.

**Reviewer Concerns:**

Reviewer nAJG has major concerns: 1) limited novelty as most components are the combination of known ideas; 2) lack of clear analysis of different components; 3) experiments limited to VRPs with no more than 1000 nodes.

The authors gave detailed responses, and added additional experimental results. I think some concerns (e.g., the experiments on instances with larger size) can be addressed to some extent, but the concern on the limited novelty of the proposed method may remain.

Reviewer aQrG has major concerns: 1) percentage of feasible solutions the proposed model achieves; 2) the approach to let an LLM write them seems questionable; 3) unclear comparison with published results in the literature.

The authors’ response has addressed several concerns of the reviewer. Reviewer aQrG considers increasing the score accordingly.

Reviewer kuky mainly has major concerns on reproducibility and missing details. The authors provided detailed response, and may address the concerns.

Reviewer t66G has major concerns on the LLM masking and the model architecture. The authors’ response has addressed some concerns, but the concern on the LLM masking remains.

**Reviewer Scores:**

I think Reviewer nAJG may not change the score, as the authors’ rebuttal may not change her/his concern on the limited novelty of the proposed method.

Reviewer aQrG considers increasing the score accordingly, as the authors’ response has addressed several concerns of the reviewer.

I think Reviewer kuky may keep the positive score, as the authors provided detailed response to the questions and weakness.

Reviewer t66G may increase the score if the authors’ further response on the LLM masking is convincing.

---

### Decision · Program_Chairs · 2026-01-26

Reject